

**Estimating the lateral transfer of organic carbon through the European river**
**network using a land surface model**
Haicheng Zhang[1*], Ronny Lauerwald[2], Pierre Regnier[1], Philippe Ciais[3], Kristof Van Oost[4],
Victoria Naipal[5], Bertrand Guenet[3], Wenping Yuan[6]
[1]Department Geoscience, Environment & Society-BGEOSYS, Université libre de Bruxelles, 1050 Bruxelles,
Belgium
[2] Université Paris-Saclay, INRAE, AgroParisTech, UMR ECOSYS, 78850, Thiverval-Grignon, France
[3]Laboratoire des Sciences du Climat et de l'Environnement, IPSL-LSCE CEA/CNRS/UVSQ, Orme des Merisiers,
91191, Gif sur Yvette, France
[4]UCLouvain, TECLIM - Georges Lemaître Centre for Earth and Climate Research, Louvain-la-Neuve, Belgium
[5]EcoAct/ ATOS, 35 rue de miromesnil, 75008, Paris, France
[6]School of Atmospheric Science, Sun Yat-sen University, Guangzhou, Guangdong, 510275, China
*Correspondence to*: Haicheng Zhang (*haicheng.zhang@ulb.be*)





**Abstract.** Lateral carbon transport from soils to the ocean through rivers has been acknowledged
as a key component of global carbon cycle, but is still neglected in most global land surface
models (LSMs). Fluvial transport of dissolved organic carbon (DOC) and $CO_2$ has been
implemented in the ORCHIDEE LSM, while erosion-induced delivery of sediment and
particulate organic carbon (POC) from land to river was implemented in another version of the
model. Based on these two developments, we take the final step towards the full representation
of biospheric carbon transport through the land-river continuum. The newly developed model,
called ORCHIDEE-C$_{lateral}$, simulates the complete lateral transport of water, sediment, POC,
DOC and $CO_2$ from land to sea through the river network, the deposition of sediment and POC in
the river channel and floodplains, and the decomposition of POC and DOC in transit. We
parameterized and evaluated ORCHIDEE-C$_{lateral}$ using observation data in Europe. The model
satisfactorily reproduces the observed riverine discharges of water and sediment, bankfull flows
and sediment delivery rate from land to river, as well as the observed concentrations of organic
carbon in rivers. Application of ORCHIDEE-C$_{lateral}$ for Europe reveals that the lateral carbon
transfer affects land carbon dynamics in multiple ways and omission of this process in LSMs
may result in significant biases in the simulated regional land carbon budgets. Overall, this study
presents a useful tool for simulating large scale lateral carbon transfer and for predicting the
feedbacks between lateral carbon transfer and future climate and land use changes.


## 1 Introduction

Lateral transfer of organic carbon along the land-river-ocean continuums, involving both spatial redistribution of terrestrial organic carbon and the vertical land-atmosphere carbon exchange, has been acknowledged as a key component of the global carbon cycle (Ciais et al., 2013; Ciais et al., 2021; Drake et al., 2018; Regnier et al., 2013). Erosion of soils and the associated organic carbon, but also leaching of dissolved organic carbon (DOC), represent a non-negligible leak in the terrestrial carbon budget and a substantial source of allochthonous organic carbon to inland waters and oceans (Battin et al., 2009; Cole et al., 2007; Raymond et al., 2013; Regnier et al., 2013). As a result of soil aggregate breakdown and desorption, the accelerated mineralization of these eroded and leached soil carbon loads leads to considerable $CO_2$ emission to the atmosphere (Chappell et al., 2016; Lal, 2003; Van Hemelryck et al., 2011). Meanwhile, the organic carbon that is redeposited and buried in floodplains and lakes might be preserved for a long time, thus creating a $CO_2$ sink (Stallard, 1998; Van Oost et al., 2007; Wang et al., 2010). In addition, lateral redistribution of soil material can alter land-atmosphere $CO_2$ fluxes indirectly by affecting soil nutrient availability, terrestrial vegetation productivity and physiochemical properties of inland and coastal waters (Beusen et al., 2005; Vigiak et al., 2017).

Although the important role of lateral carbon transfer in the global carbon cycle has been widely recognized, to date, the estimates of land carbon loss to inland waters, the fate of the terrestrial organic carbon within inland waters, as well as the net effect of lateral carbon transfer on land-atmosphere $CO_2$ fluxes remain largely uncertain (Berhe et al., 2007; Doetterl et al., 2016; Lal, 2003; Stallard, 1998; Wang et al., 2014b; Zhang et al., 2014). Existing estimates of global carbon loss from soils to inland waters vary from 1.1 to 5.1 Pg (=$10^{15}$ g) C per year (yr$^{-1}$) (Cole et al., 2007; Drake et al., 2018), and the estimated net impact of global lateral carbon redistribution on land-atmosphere carbon budget ranges from an uptake of atmospheric $CO_2$ by 1 Pg C yr$^{-1}$ to a land $CO_2$ emission of 1 Pg C yr$^{-1}$ (Lal, 2003; Stallard, 1998; Van Oost et al., 2007; Wang et al., 2017). A reliable model which is able to explicitly simulate the lateral carbon along the land-river continuum and also the interactions between these lateral processes and the comprehensive terrestrial carbon cycle, would thus be necessary for predicting changes in the global carbon cycle more accurately.





Global land surface models (LSMs) are important tools to simulate the feedbacks between
terrestrial carbon cycle, increasing atmospheric $CO_2$, and climate and land use change. However,
the lateral carbon transfer, especially for the particulate organic carbon (POC), is still missing or
incompletely represented in existing LSMs (Lauerwald et al., 2017; Lauerwald et al., 2020;
Lugato et al., 2016; Naipal et al., 2020; Nakhavali et al., 2021; Tian et al., 2015). It has been
hypothesized that the exclusion of lateral carbon transfer in LSMs implies a significant bias in
the simulated global land carbon budget (Ciais et al., 2013; Ciais et al., 2021; Janssens et al.,
2003). For instance, the study of Nakhavali et al. (2021) suggested that about 15% of the global
terrestrial net ecosystem production is exported to inland waters as leached DOC. Lauerwald et
al. (2020) showed that the omission of lateral DOC transfer in LSM might lead to significant
underestimation (8.6%) of the net uptake of atmospheric carbon in the Amazon basin while
terrestrial carbon storage changes in response to the increasing atmospheric $CO_2$ concentrations
were overestimated.
Over the past decade, a number of LSMs has been developed which represent leaching of DOC
from soils (Nakhavali et al. 2018, Kicklighter et al. 2013) or the full transport of DOC through
the land-river continuum (Lauerwald et al., 2017; Tian et al., 2015). However, the erosion-
induced transport of POC, which is maybe even more important than the DOC transport in terms
of lateral carbon flux (Lal., 2003; Tian et al., 2015; Tan et al., 2017), is still not or poorly
represented in LSMs. The explicit simulation of the complete transport process of POC at large
spatial scales is still a major challenge, due to the complexity of the  processes involved,
including  erosion-induced sediment and POC delivery to rivers, deposition of sediment and
POC in river channels and floodplains, re-detachment of the previously deposited sediments and
POC, decomposition and transformation of POC in riverine and flooding waters, as well as the
changes of soil profile caused by erosion and deposition (Doetterl et al., 2016; Naipal et al.,
2020; Zhang et al., 2020).
Several recent model developments have led to the implementation of the lateral transfer of POC
in large-scale LSMs. Despite this, there are still some inevitable limitations in these
implementations. The Dynamic Land Ecosystem Model (DLEM v2.0, Tian et al., 2015) is able
to simulate the erosion-induced POC loss from soil to river and the transport and decomposition
of POC in river networks. However, it does not represent the POC deposition in floodplains, nor



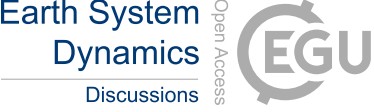

the impacts of soil erosion and floodplain deposition on the vertical profiles of soil organic
carbon (SOC). The Carbon Erosion DYNAMics model (CE-DYNAM, Naipal et al., 2020)
simulates erosion of SOC and its re-deposition on the toe-slope or floodplains, transport of POC
along river channels, as well as the impact on SOC dynamics at the eroding and deposition sites.
However, running at annual time scale, it mostly addresses the centennial timescale and does not
represent deposition and decomposition of POC in river channels. Moreover, CE-DYNAM was
only applied over the Rhine catchment and has not been fully coupled into a land surface model,
therefore excluding the feedbacks of soil erosion on the fully coupled land and aquatic carbon
cycles. There are of course more dedicated hydrology and soil erosion models that explicitly
simulate the complete transport, deposition and decomposition processes of POC in small river
basins (e.g. Jetten et al., 2003; Nearing et al., 1989; Neitsch et al., 2011). However, it is difficult
to apply these models at large spatial scales (e.g. continental or global scale) due to the limited
availability of forcing data (e.g. geometric attributes of river channel), suitable model
parameterization and computational capacity. Moreover, these models have limited capability of
representing the full terrestrial C cycle in response to climate change, increasing atmospheric
$CO_2$ and land use change. Therefore, basin-scale models are not an option to assess the impact of
soil erosion on the large-scale terrestrial C budget in response to global changes.
Here we describe the development, application and evaluation of a new branch of the
ORCHIDEE LSM (Krinner et al., 2005), hereafter ORCHIDEE-$C_{lateral}$, that can be used to
simulate the complete lateral transfer processes of water, sediment, POC and DOC along the
land-river-ocean continuum at large spatial scale (e.g. continental and global scale). In previous
studies, the leaching and fluvial transfer of DOC and the erosion-induced delivery of sediment
and POC from upland soil to river network have been implemented in two different branches of
the ORCHIDEE LSM (Lauerwald et al., 2017; Zhang et al., 2020). For this new branch, we first
merged these two branches, and subsequently implemented the fluvial transfer of sediment and
POC in the coupled model. ORCHIDEE-$C_{lateral}$ is calibrated and evaluated using observation data
of runoff, bankfull flow, and riverine loads and concentrations of sediment, POC and DOC
across Europe. By applying the calibrated model at European scale, we estimate the magnitude
and spatial distribution of the lateral carbon transfer in European catchments during the period
1901-2014, as well as the potential impacts of lateral carbon transfer on the land carbon balance.
Comparing simulations results to those of an alternative simulation run with lateral displacement



of C deactivated, we finally quantify the biases in simulated land C budgets that arise ignoring
the lateral transfers of C along the land-river continuum.

**2 Model development and evaluation**
**2.1 ORCHIDEE land surface model**
The ORCHIDEE LSM comprehensively simulates the cycling of energy, water and carbon in
terrestrial ecosystems (Krinner et al., 2005). The hydrological processes (e.g. rainfall
interception, evapotranspiration and soil water dynamics) and plant photosynthesis in
ORCHIDEE are simulated at a time step of 30 minutes. The carbon cycle processes (e.g.
maintenance and growth respiration, carbon allocation, litter decomposition, SOC dynamics,
plant phenology and mortality) are simulated at daily time step. In its default configuration,
ORCHIDEE represents vegetation by 13 plant functional types (PFTs), with eight PFT for
forests, two for grasslands, two for croplands, and one for bare soil. Given appropriate land cover
maps and parametrization, the number of PFTs to be represented can however be adapted (Zhang
et al., 2020).
Our previous implementations of lateral DOC transfer (Lauerwald et al., 2017) and of POC
delivery from upland to river network (Zhang et al., 2020) were both based on the ORCHIDEE
branch ORCHIDEE-SOM (Camino-Serrano et al., 2018), which provides a depth-dependent
description of the water and carbon dynamics in soil column. In specific, the vertical soil profile
in ORCHIDEE-SOM is described by an 11-layer discretization of a 2 m soil column (Camino-
Serrano et al., 2018). Water flows between adjacent soil layers are simulated using the Fokker–
Planck equation that resolves water diffusion in non-saturated conditions (Campoy et al., 2013;
Guimberteau et al., 2018). Free gravitational drainage occurs in the lowest soil layer when actual
soil water content is higher than the residual water content (Campoy et al., 2013). Following the
CENTURY model (Parton et al., 1988), ORCHIDEE-SOM subdivides the particulate organic
carbon stored in soil into two litter pools (metabolic and structural) and three SOC pools (active,
slow and passive) that differ in their respective turnover times. The decomposition of each
carbon pool is calculated by first order kinetics based on the corresponding turnover time, soil
moisture and temperature as controlling factors, as well as the priming effects of fresh organic
matter (Guenet et al., 2018; Guenet et al., 2016). Soil DOC is represented by a labile and a stable





DOC pools, with a high and low turnover rate, respectively. Each DOC pool may be in the soil
solution or adsorbed on the mineral matrix. The products of litter and SOC decomposition go to
free DOC, which in turn is decomposed following first order kinetics (Kalbitz et al., 2003) and
returns back to SOC. "The free DOC can then be adsorbed to soil minerals or remain in solution
following an equilibrium distribution coefficient (Nodvin et al., 1986), which depends on soil
properties (clay and pH). Adsorbed DOC is assumed to be protected and thus is neither
decomposed nor transported within the soil column. Free DOC is subject to transport with the
water flux between layers calculated by the soil hydrological module of ORCHIDEE, i.e., by
advection. Also, SOC and DOC are subject to diffusion that is represented using the second
Fick's law of diffusion" (Camino-Serrano et al., 2018, p. 939). All the described processes occur
within each soil layer. At each time step, "the flux of DOC leaving the soil is calculated by
multiplying DOC concentrations in soil solution with the runoff (surface layer) and drainage
(bottom layer) flux simulated by the hydrological module" (Camino-Serrano et al., 2018, p. 939).
More detailed information about the simulation of soil hydrological and biogeochemical
processes in ORCHIDEE-SOM can be found in Guenet et al. (2016) and Camino-Serrano et al.

168    (2018).

**2.1.1 Lateral transfer of DOC and $CO_2$**
Lateral transfer of DOC and dissolved $CO_2$ from land to ocean through river network has been
implemented in the ORCHILEAK (Lauerwald et al., 2017), an ORCHIDEE branch developed
from ORCHIDEE-SOM. The adsorption, desorption, production, consumption and transport of
DOC within the soil column, as well as DOC export from soil along with surface runoff and
drainage in ORCHILEAK is simulated using the same method as ORCHIDEE-SOM. Besides the
decomposition of SOC and litter, ORCHILEAK also represents the contribution of wet and dry
deposition to soil DOC via throughfall. The direct DOC input from rainfall to aquatic DOC pools
is simulated based on the DOC concentration in rainfall and the area fraction of stream and
flooding waters in each basin. Simulation of the lateral transfer of DOC and $CO_2$ in river
networks, i.e. the transfer of DOC and $CO_2$ from one basin to another based on the stream flow
directions obtained from forcing file (0.5°, Table 1), follows the routing scheme of water
(Guimberteau et al., 2012). For each basin with floodplain (defined by forcing data), bankfull
flow occurs when stream volume in the river channel exceeds a threshold prescribed by the





forcing file (Table 1). DOC and $CO_2$ in flooding waters can enter into soil DOC and $CO_2$ pools
along with the infiltrating water. On the contrary, DOC and $CO_2$ originated from the
decomposition of submerged litter and SOC in the floodplains are added to the overlying
flooding waters. Note that the turnover times of litter and SOC under flooding waters are
assumed to be three times of the litter and SOC turnover times in upland soil (Reddy & Patrick
Jr, 1975; Neckles & Neill, 1994; Lauerwald et al., 2017). After removing the infiltrated and
evaporated water, the amount of the remaining flooding water, as well as the DOC and dissolved
$CO_2$ returning to river channel at the end of each day is calculated based on a time constant of
flooding water (= 4.0 days, d'Orgeval et al., 2008) modified by basin-specific topographic index
($f_{topo}$, unitless) (Lauerwald et al., 2017).

**Table 1.** List of forcing data needed to run ORCHIDEE-$C_{lateral}$ and the data used to evaluate the
simulation results. $S_{res}$ and $T_{res}$ are the spatial and temporal resolution of the forcing data,
respectively.

| | Data | $S_{res}$ | $T_{res}$ | Data source |
|---|---|---|---|---|
| Forcing | Climatic forcing data (precipitation, temperature, incoming shortwave/longwave radiation, air pressure, wind speed, relative humidity) | 0.5° | 3 hour | GSWP3 database (Dirmeyerm et al., 2006) |
| | Land cover | 0.5° | 1 year | LUHa.rc2 database (Chini et al., 2014) |
| | Soil texture class | 0.5° | – | Reynolds et al. (1999) |
| | Soil bulk density and pH | 30″ | – | HWSD v1.2 (FAO/IIASA/ISRIC/ISSCAS/JRC, 2012) |
| | Stream flow directions, topographic index ($f_{topo}$) | 0.5° | – | STN-30p (Vörösmarty et al., 2000) |
| | Area fraction of floodplains | 250 m | – | GFPLAIN250m (Nardi et al., 2019)[a] |
| | Area fraction of river surface | 0.5° | – | Lauerwald et al. (2015) |
| | Maximum water storage in river channel ($S_{rivmax}$) | 0.5° | – | Derived from pre-runs with ORCHIDEE-$C_{lateral}$ (see section 2.3) |
| | Reference sediment delivery rate ($SED_{ref}$) | 0.5° | – | Zhang et al. (2020) |
| | Digital Elevation Model (DEM) | 3″ | – | HydroSHEDS (Lehner et al., 2008) and GDEM v3 (Abrams et al., 2020)[b] |
| Validation | Riverine water discharge | – | 1 day | GRDC[c] |
| | Bankfull flow | – | 1 year | Schneider et al. (2011) |
| | Sediment delivery from upland to inland waters | 100 m | 1 year | Borrelli et al. (2018) |
| | Riverine sediment discharge | – | 1 year | European Environment Agency[d] and publications[e] |
| | Riverine POC and DOC concentration | – | Instantaneous | GLORICH (Hartmann et al., 2019) |
| | | 30″ | | HWSD v1.2 |
| | SOC stock | 5′ | – | GSDE (Shangguan et al., 2014) |
| | | 250 m | | SoilGrids (Hengl et al., 2014) |



| | 10 km | S2017 (Sanderman et al., 2017) |
| | 250 m | LandGIS[f] |

[a] The GFPLAIN250m only covers the regions south of 60° N. We produced map of floodplain distribution in
regions north of the 60° N using the same method for producing GFPLAIN250m (Nardi et al., 2019) based on the
ASTER GDEM v3 database (Abrams et al., 2020). [b] The DEM data from HydroSHEDS and GDEM v3 are used to
extract the topographic properties (e.g. location, area and average slope) of headwater basins in regions south and
north of 60° N, respectively. [c] The Global Runoff Data Centre, 56068 Koblenz, Germany. [d]
https://www.eea.europa.eu/data-and-maps/data/sediment-discharges. [e] Publications including Van Dijk & Kwaad,
1998; Vollmer & Goelz, (2006) and Reports of the DanubeSediment project (Sediment Management Measures for
the Danube, http://www.interreg-danube.eu/approved-projects/danubesediment). [f]
https://zenodo.org/record/2536040#.YC-QGo9KiUm.

DOC decomposition and $CO_2$ evasion in inland waters are simulated using a much fine
integration time step of 6 minutes. The decomposition of DOC in stream and flooding waters is
calculated based on the prescribed turnover times of labile (2 days) and refractory (80 days)
DOC in waters (when temperature is 28 °C) and a temperature factor obtained from Hanson et al.
(2011). As described in Lauerwald et al. (2017), besides $CO_2$ originated from fluvial DOC,
"dissolved $CO_2$ inputs from the decomposition from flooded SOC and litter are also added at the
time step of 6 minutes to represent the continuous additions of $CO_2$ during the water–atmosphere
gas exchange. For each time step, the $CO_2$ partial pressures ($pCO_2$) in the water column is
calculated from the concentration of dissolved $CO_2$ and the temperature-dependent solubility of
$CO_2$ (Telmer and Veizer, 1999). The $CO_2$ evasion is finally calculated based on the water–air
gradient in $pCO_2$, the gas exchange velocity and the surface water area available for gas
exchange" (p. 3835). In addition, swamp and wetland are also represented in the routing scheme
of ORCHILEAK. More detailed descriptions can be found in Lauerwald et al. (2017).
**2.1.2 Sediment and carbon delivery from upland soil to river network**
Using an upscaling scheme, the erosion-induced sediment and POC delivery from upland soil to
river network, as well as the dynamics of vertical SOC distribution due to soil erosion had
already been implemented in ORCHIDEE-MUSLE (Zhang et al., 2020). The sediment delivery
from small headwater basins to river network (i.e. gross upland soil erosion – sediment
deposition within headwater basins) is simulated using the Modified Universal Soil Loss
Equation model (MUSLE, Williams, 1975). For the upscaling, MUSLE is first applied to high-
resolution (3″) topographic and soil erodibility data. As introduced in Zhang et al. (2020), "the





daily sediment delivery rate from each headwater basin ($S_{i\_ref}$, Mg day⁻¹ basin⁻¹) is first calculated
for a given set of reference runoff and vegetation cover conditions:
$$S_{i_{ref}} = a\left(Q_{i_{ref}} q_{i_{ref}}\right)^{b} K_i LS_i C_{ref} P_{ref} \qquad (1)$$

where $Q_{i\_ref}$ is the total water discharge (m³ day⁻¹) at the outlet of headwater basin $i$ for the daily
reference runoff condition ($R_{ref}$) of 10 mm day⁻¹. In Eq. 1, $q_{i\_ref}$ is the daily peak flow rate (m³ s⁻¹)
at the headwater basin outlet under the assumed reference runoff condition. Similar to the SWAT
model (Soil and Water Assessment Tool, Neitsch et al., 2011), $q_{i\_ref}$ was calculated from the
reference maximum 30-minutes runoff (= 1 mm 30-minutes⁻¹) depth and drainage area according
to the following equation:
$$q_{i\_ref} = \frac{R_{30\_ref}}{30 \times 60}\left(DA_i^{(d\ DA_i^c)}\right) 1000 \qquad (2)$$

where $R_{30\_ref}$ (= 1 mm 30-minutes⁻¹) is the assumed daily maximum 30-minutes runoff" (p. 5-6).
The coefficients $a$ and $b$ in Eq. 1 and $c$ and $d$ in Eq. 2 need to be calibrated (see section 2.3 and
Table A1). In Eq. 1, the term $LS_i$ is the combined dimensionless slope length and steepness factor
calculated based on the $DA_i$ and the average slope steepness (extracted from DEM) of headwater
basin $i$ (Moore and Wilson, 1992). $C_{ref}$ (0-1, dimensionless) in Eq. 1 represents the cover
management factor and is set to 0.1 for the reference state. The soil erodibility factor $K_i$ (Mg MJ⁻
¹ mm⁻¹) is calculated using the method of the EPIC model (Sharpley and Williams, 1990) based
on SOC and soil texture data obtained from the GSDE database (Table 1). The term $P_{ref}$ (0-1,
dimensionless) in Eq. 1 is a factor representing erosion control practices. It was set to 1, as we
did not consider the impacts of soil conservation practices in reducing soil erosion rate. Note that
it does not matter which value is chosen for the $R_{ref}$, $R_{30\_ref}$, $C_{ref}$ and $P_{ref}$ as long as they are used
consistently throughout a study.
For the use of these reference sediment delivery estimates in ORCHILEAK Clateral, the values
were first calculated for each headwater basin derived from high resolution geodata, then
aggregated to 0.5° grid cells – the scale used in our simulations and required to maintain
computational efficiency (also limited by the availability of climate and land cover forcing data).
This aggregated dataset is then used to force the simulation of Then, the actual daily sediment
delivery ($S_{iday}$, g day⁻¹ grid⁻¹) in ORCHIDEE Clateralis calculated, by comparing the simply





based on the estimated reference sediment delivery rates of Eq. (1) and on the ratios between
actual runoff and land cover conditions to and the assumed reference conditions used to create
that forcing file (Eq. 4).
$$S_{ref} = \sum_{i=1}^{n}(S_{i\_ref}) \times 10^6 \qquad (3)$$
$$S_{iday} = S_{ref} \left(\frac{R_{iday}\ R_{30\_iday}}{R_{ref}\ R_{30\_ref}}\right)^b \frac{C_{iday}}{C_{ref}} \qquad (4)$$
where $R_{iday}$ (mm day$^{-1}$) is the daily total surface runoff simulated by the hydrological module or
ORCHIDEE-MUSLE at 0.5° spatial resolution every 30 minutes. $R_{30\_k}$ (mm 30-min$^{-1}$) is the
maximum value of the 48 half-hour runoffs in each day. $C_{iday}$ (0-1, unitless) is the daily actual
cover management factor, calculated based on the fraction of surface vegetation cover, the
amount of litter carbon and the biomass of living roots in each PFT within each 0.5°×0.5° grid
cell. $R_{ref}$, $R_{30\_ref}$, $C_{ref}$ and $P_{ref}$ are the reference values used to estimate the reference sediment
delivery rates as describe above.
Daily POC delivery to river headstream in each 0.5° grid cell is finally simulated based on the
sediment delivery rate and the average SOC concentration of surface soil layers (0-20 cm). The
vertical SOC profile is updated every day based on the average depth of eroded soil for each PFT
in each 0.5° grid cell of ORCHIDEE. For more detailed description of the ORCHIDEE-MUSLE,
we refer to Zhang et al. (2020).
**2.2 Sediment and POC transport in inland water network**
Through the merge of the model branches ORCHILEAK and ORCHIDEE-MUSLE, the new
branch ORCHIDEE-C$_{lateral}$ combines the novel features of both sources (DOC and POC)
described above. The development of ORCHIDEE-C$_{lateral}$ is complemented by a representation of
the sediment and POC transport through the river network that is completely novel and described
below.
**2.2.1 Sediment transport**



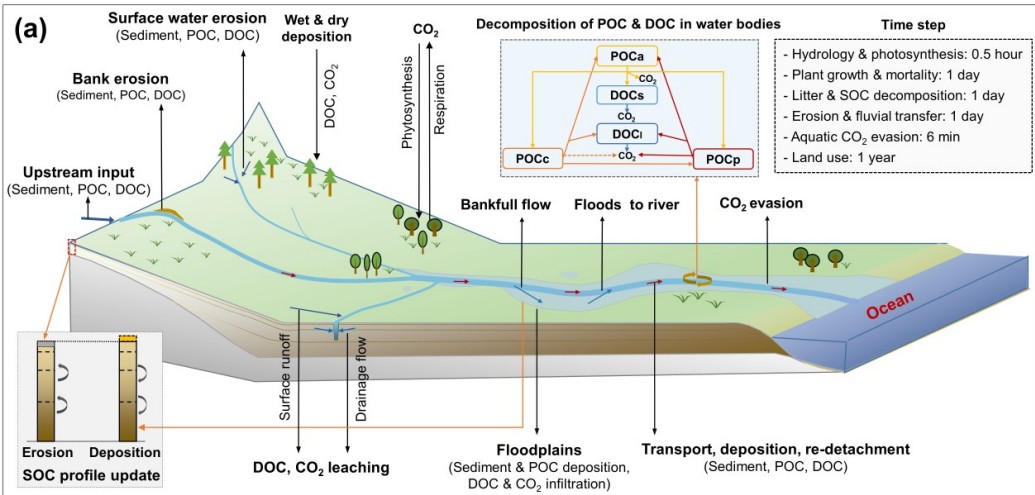

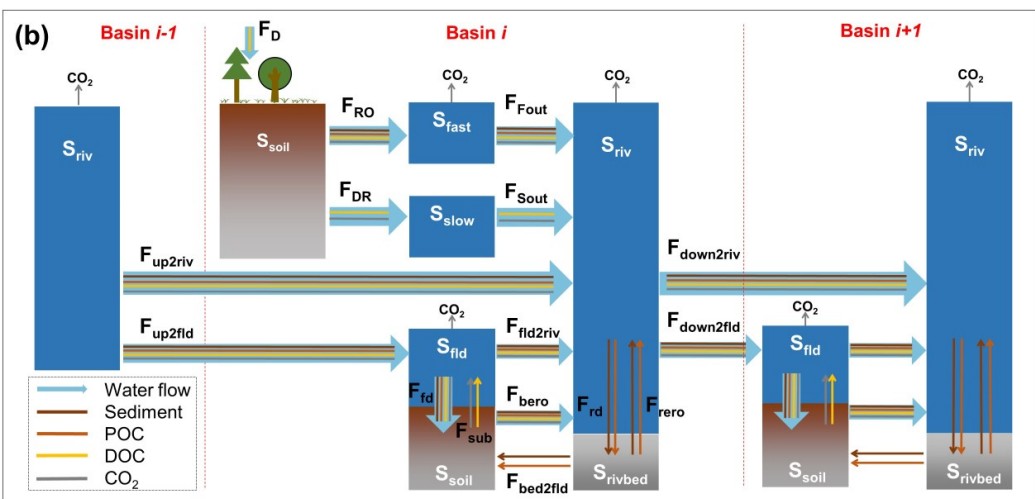


**Figure 1** Simulated lateral transfer processes of water, sediment and carbon (POC, DOC and $CO_2$) in ORCHIDEE-$C_{lateral}$ (a) and a schematic plot for the reservoirs and flows of water, sediment and carbon represented in the routing module of ORCHIDEE-$C_{lateral}$. $S_{soil}$ is the soil pool. $S_{rivbed}$ is the sediment (also POC) deposited in river bed. $S_{fast}$, $S_{slow}$, $S_{river}$ and $S_{flood}$ are the 'fast', 'slow', stream and flooding water reservoir, respectively. $F_{RO}$ and $F_{DR}$ are the surface runoff and belowground drainage, respectively. $F_{Fout}$ and $F_{Sout}$ are the flows from fast and slow reservoir to the stream reservoir, respectively. $F_{up2riv}$ and $F_{up2fld}$ are the inputs from upstream basins to the stream reservoir and flooding reservoir of the target basin, respectively. $F_{dow2riv}$ and





$F_{down2fld}$ are the outputs from the stream reservoir of the target basin to the stream reservoir and
flooding reservoir of the neighbouring downstream basin, respectively. $F_{fld2riv}$ is the return flow
from flooding reservoir to stream reservoir. $F_{bed2fld}$ is the transform from deposited sediment in
river bed to floodplain soil. $F_{bero}$ is bank erosion. $F_{rd}$ and $F_{rero}$ are the deposition and re-
detachment of sediment and POC in river channel, respectively. $F_{sub}$ is the flux of DOC and CO2
from floodplain soil (originated from the decomposition of submerged litter and soil carbon) to
the overlying flooding water. $F_{fd}$ is the deposition of sediment and POC and the infiltration of
water and DOC. $F_D$ is the wet and dry deposition of DOC from atmosphere and plant canopy.

Simulation of sediment transport through the river network basically follows the routing scheme
of surface water and DOC of ORCHILEAK (Fig. 1). Along with surface runoff ($F_{RO\_h2o}$, m$^3$ day$^{-1}$
), the sediment delivery ($F_{RO\_sed}$, g day$^{-1}$) from uplands in each basin (i.e. each 0.5° grid in the
case of this study) initially feeds an aboveground water reservoir with a so-called fast water
residence time ($S_{fast\_h2o}$, m$^3$). From this fast water reservoir, a delayed outflow feeds into the so-
called stream reservoir ($S_{riv}$, m$^3$, Fig. 1b). Daily water ($F_{Fout\_h2o}$, m$^3$ day$^{-1}$) and sediment ($F_{Fout\_sed}$,
g day$^{-1}$) flows from fast water reservoir to stream reservoir are calculated from a basin-specific
topographic index $f_{topo}$ (unitless, ) extracted from a forcing file (Table 1) and a reservoir-specific
factor $\tau$ which translates $f_{topo}$ into a water residence time of each reservoir (Eqs. 5, 6). Following
Guimberteau et al. (2012), the $\tau$ of the fast water reservoir ($\tau_{fast}$) is set to 3.0 days. As the
sediment delivery calculated from MUSLE is the net soil loss from headwater basins (gross soil
erosion – soil deposition within headwater basins), we assumed that there is no sediment
deposition in the fast reservoir, and that all of the sediment in the fast reservoir enter into stream
reservoir. In addition, only the surface runoff causes soil erosion. The belowground drainage
($F_{DR\_h2o}$, m$^3$ day$^{-1}$) only transport DOC and dissolved CO$_2$ to the stream reservoir (Fig. 1b).
$$F_{Fout\_h2o} = \frac{S_{fast\_h2o}}{\tau_{fast}\, f_{topo}}$$    (5)
$$F_{Fout\_sed} = \frac{S_{fast\_sed}}{\tau_{fast}\, f_{topo}}$$    (6)
The budget of the suspended sediment in stream reservoir ($S_{riv\_sed}$, g) is determined by the
$F_{Fout\_sed}$, upstream sediment input ($F_{up2riv\_sed}$, g day$^{-1}$), the sediment input in flooding water
returning to the river ($F_{fld2riv\_sed}$, g day$^{-1}$),  re-detachment of the previously deposited sediment in
the river bed ($F_{rero\_sed}$, g day$^{-1}$), bank erosion ($F_{bero\_sed}$, g day$^{-1}$), sediment deposition in the river

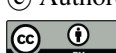



bed ($F_{rd\_sed}$, g day$^{-1}$) and sediment transported to downstream river stretches ($F_{down2riv\_sed}$, g day$^{-1}$)
and, occasionally, floodplains ($F_{down2fld\_sed}$, g day$^{-1}$) (Eq. 7).
$$\frac{dS_{riv\_sed}}{dt} = F_{Fout\_sed} + F_{up2riv\_sed} + F_{fld2riv\_sed} + F_{rero\_sed} + F_{bero\_sed} - F_{rd\_sed} - F_{down2riv\_sed} - F_{down2fld\_sed} \qquad (7)$$
Sediment transport capacity ($TC$, g m$^{-3}$), defined as the maximum load of sediment that a given
flow rate can carry, determines the amount of suspended sediment that can be transported to the
downstream grid cell (e.g. $F_{down2riv\_sed}$, $F_{down2fld\_sed}$), as well as the amount of suspended sediment
that will deposit on the river bed ($F_{rd\_sed}$) or the erosion rate of the river bed ($F_{rero\_sed}$) or river
bank ($F_{bero\_sed}$) (Arnold et al., 1995; Nearing et al., 1989; Neitsch et al., 2011). Several physics-
based algorithms have been proposed to accurately calculate the $TC$ of stream flows (Arnold et
al., 1995; Molinas and Wu, 2001; Nearing et al., 1989). These algorithms mostly require detailed
information about the stream power (e.g. flow speed and depth), geomorphic properties of the
river channel (e.g. slope and hydraulic radius) and the physical properties of the sediment
particles (e.g. median grain size) (Neitsch et al., 2011). They are good predictors to estimate $TC$
in rivers with detailed observation data on local stream, soil, geomorphic properties.
Unfortunately, it is not practical to implement those algorithms in ORCHIDEE-C$_{lateral}$ due to the
lack of appropriate forcing data at large scale as well as the relatively rough representation of
stream flow dynamics compared to hydrological models for small basins. For example, runoff
and sediment from all headwater basins in one 0.5° grid cell of ORCHIDEE-C$_{lateral}$ are assumed
to flow into one single virtual river channel. Although the total river surface area in each grid cell
is represented (obtained from forcing file (Table 1), Lauerwald et al., 2015), the length, width
and depth of the river channel are unknown. Furthermore, in reality, there can be multiple river
channels in the area represented by each grid cell, and these channels might flow to different
directions. This illustrates the difficulty to simulate the detailed hydraulic dynamics of the stream
flow in each grid.
We also noticed that previous studies have derived empirical functions of upstream drainage area
(e.g. Luo et al., 2017) or upstream runoff (e.g. Yamazaki et al., 2011) to calculate the river width
and depth, allowing to simulate the water flow in the river channel using physically-based
algorithms. Unfortunately, to obtain a good fit of the simulated river discharges against
observations, the parameters in the empirical functions for calculating river width and depth
generally need to be calibrated separately for each catchment (Luo et al., 2017), an approach that



is incompatible with large-scale simulations like those performed here. Without such calibration,
the simulated geometrical properties of the river channel and runoff are prone to large
uncertainties, thus rendering the simulation of sediment transport at continental or global scale
using physically-based algorithms a more challenging task.
In this study, we used an empirical equation adapted from the WBMsed model, which has been
proven effective in simulating the suspended sediment discharges in global large rivers (Cohen et
al., 2014), to estimate the $TC$ (g m$^{-3}$) of stream flow:
$$TC = \frac{\omega \, q_{ave}^{0.3} \, A^{0.5} \left(\frac{q_{iday}}{q_{ave}}\right)^{e_1} (24 \times 60 \times 60)}{F_{down2riv\_h2o}} \tag{8}$$
$$e_1 = 1.5 - max(0.8, 0.145 \log_{10} A) \tag{9}$$
where $\omega$ is the coefficient of proportionality, $q_{ave}$ (m$^3$ s$^{-1}$) is long-term average stream flow rate
obtained from an historical simulation by ORCHILEAK (Table 1), $q_{iday}$ (m$^3$ s$^{-1}$) is stream flow
rate on day $i$, $A$ (m$^2$) is the upstream drainage area, $F_{down2riv\_sed}$ (m$^3$ day$^{-1}$) is the daily downstream
water discharge from the stream reservoir. In the stream reservoir of each basin, net deposition
occurs when $TC$ is smaller than the concentration of suspended sediment, and the daily deposited
sediment ($F_{rd\_sed}$, g day$^{-1}$) is calculated based on the surplus of the suspended sediment:
$$F_{rd\_sed} = c_{rivdep}(S_{riv\_sed} - TC \, S_{riv\_h2o}) \tag{10}$$
where $c_{rivdep}$ (0-1, unitless) is the daily deposited fraction of the sediment surplus. Net erosion of
the previously deposited sediment in river bed ($S_{rivbed\_sed}$, Fig. 1) or the river bank occurs when
$TC$ is larger than the concentration of suspended sediment. We assumed that the erosion of river
bank occurs only after all of the $S_{rivbed\_sed}$ has been eroded. Thus the daily erosion rate ($F_{rero\_sed}$, g
day$^{-1}$) in river channel is calculated as:
$$F_{rero\_sed} = \begin{cases} c_{ebed}(TC \, S_{riv\_h2o} - S_{riv\_sed}), & c_{ebed}(TC \, S_{riv\_h2o} - S_{riv\_sed}) \le S_{rivbed\_sed} \\ S_{rivbed\_sed} + c_{ebank}(TC \, S_{riv\_h2o} - S_{riv\_sed} - S_{rivbed\_sed}), & c_{ebed}(TC \, S_{riv\_h2o} - S_{riv\_sed}) > S_{rivbed\_sed} \end{cases} \tag{11}$$
where $c_{ebed}$ (0-1, unitless) and $c_{ebank}$ (0-1, unitless) are the fraction of sediment deficit that can be
complemented by erosion of river bed and bank, respectively. After updating the $S_{riv\_sed}$ based on
the $F_{rd\_sed}$ or $F_{rero\_sed}$, the sediment discharge to downstream basin ($F_{down2riv\_sed}$, g day$^{-1}$) is
calculated based on the ratio of downstream water discharge to the total stream reservoir:
$$F_{down2riv\_sed} = (S_{riv\_sed} - F_{rd\_sed} + F_{rero\_sed}) \frac{F_{down2riv\_h2o}}{S_{riv\_sh2o}} \tag{12}$$



In each basin, the bankfull flow occurs when $S_{riv\_h2o}$ exceeds the maximum water storage of river
channel ($S_{rivmax}$, g), which is defined by a forcing file (Table 1). Sediment flow from stream to
floodplain ($F_{down2fld\_sed}$, g day$^{-1}$) follows the flooding water, and it is calculated as:
$$F_{down2fld\_sed} = \left(S_{riv\_sed} - F_{rd\_sed} + F_{rero\_sed}\right)\frac{F_{down2fld\_h2o}}{S_{riv\_sh2o}} \tag{13}$$

$$F_{down2fld\_h2o} = \left(S_{riv\_h2o} - F_{down2riv\_h2o} - S_{rivmax}\right)\frac{f_{A\_fld}}{f_{A\_fld}+f_{A\_riv}} \tag{14}$$

where $f_{A\_fld}$ (0-1, unitless) and $f_{A\_riv}$ (0-1, unitless) is the fraction of floodplain area and river
surface area in each basin, respectively. Following the routing scheme of ORCHILEAK, the
bankfull flow of a specific basin is assumed to enter the floodplain in the neighbouring
downstream basin instead of the basin where it originates.
The sediment balance in flooding reservoir ($S_{fld\_sed}$, g) is controlled by sediment input from the
upstream basins ($F_{up2fld\_sed}$, g day$^{-1}$), the sediment flowing back to the stream reservoir ($F_{fld2riv\_sed}$,
g day$^{-1}$) and the sediment deposition ($F_{fd\_sed}$, g day$^{-1}$) (Fig. 1):
$$\frac{dS_{fld\_sed}}{dt} = F_{up2fld\_sed} - F_{fld2riv\_sed} - F_{fd\_sed} \tag{15}$$

Sediment deposition in flooding water is calculated as the sum of a natural deposition and the
deposition due to evaporation ($E_{h2o}$, m$^3$ day$^{-1}$) and infiltration ($I_{h2o}$, m$^3$ day$^{-1}$) of the flooding
waters:
$$F_{fd\_sed} = c_{flddep}\, S_{fld\_sed} - S_{fld\_sed}\frac{E_{h2o}+ I_{h2o}}{S_{fld\_h2o}} \tag{16}$$

where $c_{flddep}$ (0-1, unitless) is the daily deposited fraction of the suspended sediment in flooding
waters. After removing the deposited sediment from $S_{fld\_sed}$, $F_{fld2riv\_sed}$ is calculated based on the
ratio of ratio of $F_{fld2riv\_h2o}$ to the total flooding reservoir:
$$F_{fld2riv\_sed} = S_{fld\_sed}\frac{F_{fld2riv\_h2o}}{S_{fld\_h2o}- E_{h2o}- I_{h2o}} \tag{17}$$


$$F_{fld2riv\_h2o} = \frac{S_{fld\_h2o}- E_{h2o}- I_{h2o}}{\tau_{flood}\, f_{topo}} \tag{18}$$

where $\tau_{flood}$ is a factor which translates $f_{topo}$ into a water residence time of the flooding reservoir.
Same to ORCHILEAK, it is set to 1.4 (day m$^{-2}$) in this study.
Note that as the upland soil in ORCHIDEE is composed of clay, silt and sand particles, so that
the dynamics of clay-, silt- and sand-sediment in inland waters are simulated separately. To
represent the selective transport of clay-, silt- and sand-sediment, the model parameter $\omega$ (Eq. 8)





and $c_{rivdep}$ (Eq. 10) are set to different values when calculating the sediment transport capacity
and the deposition of surplus suspended sediment for different particle sizes (Table A1).
**2.2.2 POC transport and decomposition**
Many studies described the selective transport of POC and sediment of different particles sizes.
The enrichment ratio (defined as the ratios of fraction of any given component in the transported
sediment to that in the eroded soils) of POC in the transported sediment generally showed
significant positive correlation to the fine sediment particles (e.g. fine silt and clay), but negative
correlation to the coarse sediment particles (Galy et al., 2008; Haregeweyn et al., 2008; Nadeu et
al., 2011; Nie et al., 2015). In ORCHIDEE-C$_{lateral}$, the physical movements of POC in inland
water systems are simply assumed to follow the flows of finest clay-sediment (Fig. 1b). For
example, the fractions of riverine suspended POC which is deposited on the river bed ($F_{rd\_POC}$, g
C day$^{-1}$) or is transported to the river channel ($F_{down2riv\_POC}$, g C day$^{-1}$) or floodplain
($F_{down2fld\_POC}$, g C day$^{-1}$) of the downstream grid cell are assumed to be equal to the
corresponding fractions of clay-sediment (Eqs. 19-21). Also flows of suspended POC in flooding
waters to floodplain soil ($F_{fld\_POC}$, g C day$^{-1}$) or back to the stream reservoir ($F_{fld2riv\_POC}$, g C day$^{-1}$
$^{-1}$), as well as the resuspension of POC from the river bed ($F_{rero\_POC}$, g C day$^{-1}$) are scaled to the
simulated flows of clay-sediment (Eqs. 22-24). Note that, similar to SOC, the POC in aquatic
reservoirs are divided into three pools: the active ($POC_a$), slow ($POC_s$) and passive pool ($POC_p$)
(Fig. 1a). The eroded active, slow and passive SOC flow into the corresponding POC pools in
the 'fast' water reservoir (Fig. 1b).
$$F_{rd\_POC} = S_{riv\_POC} \frac{F_{rd\_sed\_clay}}{S_{riv\_sed\_clay}} \tag{19}$$

$$F_{down2riv\_POC} = S_{riv\_POC} \frac{F_{down2riv\_sed\_clay}}{S_{riv\_sed\_clay}} \tag{20}$$

$$F_{down2fld\_POC} = S_{riv\_POC} \frac{F_{down2fld\_sed\_clay}}{S_{riv\_sed\_clay}} \tag{21}$$

$$F_{fld\_POC} = S_{fld\_POC} \frac{F_{fld\_sed\_clay}}{S_{fld\_sed\_clay}} \tag{22}$$

$$F_{fld2riv\_POC} = S_{fld\_POC} \frac{F_{fld2riv\_sed\_clay}}{S_{fld\_sed\_clay}} \tag{23}$$

$$F_{bed2fld\_POC} = S_{rivbed\_POC} \frac{F_{bed2fld\_sed}}{S_{rivbed\_sed}} \tag{24}$$



The representation of POC dynamics in the aquatic reservoirs and bed sediment involve as well
decomposition, which follows largely the scheme used for SOC (Fig. 1a). However, instead of
using the rate modifiers for soil temperature and moisture used in the soil carbon module, daily
decomposition rates ($F_{POC\_i}$, g C day$^{-1}$) of each POC pool ($S_{POC\_i}$, g C) are simulated to vary with
water temperature based on the Arrhenius term which is used to simulate the DOC
decomposition in ORCHILEAK (Hanson et al., 2011; Lauerwald et al., 2017):

$$F_{POC\_i} = S_{POC\_i} \frac{1.073^{(T_{water} - 28.0)}}{\tau_{poc\_i}} \qquad (25)$$

where $T_{water}$ (°C) is the temperature of water reservoirs. For the POC stored in bed sediment,
temperature of the stream reservoir is used to calculate the decomposition rate. $\tau_{POC\_i}$ is the
turnover time of the $i$ (active, slow and passive) POC pool. We assumed that the base turnover
times of active (0.3 year) and slow (1.12 years) POC pools are the same as for the corresponding
SOC pools. The passive SOC pool is generally regarded as the SOC which is associated to soil
minerals or enclosed in soil aggregates (Parton et al., 1987). During the soil erosion and sediment
transport processes, the aggregates break down and the passive POC loses its physical protection
from decomposition (Chaplot et al., 2005; Hu and Kuhn, 2016; Polyakov and Lal, 2008; Wang et
al., 2014a). To represent the acceleration of passive POC decomposition due to aggregate
breakdown, we assume that the turnover time of the passive POC is same to the active POC (0.3
year), rather than the passive SOC (462 years). Similar to the scheme used to simulate SOC
decomposition in ORCHILEAK, the decomposed POC from each of the active, slow and passive
pool flows to other POC pools, to DOC pools or is released to the atmosphere as $CO_2$ (Fig. 1).
Fractions of the decomposed POC flowing to different POC and DOC pools or to the atmosphere
are set to the same values used in ORCHILEAK for simulating the fates of the decomposed SOC
pools.
Changes in the vertical SOC profile of floodplain soils following sediment deposition is
simulated at the end of every daily modelling time-step, after physical transfers and
decomposition of POC have been calculated. The sediment deposited on the floodplain becomes
the new surface soil layer, and the active, slow and passive POC flow into the active, slow and
passive SOC pools in surface soil layer, respectively. SOC in the original surface and subsurface
soil layers is transferred sequentially to the adjacent deeper soil layers. As the vertical soil profile
in ORCHILEAK is described by an 11-layer discretization of a 2 m soil column, we introduce a
deep (> 2 m) soil pool ($S_{deep}$) to represent the soil and carbon transferred down from the 11$^{th}$ soil





layer following ongoing floodplain deposition. Decomposition rates of the organic carbon in this
deep soil pool are assumed to be same to those in the 11[th] (deepest) soil layer.  Note that when
the soil erosion rate of the floodplain soil is larger than the sediment deposition rate, sediment
and organic carbon in $S_{deep}$ move up to replenish the stocks of the 11[th] soil layer.

**2.3 Model application and evaluation**

In this study, the ORCHIDEE-C$_{lateral}$ was applied over Europe (-30W– 70E, 34N-75N, also
includes a part of Middle East and Africa, Fig. S1 in the Supplement), where extensive
observation datasets are available to calibrate and evaluate our model (Table 1). The return
period of daily bankfull flow ($P_{flooding}$, year), which represents the average interval between two
flooding days and is used in this study to produce the forcing file of $S_{rivmax}$ from a pre-run of
ORCHILEAK. Note that $P_{flooding}$ is generally shorter than the return period of real flooding
events, as the flooding may occur in several continuous days and the all flooding waters
occurring on these continuous days are generally regarded to belong to the same flooding event
(Fig. S1). $P_{flooding}$ shows substantial spatial variations following climate and topography
(Schneider *et al.*, 2011). In this study, we assumed that $P_{flooding}$ for all rivers in Europe are the
same and the observed long-term (1961–2000) average bank full flow rate (m$^3$ s$^{-1}$) at 66 sites
obtained from Schneider *et al.* (2011) was used to calibrate $P_{flooding}$ (= 0.1 year, Table A1). Same
to Zhang et al. (2020), the parameters *a*, *b*, *c* and *d* in Eq. 1 and 2 (Table A1) were calibrated at
57 European catchments (Fig. S2d) against the modelled sediment delivery data obtained from
the European Soil Data Centre (ESDAC, Borrelli et al., 2018). The sediment delivery data from
the ESDAC product is simulated by the WaTEM/SEDEM model using high-resolution data of
topography, soil erodibility, land cover and rainfall. It has been calibrated and validated using
observed sediment fluxes from 24 European catchments (Borrelli et al., 2018).
Parameters controlling sediment transport, deposition and re-detachment (i.e. $\omega$, $c_{rivdep}$, $c_{flddep}$,
$c_{ebed}$ and $c_{ebank}$, Table S1) in stream and flooding reservoirs were calibrated against the observed
long-term averaged sediment discharge rate (Table 1). We also conducted a sensitivity analysis
to test the sensitivity of the simulated riverine sediment and carbon discharges to these
parameters, following the method used in Tian et al. (2015). The sensitivity of simulation results
was evaluated based on the relative changes in simulated riverine sediment and carbon
discharges to a 10% increase and decrease of each parameter (Table S1). Result of the sensitivity



analysis shows that the simulated riverine sediment and POC discharges are most sensitive to
$c_{rivdep}$ in Eq. 5, followed by $\omega$ in Eq. 8 (Fig. S3). Compared to $c_{rivdep}$ and $\omega$, the simulated riverine
sediment and POC discharges are less sensitive to $c_{flddep}$, $c_{ebed}$ and $c_{ebank}$. With 10% changes in
$c_{flddep}$, $c_{ebed}$ or $c_{ebank}$, the changes in riverine sediment and POC discharges are generally less than
3%. In addition, the changes in simulated riverine DOC and $CO_2$ discharges are mostly less than
1% with 10% changes in $\omega$, $c_{flddep}$, $c_{ebed}$ and $c_{ebank}$. Nonetheless, a 10% change in $c_{rivdep}$ can lead
to a change of about 5% in the simulated riverine $CO_2$ discharge (Fig. S3).
After parameter calibration, ORCHIDEE-C$_{lateral}$ was applied to simulate the lateral transfers of
water, sediment and organic carbon in European rivers over the period 1901-2014. Before this
historical simulation, ORCHIDEE-C$_{lateral}$ was run over 10,000 years (spin-up) until the soil
carbon pools reached a steady state. In the 'spin-up' simulation, the PFT maps, atmospheric $CO_2$
concentrations and meteorological data during 1901–1910 were used repeatedly as the forcing
data.  The finally simulated water discharge rates in European rivers were evaluated using
observation data at 93 gauging sites (Fig. S2a) from the Global Runoff Data Base (GRDC, Table
1). The simulated bankfull flows were evaluated against observed long-term (1961–2000)
average bankfull flows at 66 sites (Fig. S2b) from Schneider *et al*. (2011). The simulated riverine
sediment discharge rate is evaluated using observation data from the European Environment
Agency and existing publications (see Table 1) at 221 gauging sites (Fig. S2c). The riverine total
organic carbon (TOC), POC and DOC concentrations provided by the GLObal RIver Chemistry
Database (GLORICH, Hartmann et al., 2019) at 346 sites (Fig. S2d) were used to evaluate the
simulated riverine POC and DOC concentrations. Note that observations in the GLORICH
database which are measured at gauging sites with drainage area $<1.0\times10^4$ km$^2$ were excluded
from our model evaluation, because these small catchments cannot be represented by the coarse
river network scheme at 0.5 degree (ca. 55 km at the equator). Among the retained 346 gauging
sites, TOC concentrations were measured at 188 sites, DOC was measured at 314 sites. POC was
measured at only 3 sites in the Rhine catchment.
**3 Results and Discussion**
**3.1 Model evaluation**
**3.1.1 Stream water discharge and bankfull flow**



Evaluation of our simulation results using *in situ* observation data from Europe rivers indicates
that ORCHIDEE-C$_{lateral}$ well reproduces the magnitude and interannual variation of water
discharge rates in major European rivers (Figs. 2a and S4). Overall, the simulated riverine water
discharge rate explained 94% (Fig. 2a) of the spatial variation of the observed long-term average
water discharge rates across 93 gauging sites in Europe (Fig. S2a). Relative biases (calculated as:
$\frac{simulation-observation}{observation} \times 100\%$, as used through the manuscript if not otherwise stated) of the
simulated average water discharge rates compared to the observations are mostly smaller than
30% (Fig. 2a). For major European rivers, such as the Rhine, Danube, Elbe, Rhone and Volga,
ORCHIDEE-C$_{lateral}$ also captures the interannual variation of the water discharge rate (Fig. S4).
We recognize that ORCHIDEE-C$_{lateral}$ may overestimate or underestimate the water discharge
rate in some rivers (Fig. 2a), particularly in smaller rivers where discrepancy between the stream
routing scheme (delineation of catchment boundaries) extracted from the forcing data at 0.5°
resolution and the real river network (Fig. S5) can be substantial. An over- or underestimation of
the catchment area by the forcing data will introduce a proportional bias to the average amount
of simulated discharge from that catchment. Another problem are stream channel bifurcations
which occur in reality, but which are not represented in a stream network derived from a digital
elevation model. For example, in the Danube river delta, a fraction of the discharge is actually
exported to the sea through the Saint George Branch, in addition to the water discharge through
the main river channel (Fig. S5b). This explains why the simulated water discharge rate at the
outlet of Danube catchment is larger than the observation at the Ceatal, Romania (identify
number in the GRDC database is 6742900, Fig. S4m), where only the main stream discharge was
measured.

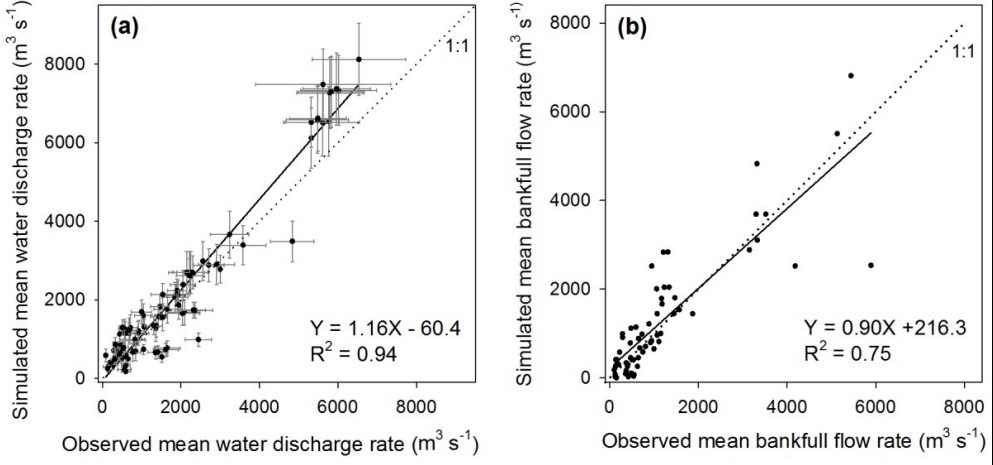


**Figure 2** Comparison between observed and simulated riverine water discharge rates (a) and
bankfull flow rates (b). In figure (a), the error bar denotes the standard deviation of interannual
variation. Sources of the observed riverine water discharge rate and bankfull flow rate can be
found in Table 1.

548

By setting the return period of the daily flooding rate to 0.1 year, the simulated bankfull flow
rates compare well to observations at the 66 sites for which data was available (Fig. 2b). Overall,
the simulation result explained 75% of the inter-site variation of the observed bankfull flow
rates. Relative biases of the simulated bankfull flow rates are generally lower than 30%, although
the relative bias may be larger than 100% at some sites.

**3.1.2 Sediment transport**

The simulated area-averaged sediment delivery rates from upland to river network by the
ORCHIDEE-C$_{lateral}$ are overall comparable to those simulated by the WaTEM/SEDEM for most
catchments in Europe (Figs. 3a and S2d). In the two catchments in the Apennine Peninsula,
ORCHIDEE-C$_{lateral}$ gives a drastically lower estimation on the sediment delivery rates compared
to WaTEM/SEDEM. By excluding these two catchments, ORCHIDEE-C$_{lateral}$ reproduces 72% of
the spatial variation of the sediment delivery rates estimated by the WaTEM/SEDEM (Fig. 3a).

ORCHIDEE-C$_{lateral}$ reproduces 83% of the inter-site variation of the sediment discharge rates
across Europe (Fig. 3b). Simulation of the riverine sediment discharge rate at large spatial scale





is still a big challenge. It generally needs detailed information on the stream flow, geomorphic
properties of river channel and the particle composition of the suspended sediment (Neitsch et
al., 2011). Moreover, the parameters of existing sediment transport models usually require
recalibration when they are applied to different catchments (Gassman et al., 2014; Oeurng et al.,
2011; Vigiak et al., 2017). In ORCHIDEE-$C_{lateral}$, the sediment processes in river networks are
simulated using simple empirical functions and parameters based on a routing scheme at a spatial
resolution of 0.5° (section 2.2.1). Detailed information about the stream flow (e.g. cross-
sectional area) and the geomorphic properties of river channels are not represented. Sediment
discharge in all catchments was simulated using a universal parameter set. This may explain why
ORCHIDEE-$C_{lateral}$ fails to capture the sediment discharge rates in some specific catchments,
especially those with relatively small drainage areas (e.g. $< 5 \times 10^3$ km$^2$).

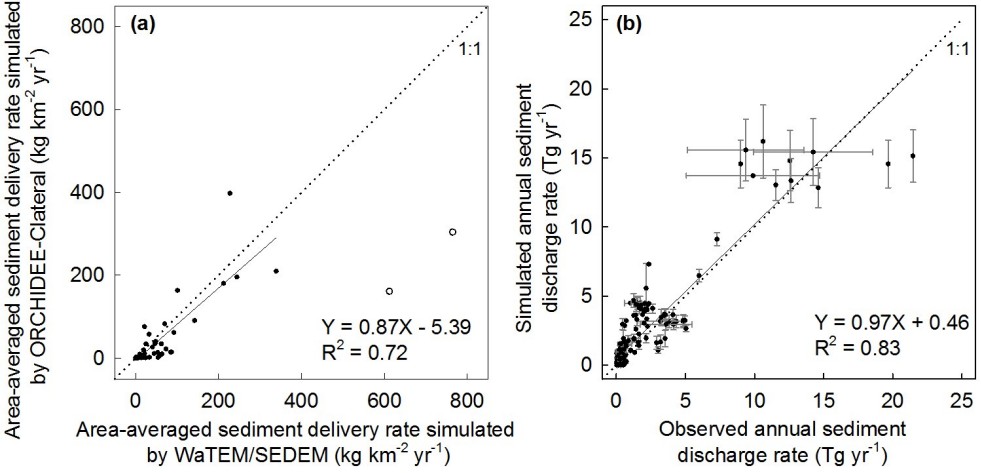


**Figure 3** Comparison between the simulated area-averaged sediment delivery rate from uplands
to river network from ORCHIDEE-$C_{lateral}$ and WaTEM/SEDEM (a), and the comparison between
observed and simulated annual sediment discharge rates at 221 gauging sites (b). In figure (a),
the two hollow dots represent the sediment delivery rates at the two catchments in the Apennine
Peninsula (Fig. S1d). The regression function in figure (a) was obtained based on the values of
all solid dots, excluding the two hollow dots. In figure (b), the error bar denotes the standard
deviation of interannual variation. Sources of the observed annual sediment discharge rate in
Table 1.


### 3.1.3 Organic carbon transport

Simulation of the riverine carbon discharge rate at large spatial scale is even a bigger challenge than simulating sediment discharge, as the riverine carbon discharge is controlled by many factors, such as upland topsoil SOC concentrations, soil erosion rate, transport and deposition rate of clay fraction in river channel and on floodplain, and the decomposition of POC in transit and in aquatic sediments. As described above, the simulated water discharge rate, bankfull flow and sediment discharge rate are overall comparable to observation (Figs. 2 and 3). The simulated total SOC stock in the top 0-30 cm soil layer in Europe of 107 Pg C is close to the value extracted from the HWSD database (106 Pg C), but significantly lower than the values extracted from some other databases, such as the GSDE (249 Pg C), SoilGrids (202 Pg C), S2017 (148 Pg C) and landGIS (226 Pg C) (Fig. S6a). Distribution of the simulated SOC stock along the latitude gradients (30° N – 75° N) are overall comparable to those extracted from the HWSD and S2017 databases (Fig. S6). But even compared to these two databases, our model still underestimated the SOC stock in southern Europe (30° N – 41° N).

Comparison of the simulated concentrations of riverine organic carbon and the observations obtained from the GLORICH database (Hartmann et al., 2019) indicates that our model can basically capture the TOC and DOC concentrations in European rivers (Figs 4, 5, S7 and S8). The simulation results explain 34% and 32% of the inter-site variation of the observed TOC and DOC concentrations, respectively (Fig. 4). For major European rivers, such as the Rhine, Elbe, Danube, Spree and Weser, the simulated long-term average TOC and DOC concentrations are overall close to the observations (Fig. 5, S7 and S8). But for the Rhone river in southern France, the DOC concentrations have been systematically overestimated by more than 50% (Fig. 5 and S8m). In addition, both simulated and observed TOC and DOC concentrations show drastic temporal (both seasonal and interannual) variations (Figs 4, S7 and S8). Our model seems to have overestimated the temporal variation of TOC and especially DOC concentrations (Figs S7 and S8).



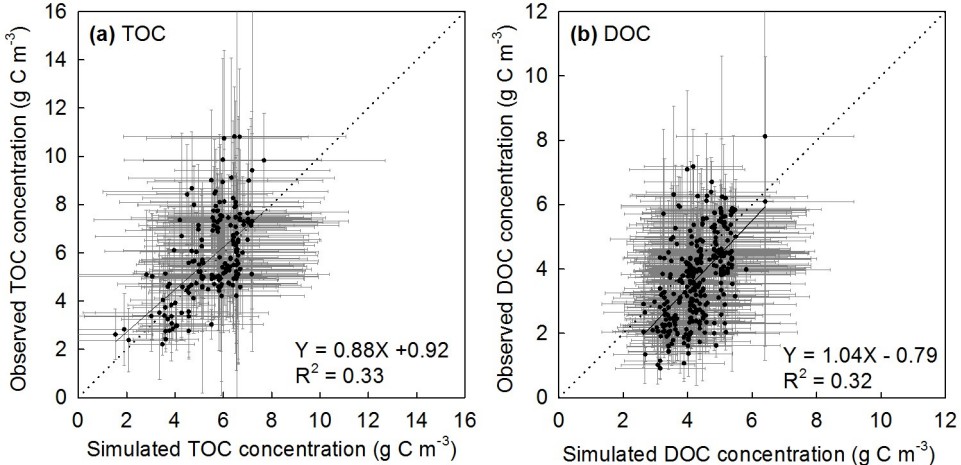

610

**Figure 4** Comparison between the observed and simulated riverine TOC (a, POC+DOC) and DOC (b) concentrations. The dot and error bar denote the mean and standard deviation at each gauging site, respectively. Not that the mean and standard deviation of the simulated concentrations at each site are calculated based on the monthly average value, but the mean and standard deviation of the observed concentrations are based on instantaneous observation.

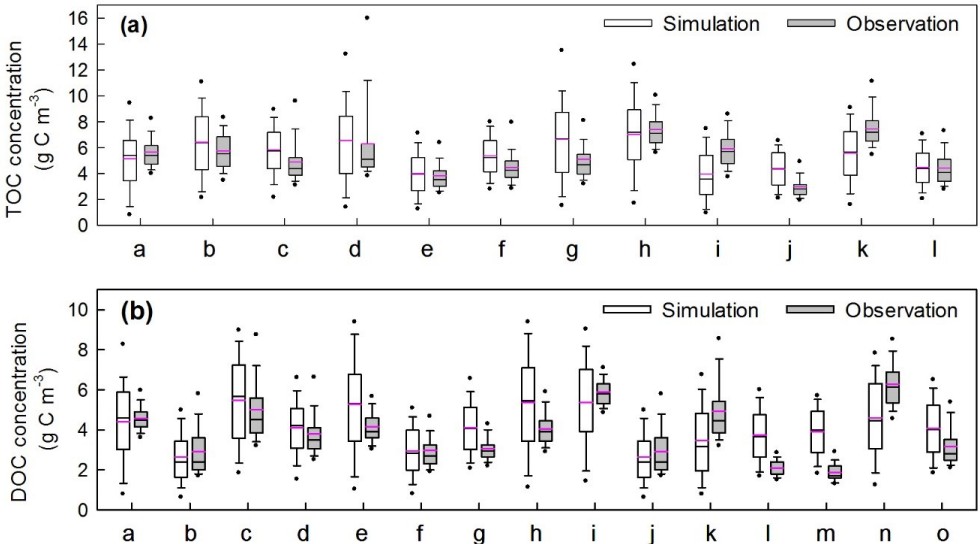


**Figure 5** Comparison between the observed and simulated concentrations of total organic carbon (TOC, a) and dissolved organic carbon (DOC, b) in river flows. The black and pink lines in each box denote the median and mean value, respectively. Box boundaries show the 25[th] and 75[th]



percentiles, whiskers denote the 10th and 90th percentiles, the dots below and above each box
denote the 5th and 95th percentiles, respectively. The specific gauging station represented by a-o
in figure (a) and (b) can be found in the corresponding sub-plot in Figure S7 and S8,
respectively.

In Europe, the GLORICH database only provides POC concentrations measured at three gauging
stations in northwestern Germany (Figs. 6, S2d). The simulated POC concentrations in the Ems
river at Rheine are overall comparable to the observation (Fig. 6e, f). However, at the two
gauging sites at the river Rhine, the POC concentrations have been significantly underestimated
(Figs. 6a-d). We noticed that the stream routing scheme of Rhine catchment at 0.5° obtained
from the forcing data STN-30p (Vörösmarty et al., 2000) differs significantly from the stream
routing scheme extracted based on high resolution (3″) DEM. Thus, besides the errors in
simulated SOC stocks, soil erosion rate, stream discharge rate, and sediment transport and
deposition rate, the inaccurate stream routing scheme used in this study might also be an
important reason for the underestimation of POC concentration in Rhine river.

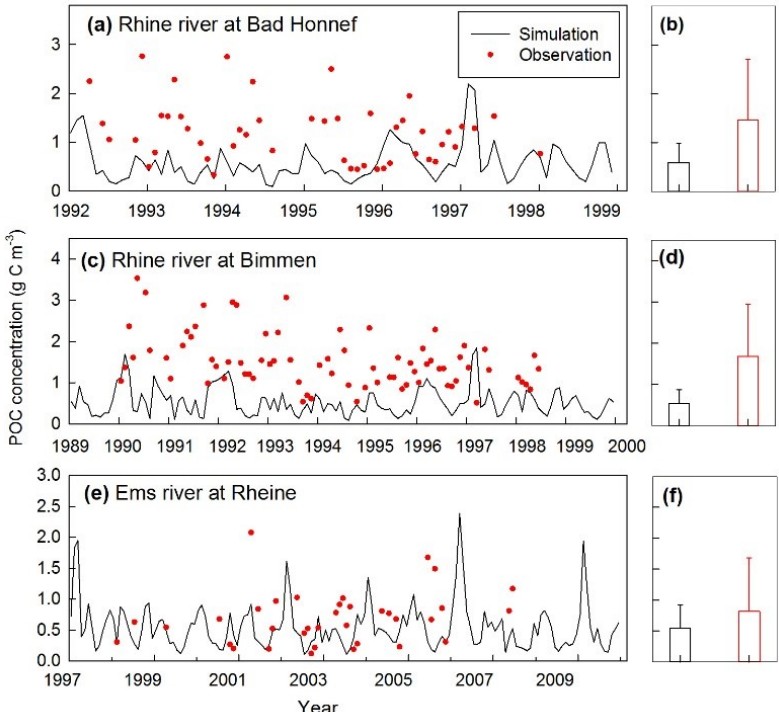


**Figure 6** Comparison between the observed (instantaneous measurement) and simulated
(monthly average value) riverine POC concentrations at three gauging sites. In figure (b), (d) and
(f), the histogram and error bar denote the mean and standard deviation of POC concentrations,
respectively.

640

**3.2 Lateral carbon transfers in Europe**

Based on our simulation results, the average annual sediment delivery from upland to the river
network caused by water erosion in Europe (-30W– 70E, 34N-75N) during 1901-2014 is 2.8±0.4
Pg yr$^{-1}$ (Fig. 7a). From Northern to Southern Europe, the sediment delivery rate from upland to
river increase from less than 1.0 g m$^{-2}$ yr$^{-1}$ in the Scandinavia Peninsula, which is covered by
mature boreal forests (Fig. S9a), and in the Northern European Plain to more than 600 g m$^{-2}$ yr$^{-1}$
in the mountainous regions of the Apennine Peninsula, Balkan Peninsula and the Middle East
(Figs. 8a, S10a). The Caucasus is mainly covered by ice and bare rock (Fig. S9), thus the
sediment delivery rate in this region is also very low. In total across Europe, 55.2% (1.8±0.2 Pg



yr$^{-1}$) of the sediment delivered into river network is deposited in river channels and floodplains,
and the remaining 36.8% (1.0±0.1 Pg yr$^{-1}$) is exported to the sea (Fig. 7a). Generally, large
rivers, like Danube, Volga, and Ob rivers, carry more sediment to the sea than small rivers (Figs.
8b, c). But several relatively small rivers in the Middle East and the Po river in northern Italy
also carry similarly large amount of sediment to the sea, as the upland soil erosion rates are very
high (> 200 g m$^{-2}$ yr$^{-1}$) in these catchments (Figs. 8a, c). Spatial distribution of the sediment
deposition is controlled by the stream routing scheme and the spatial distribution of floodplains
(Fig. 9b). In Northern and Central Europe, the area-averaged sediment deposition rates (i.e.
amount of annual sediment deposition /area of 0.5°×0.5° grid cell) in river channels and
floodplains are mostly less than 100.0 g m$^{-2}$ yr$^{-1}$ (Fig. 8d). In the downstream part of the Danube,
Po and several rivers in the Middle East, the sediment deposition rate can exceed 800.0 g m$^{-2}$ yr$^{-}$
$^1$. From 1901 to 1960s, the annual total sediment delivery from uplands to the whole river
network of Europe declined from about 3.0 Pg yr$^{-1}$ to about 2.3 Pg yr$^{-1}$ (Fig. S11a). From 1960 to
2014, the annual sediment delivery rate did not show a significant trend, but revealed large
interannual variations.
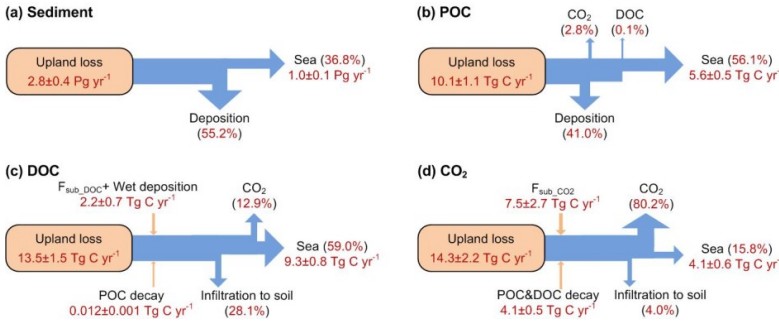
**Figure 7** Averaged annual lateral redistribution rate of sediment (a), POC (b), DOC (c) and $CO_2$
(d) in Europe for the period 1901-2014. $F_{sub\_DOC}$ and $F_{sub\_CO2}$ are the DOC and $CO_2$ inputs from
floodplain soil (originated from the decomposition of submerged litter and soil carbon) to the
overlying flooding water, respectively.

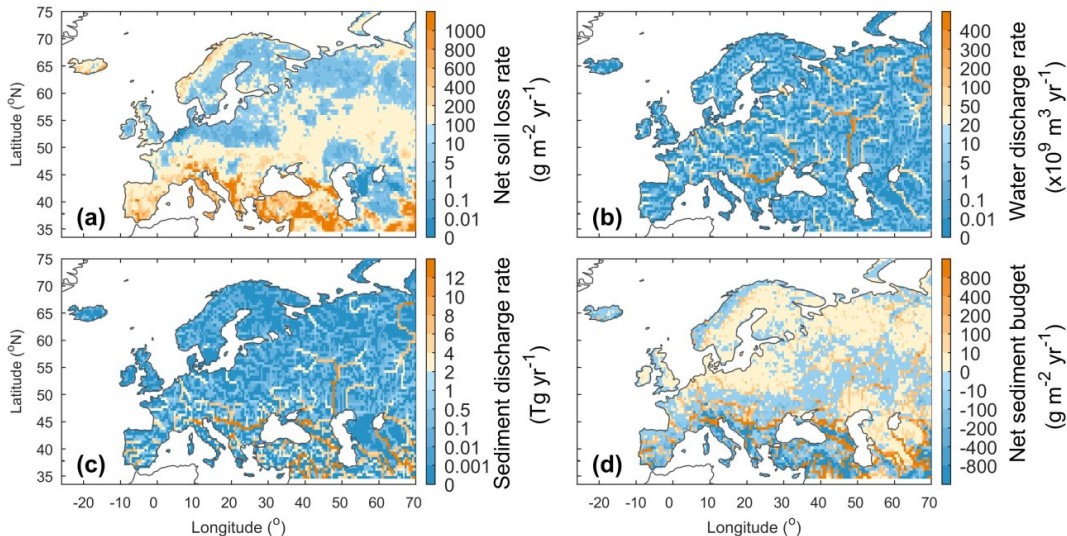

**Figure 8** Averaged annual lateral redistribution rate of water and sediment in Europe during 1901-2014. (a) Annual sediment delivery rate from upland to river network; (b) annual water discharge rate; (c) annual sediment discharge rate and (d) annual net sediment budget in each 0.5°×0.5° grid cell. In figure d, the positive and negative values denote net gain and net loss of sediment, respectively.

Along with soil erosion and sediment transport, the average annual POC delivery from upland to river network in the whole Europe during 1901-2014 is $10.1\pm1.1$ Tg C yr$^{-1}$ (Fig. 7b). 41.0% of the POC delivered into the river network is deposited in river channels and floodplains, 2.9% is decomposed during transport, and the remaining 56.1% is exported to the sea. Spatial patterns of the area-averaged SOC delivery rate and POC discharge rate basically follow that of sediment (Fig. 9a, c). But although the sediment discharge rates in some small rivers in the Middle East can be as high as that in the Danube or Volga river (Fig. 8c), the POC delivery rates in these small rivers is much smaller than in the larger ones (Fig. 9c). This is mainly due to the lower SOC stocks in the Middle East compared to those found in the Danube and Volga catchments (Fig. S6). We also note that different from the sediment delivery, the annual total POC delivery from upland to river network in Europe did not show a significant declining trend from 1901 to 1960s (Fig. S11b). The increase in SOC stock (Fig. S11c) may have partially offset the decline in sediment delivery rate.



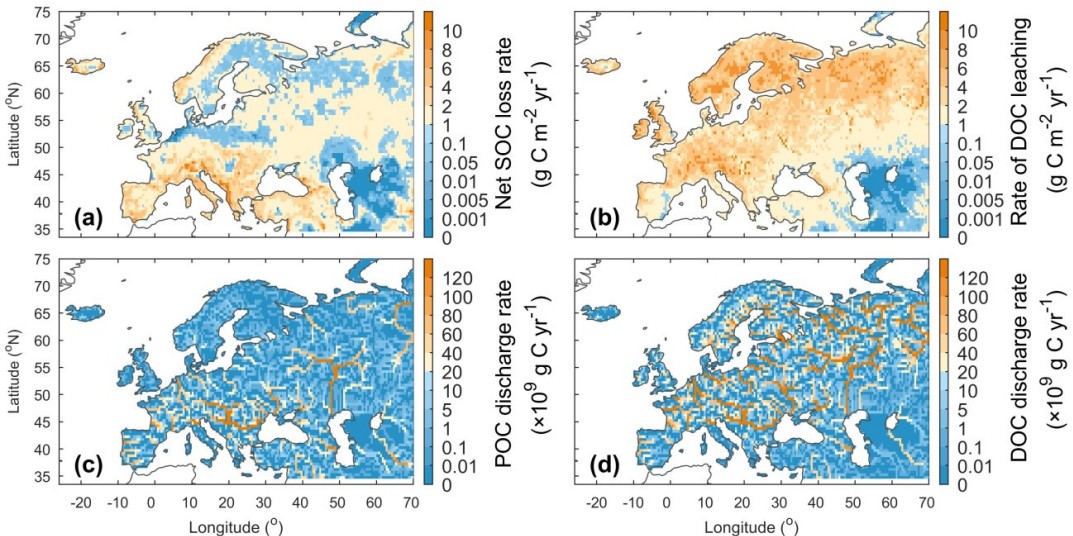

**Figure 9** Averaged annual lateral redistribution rate of organic carbon in Europe during 1901-2014. (a) Annual SOC delivery rate from upland to river network; (b) annual DOC leaching rate; (c) annual POC discharge rate and (d) annual DOC discharge rate.

Leaching results in an average annual DOC input of 13.5±1.5 Tg C yr$^{-1}$ from soil to the river network in Europe, and the *in-situ* DOC production caused by wet deposition and the decomposition of riverine POC and submerged litter and soil organic carbon under flooding waters amounts to 2.2±0.7 Tg C yr$^{-1}$ (Fig. 7c). 28.1% of the total riverine DOC is then infiltrating into the floodplain soils, 12.9% is decomposed during riverine transport, and the remaining 59.0% is exported to the sea. The spatial distribution of the DOC leaching rate is very different from that of POC (Fig. 9b). From North-western Europe to Southeast Europe and the Middle East, the DOC leaching rates decrease from over 6 g C m$^{-2}$ yr$^{-1}$ to less than 1.0 g C m$^{-2}$ yr$^{-1}$. DOC discharge rates in major European rivers, such as Rhine, Danube, Volga, Elbe and Ob, are mostly higher than 100 Tg C yr$^{-1}$ (Fig. 9d). Comparatively, the DOC discharge rates in Southern Europe and the Middle East are significantly lower (<60 Tg C yr$^{-1}$).

The average annual leaching rate of $CO_2$ sourced from the decomposition of upland litter and soil organic carbon (incl. DOC) in the whole Europe is 14.3±2.2 Tg C yr$^{-1}$ (Fig. 7a). Decomposition of the submerged litter and organic carbon in floodplains and the decomposition



of riverine POC and DOC add an an *in-situ* $CO_2$ production amounting to 7.5±2.7 Tg C yr[-1] and
4.1±0.5 Tg C yr[-1], respectively. Most of this $CO_2$ (80.2%) feeding stream waters is then released
back to the atmosphere quickly, in such a way that only 15.8% of the $CO_2$ is exported to the sea,
and 4.0% is infiltrated into the floodplain soils.
**3.3 Implications for the terrestrial C budget of Europe**
Representing the lateral carbon transport in LSM is helpful to estimate the terrestrial carbon
cycle more accurately. From the year 1901 to 2014, soil erosion and leaching combined resulted
in a 5.4 Pg loss of terrestrial carbon to the European river network, this amount corresponding to
about 5% of the total SOC stock (106 Pg C, Fig. S6a) in the 0-30 cm soil layer. The average
annual total delivery of organic carbon (POC+DOC) during the same period is 47.3±6.6 Tg C yr[-
1] (Fig. 7), which is about 4.7% of the net ecosystem exchange (NEE (993±255 Tg C yr[-1]),
defined as the difference between the vegetation primary production (NPP) and the soil
heterotrophic respiration (Rh) due to the decomposition of litter and soil organic matter (i.e.
NEE=NPP–Rh)), and 19.2% of the net biome production (NBP (243±189 Tg C yr[-1]), defined as
the difference between NEP and the land carbon loss (Rd) due to the additional disturbances (e.g.
harvest, land cover change, and soil erosion and leaching, i.e. NBP=NEP–Rd–DOC and POC to
river) (Fig. 10b). The annual total export of carbon to the sea surrounding Europe is 19.0±1.4 Tg
C yr[-1], which amounts to 1.9% and 8.7% of the NEE and NBP, respectively.

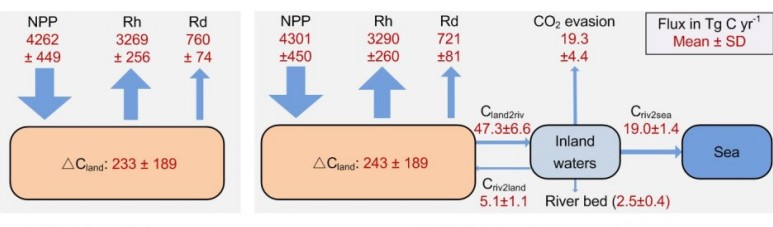

**Figure 10** The simulated average annual carbon budget of the terrestrial ecosystem in Europe
during the 1901-2014 when the lateral carbon transport is ignored (a) and considered (b). All
fluxes are presented as mean ± standard deviation. NPP is the net primary production. Rh and Rd
are the heterotrophic respiration and the respiration due to disturbances like harvest and land
cover change, respectively. $\Delta C_{land}$ is the average annual changes of the total land carbon stock.
Percentage following each of these changes in blue is the average annual relative changes of the



corresponding carbon pool. $C_{land2riv}$, $C_{riv2land}$ and $C_{riv2sea}$ are the average annual carbon fluxes
from land to inland waters, from inland waters to river and from inland waters to the sea,
respectively. SD is the standard deviation.

Besides direct transfers of organic carbon from soil to aquatic systems, the lateral transport of
water, sediment and carbon can also affect the land carbon budget through several indirect ways.
First, the lateral redistribution of surface runoff can affect the land carbon budget by altering soil
wetness. Our simulation results reveal that the lateral redistribution of runoff can significantly
change local soil wetness, especially in floodplains (Fig. S10b), where the increase in soil
wetness can be larger than 10% (Fig. S13b). Soil wetness is a key controlling factor of plant
photosynthesis (Knapp et al., 2001; Stocker et al., 2019; Xu et al., 2013). Benefiting from the
increase in soil wetness, the NPP in many grid cells with a large area of floodplain has increased
by more than 5% (Fig. 10b), although the NPP over the whole Europe only increased by 1%
(Fig. 10). Changes in soil wetness can further alter soil temperature (Fig. S13a). As soil wetness
and temperature are the two most important controlling factors of organic matter decomposition,
the lateral redistribution of runoff can affect local land carbon budget by changing the Rh.
Moreover, in ORCHIDEE-C$_{lateral}$, the turnover times of litter and SOC under flooding waters are
set to be three times of the litter and SOC turnover times in upland soil (Reddy & Patrick Jr,
1975; Neckles & Neill, 1994; Lauerwald et al., 2017). Accounting for flooding thus decreases
the decomposition rate of litter and SOC stored in floodplain soils.
Second, soil erosion and sediment deposition can affect land carbon budget by altering the
vertical distribution of litter and soil organic carbon. At the net erosion sites of the uplands, the
loss of surface soil results in a part of the belowground litter and SOC that were originally stored
in deeper soil layers emerging to the surface soil layers, and also results in a fraction of the
belowground litter becoming the aboveground litter. In the floodplains, the newly deposited
sediment becomes the new surface soil layer, and the belowground litter and SOC in the original
surface soil layer is transferred down to the deeper soil layers. As the temperatures and fresh
organic matter inputs (sourced from the aboveground litterfall and dead roots), which can impact
SOC decomposition rates through the priming effect (Guenet et al., 2016; Guenet et al., 2010), in
different soil layers are different, changes in the vertical distribution of belowground litter and



SOC can therefore lead to changes in the overall decomposition rate of the organic matter in the
whole soil column.
Third, soil aggregates mostly break down during soil erosion and sediment transport, the riverine
POC thus loses part of its physically protection from decomposition (Hu and Kuhn, 2016; Lal,
2003). Some modelling studies have assumed that at least 20% of the eroded SOC would be
decomposed during the soil erosion and transport processes (Lal, 2003, 2004; Zhang et al.,
2014). However, the estimation by Smith et al. (2001) using a conceptual mass balance model
suggest that only a tiny fraction of the eroded POC is decomposed and released as $CO_2$ to the
atmosphere. Using laboratory rainfall-simulation experiments, van Hemelryck et al. (2010)
estimated a 2%-12% mineralization of the eroded SOC from a loess soil, and Wang et al. (2014)
estimated a mineralization of only 1.5%. In ORCHIDEE-$C_{lateral}$, the passive SOC pool is
regarded as the SOC associated to soil minerals and protected by soil aggregates. The turnover
time of the passive POC in river stream and flooding waters is assumed to be same to that of the
active POC (0.3 year). Our simulation results suggest that the fraction of total riverine POC that
is decomposed during the lateral transport from uplands to the sea is 2.9% in Europe (Fig. 7b),
and the acceleration of POC decomposition rate due to the breakdown of soil aggregates can thus
slightly affect the estimate of the regional land-atmosphere carbon flux. Moreover, the riverine
POC and DOC can be transported over a long distance and finally settle or infiltrate in
floodplains or river channels (especially the Estuarine deltas) where the local environmental
conditions might be quite different from those encountered  in the uplands from where these C
pools originate. These changes in environmental conditions can affect the decomposition rate of
the laterally redistributed organic carbon (Abril et al., 2002).
Comparison between the simulation results from ORCHIDEE-$C_{lateral}$ with activated and
deactivated erosion and river routing modules indicate that the ignoring of lateral carbon
transport processes in LSM may lead to significant biases in the simulated land carbon budget
(Figs. 10 and S11). Although the omission of lateral carbon transport in ORCHIDEE-$C_{lateral}$ only
resulted in a 1% decrease in simulated average annual total NPP in Europe during 1901-2014
and a 1% increase of annual total Rh, the annual total NBP (=NPP–Rh–Rd–DOC and POC to
river) is underestimated by 4.5%. Over the same period, the lateral carbon transport only induced
a 0.09% increase in the total SOC and DOC stock in Europe (Fig. S12c), but their spatial

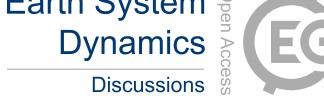

distribution was significantly altered (Figs. 11e,f). For instance, in some mountainous regions,
the soil erosion induced a reduction of the SOC stock by more than 8%. On the contrary, the
sediment and POC deposition in some floodplains led to an increase in SOC stock by more than
8% (Fig. 11f).

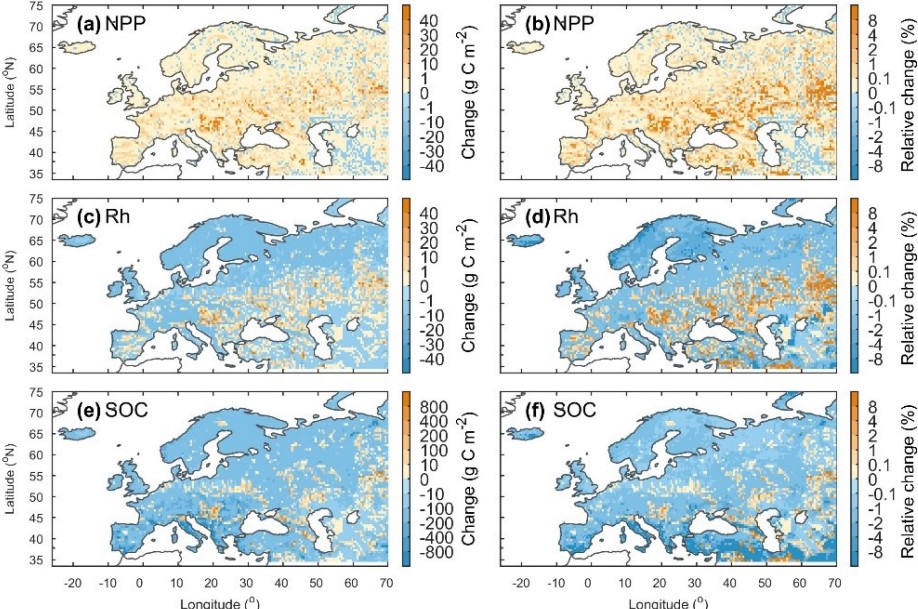


**Figure 11** Changes (first column) and relative changes (second column) of the net primary
production (NPP), heterotrophic respiration (Rh) and total soil organic carbon (SOC, 0-2 m) in
Europe due to the lateral carbon transport during 1901-2014. For each variable, the change is
calculated as $C_{lat}$ - $C_{nolat}$, where $C_{lat}$ and $C_{nolat}$ are the carbon fluxes or stocks when lateral carbon
transport is considered and ignored, respectively. The relative changes is calculated as ($C_{lat}$ -
$C_{nolat}$) / $C_{nolat}$ × 100%.

**3.4 Persisting short comings and future work**
Although most processes related to lateral carbon transport have been represented in
ORCHIDEE-$C_{lateral}$, there are still omitted processes and large uncertainties in our model. For
example, many studies suggest that a substantial portion of the eroded sediment and carbon is
deposited downhill at adjacent lowlands as colluviums, rather than exported to the river (Berhe et



al., 2007; Smith et al., 2001; Stallard, 1998; Wang et al., 2010). As the deposition of sediment and carbon within headwater basins can also significantly alter the vertical SOC profile and soil micro-environments (e.g. soil moisture, aeration and density) (Doetterl et al., 2016; Gregorich et al., 1998; Wang et al., 2015; Zhang et al., 2016), omission of this process may result in uncertainties in the simulated vegetation production and SOC decomposition. In addition, the impact of artificial dams and reservoirs on riverine sediment and carbon fluxes is also not represented in our model. Construction of dams generally leads to increased water residence time, nutrient retention, and sediment and carbon trapping in the impounded reservoir (Maavara et al., 2017), and can also affect the downstream flooding regime and frequency (Mei et al., 2016; Timpe and Kaplan, 2017). Estimation from Maavara et al. (2017) suggests that the organic carbon trapped or mineralized in global artificial reservoirs is about 13% of the total organic carbon carried by global rivers to the oceans. To more accurately simulate the lateral carbon transport, we plan to include the soil and carbon redistribution within headwater basins and the effects of dams and reservoirs on riverine sediment and carbon fluxes into our model in the near future.

The effects of lateral redistribution of water and sediment on vegetation productivity has not been fully represented in our model. As shown above, our model is able to represent the impacts of lateral water redistribution on vegetation productivity though modifying local soil wetness (Figs. 11 and S13). However, in addition to modifying soil wetness, many studies have indicated that the soil erosion and sediment deposition can affect vegetation productivity by modifying soil nutrient (e.g. e.g. nitrogen (N) and phosphorus (P)) availability (Bakker et al., 2004; Borrelli et al., 2018; Quine, 2002; Quinton et al., 2010). Recently, terrestrial N and P cycles have already been incorporated into another branch of ORCHIDEE (i.e. the ORCHIDEE-CNP developed by Goll et al., 2017). By coupling our new branch and ORCHIDEE-CNP, it will be possible to develop a more comprehensive LSM that can also simulate the effects of lateral N and P redistribution on vegetation productivity.

Although soils are the major source of riverine organic carbon, domestic, agricultural and industrial wastes, as well as the river-borne phytoplankton can also make significant contributions (Abril et al., 2002; Meybeck, 1993). Moreover, previous studies have shown that sewage generally contains highly labile POC and most of the aquatic production can be



mineralized within a short time (Abril et al., 2002; Caffrey et al., 1998). Omission of organic
carbon inputs from manure, sewage and river-borne phytoplankton may be one of the main
reasons for the underestimation of $CO_2$ evasion in the European river network, compared to the
estimates using statistical models based on observed riverine DOC concentrations (Lauerwald et
al., 2015; Raymond et al., 2013). Inclusion of these additional carbon sources should thus help
reconcile simulated and observed riverine carbon concentrations and aquatic $CO_2$ evasion.
Uncertainties in our simulation results also stem from the forcing data (Table 1) applied in our
model. The routing scheme of water, sediment and carbon is driven by a map of stream flow
direction at 0.5° spatial resolution (Guimberteau et al., 2012). Comparison between this flow
direction map and the flow direction map derived based on high resolution (3″) DEM show
discrepancies between the two river flow networks (Fig. S5). As the flow direction directly
determines the area of each catchment and the route of river flows, errors in forcing data of flow
direction may thus induce uncertainties in the simulated riverine water, sediment and carbon
discharges. Land-cover maps are another source of uncertainty. For instance, croplands generally
experience significantly larger soil erosion rates than grasslands and forests (Borrelli et al., 2017;
Nunes et al., 2011; Zhang et al., 2020). However, croplands in ORCHIDEE are only represented
in a simplified way by segmenting them into C3 and C4 crops based on their photosynthesis
characteristics. Therefore, our simulations based on land cover data with only two broad groups
of crop might not be able to fully capture the seasonal dynamics of planting, canopy growth rate
and harvesting for all crop types. Furthermore, the effects of soil conservation practices, which
would decrease erosion rates, are ignored in our model. Panagos et al. (2015) have shown that
contour farming, stone wall and grass margin techniques have been applied in Europe reduce the
risk of soil erosion. However, these soil conservation practices only reduce the average erosion
rate in European Union by 3%. Excluding soil conservation practices thus should have limited
impact in our simulation results.
Further model calibration and evaluation, especially using observation data from regions outside
of Europe, is necessary. In ORCHIDEE-$C_{lateral}$, an empirical equation (Eq. 8) adapted from the
WBMsed model, which was originally proposed to simulate the total suspended sediment
discharge in global rivers (Cohen et al., 2014), is used to estimate the transport capacities of clay,
silt and sand sediment. By calibrating the parameters controlling sediment transport capacity and





the deposition rate of excess suspended sediment (Table A1) against observed sediment
discharge rate in major European rivers (e.g. Rhine and Danube river), our model can overall
capture the sediment discharge rate in many European rivers (Fig. 3). Even so, there are still
large uncertainties in the simulated sediment discharge rate (Fig. 3), and it is unknown whether
our model would perform satisfactorily in regions with very different climates than Europe (such
as in the tropical regions). Thus, in the future, the aim is to extend the model applications to
contrasted regions or even the globe to refine the calibration of model parameters and evaluate
its ability to on predict the lateral sediment and carbon transport across a wide range of climate
regimes and terrestrial biomes. Moreover, the GLORICH database (Hartmann et al., 2019) only
provides instantaneous observations of riverine organic carbon concentrations and it is therefore
difficult to evaluate the model performance at annual or decadal scales. Therefore, future
modelling efforts should be combined with a data mining effort targeting the collection of more
continuous (e.g. daily) and long-term observational data of organic carbon content and fluxes in
streams and rivers.

**Conclusions**
By merging ORCHILEAK (Lauerwald et al., 2017) and an upgraded version of ORCHIDEE-
MUSLE (Zhang et al., 2020) for the simulation of DOC and POC from land to sea, respectively,
we developed ORCHIDEE-C$_{lateral}$, a new branch of the ORCHIDEE LSM. ORCHIDEE-C$_{lateral}$
simulates the large-scale lateral transport of water, sediment, POC, DOC and $CO_2$ from uplands
to the sea through river networks, the deposition of sediment and POC in river channels and
floodplains, the decomposition POC and DOC during fluvial transport and the $CO_2$ evasion to
the atmosphere, as well as the changes in soil wetness and vertical SOC profiles due to the lateral
redistribution of water, sediment and carbon.
Evaluation using observation data from European rivers indicate that ORCHIDEE-C$_{lateral}$ can
satisfactorily reproduce the observed riverine discharges of water and sediment, bankfull flows
and organic carbon concentrations in river flows. Application of ORCHIDEE-C$_{lateral}$ to the entire
European river network from 1901 to 2014 reveals that the average annual total carbon delivery
to streams and rivers amounts to 47.3±6.6 Tg C yr$^{-1}$, which corresponds to about 4.7% of total
NEP and 19.2% of the total NBP of terrestrial ecosystems in Europe. The lateral transfer of



water, sediment and carbon can affect the land carbon dynamics through several different
mechanisms. Besides directly inducing a spatial redistribution of organic carbon, it can also
affect the regional land carbon budget by altering vertical SOC profiles, as well as the soil
wetness and soil temperature, which in turn impact vegetation production and the decomposition
of soil organic carbon. Overall, omission of lateral carbon transport in ORCHIDEE potentially
results in an underestimation of the annual mean NBP in Europe of 4.5%. In regions
experiencing high soil erosion or high sediment deposition rate, the lateral carbon transport also
changes total SOC stock significantly, by more than 8%.
We recognize that ORCHIDEE-C$_{lateral}$ is still entailed with several limitations and significant
uncertainties. To address those, we plan to enhance our model with additional processes, such as
sediment deposition at downhills or the regulation of lateral transport by dams and reservoirs.
We also plan to calibrate and evaluate further our model by extending the observational dataset
to regions outside Europe.



**Code and data availability**

The source code of ORCHIDEE-Clateral model developed in this study is available online
(https://doi.org/10.14768/f2f5df9f-26da-4618-b69c-911f17d7e2ed) from 22 July, 2019. All
forcing and validation data used in this study are publicly available online. The specific sources
for these data can be found in section Table 1.

**Author contributions**

HZ, RL and PR designed the study. HZ and RL conducted the model development and
simulation experiments. PR, KV, PC, VN, BG and WY provided critical contribution to the
model development and the design of simulation experiments. HZ conducted the model
calibration, validation and the data analysis. RL, PR, PC, KV and BG provided support on
collecting forcing and validation data. HZ, RL and PR wrote the manuscript. All authors
contributed to interpretation and discussion of results and improved the manuscript.

**Competing interests**

The contact author has declared that neither they nor their co-authors have any competing
interests.

**Acknowledgements**

HZ and PR acknowledges the 'Lateral-CNP' project (No. 34823748) supported by the Fonds de
la Recherche Scientifique –FNRS and the VERIFY project that received funding from the
European Union's Horizon 2020 research and innovation program under grant agreement No.
776810. RL and PC acknowledge funding by the French state aid managed by the ANR under
the "Investissements d'avenir" programme [ANR-16-CONV-0003_Cland]. P.R. received funding
from the European Union's Horizon 2020 research and innovation programme under Grant
Agreement no. 101003536 (ESM2025 – Earth System Models for the Future).



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
