# Peer review of "Estimating the lateral transfer of organic carbon through the European river"

_Earth System Dynamics, 2022_

## Referee Comment (RC1)

**Review of**

**Zhang et al.** ***Estimating the lateral transfer of organic carbon through the European river network using a land surface model.***
by: Haicheng Zhang, Ronny Lauerwald, Pierre Regnier, Philippe Ciais, Kristof Van Oost, Victoria Naipal, Bertrand Guenet, and Wenping Yuan

In the presented manuscript, two branches of the land surface model (LSM) ORCHIDEE, the ORCHILEAK branch (dissolved organic carbon (DOC) processes in the soil and leakage to surface runoff) and the ORCHIDEE-MUSLE branch (DOC and $CO_2$ transport in rivers) are merged and enhanced by an explicit lateral river transport scheme for 3 particulate organic carbon (POC) and 3 sediment classes. The resulting ORCHIDEE-$C_{lateral}$ of Zhang et al. targets and is evaluated against the European river network. By comparing the model results to observations of European river discharge, sediment discharge rates, total organic carbon (TOC), DOC and partially POC concentrations, the study aims and provides insights into i) the lateral redistribution of organic carbon through the European river network and its effect on vegetation/terrestrial ecosystem budgets ii) $CO_2$ fluxes and iii) loss of carbon and sediments to the marine environment.

In general, the manuscript is well written and the content presents a valuable contribution in determining lateral fluxes and sediment/TOC budgets and their effects on ecosystem production with the aid of large scale LSMs. While not being an expert in global land surface modeling (-disclaimer-), I would recommend to publish the manuscript after addressing some major points outlined below.

**General comments**

My main concerns are centered around the rather low explanatory power of the model with regards to river DOM, TOC (Fig. 4) and particularly POC concentrations (Fig. 6) - the latter being the main development step of the new ORCHIDEE$_{lateral}$ branch. I understand that modeling such a complex river network is extremely challenging. Particularly when considering the trade offs between computational resources and limited availability of observations (for model input, parametrization and comparison/evaluation), this can lead to such deviations on seasonal timescales, while model results being still robust and valuable for long term simulations on (annual,) decadal and centennial timescales. Nevertheless, I would suggest to consider the following points:

- First, if I am not mistaken, calculating TOC-DOC=POC (according the authors caption in Fig. 4) would likely provide you with a much higher number of river measurements for POC than currently shown in Fig. 6 (assuming here that there is some overlap between TOC and DOC measurements for the river stations). Hence, a more direct comparison and evaluation between the new model development step and observations could (and should) be achieved beyond the river Rhine and Ems comparison demonstrated in Fig. 6.

- Second, since total mass fluxes are calculated via water volume transport times particulate and dissolved matter concentration, it would be helpful for the reader to show and discuss such comparison to observations in addition to the shown material (i.e. is it improving correlations compared to correlations for concentrations alone or further deteriorating them? - e.g. due to timing and/or the seasonal variability, see next point).

- Third, even though the model is targeted at large and long time scale simulations, the authors point out their models capability to potentially capture also sub-year time scales like e.g. seasons. Capturing variability is often an issue for models. The here presented model seem to overestimate the seasonal variability for DOC and TOC, while not for POC concentrations - why is that? I highly encourage a discussion on the variability. I believe it would be very valuable to investigate and discuss it (e.g. with regards to seasonality e.g. with a relative deviation to observations to enable better comparison between rivers in the time domain). It could provide insights into potential consistent deviation pattern and origins of variability (to me it looks as if there is a blurred, but consistent pattern). I suggest to discuss the potential origins of the too high (DOC,TOC) and low (POC) variability in the light of the model shortcomings and potential future development steps.

**Minor comments**

While I am not a native speaker, I make some wording suggestions below.

- Generally, I would encourage to place the two tables being currently in the supplementary material into the main text.

**Main document**

l.60 predicting or projecting?

l.75 have been developed

l.78/79 how does this eventually relate to the seemingly lower POC than DOC concentrations in rivers and to the results presented? $\rightarrow$ discussion

l.86 How about new production in rivers?

l.154 go to $\rightarrow$ enter

l.180 a forcing file

l.191 by a basin-specific

l.207 finer

l.220 Sediment and particulate organic carbon delivery... (opposed to $CO_2$ carbon)
Maybe a question of a non-initiated reader (and potentially something to clarify): are basin and grid cell in your model description interchangable or how is a 'basin' defined in the model realm?

l.230 check the units - also for $a$

l.254 sentense ends abruptly - something is missing

l.255 typesetting + space

l.257 conditions to ...? something seems to be missing

Fig.1 since $POC_{a,p,s}$ appear later in the text, please add the explanation in the caption. Further, I suspect, it's a typo: $POC_c \rightarrow POC_s$?

l.302f is the $S_{fast\_h2o}$ correctly placed? - I suspect, it should be after 'water reservoir', since the residence time is $\tau_{fast}$ or not?

l.305 from the fast water reservoir into the stream reservoir

l.308 $\tau$ is thus far undefined, if I am not mistaken

l.311 enters the stream reservoir

l.313 transports

l.316f sediment in the stream ... determined by $F_{Fout\_sed}$ ... sediment input by flooding water

l.318 maybe a matter of specialization of science communities, but is there a difference between resuspension (or erosion) and re-detachment of sediment? (also in Fig.1)

l.343 each grid, and we thus require/apply a different approach described in the following (or something similar - as a reader, I was a bit lost with this last sentence)

l.408 different particle sizes

l.447ff Why did you chose the turnover time for the passive soil organic carbon content the same as the active one and not as the slow one? This also puzzles me a bit later on in the discussion, where you seem to provide good reasons for slower turnover times when soil passive organic carbon is released to rivers (see below).

l.456 I believe this is a bit misleadingly written. I suspect that you mean that the deposited sediment becomes part of the surface soil layer (and does not become a new layer - otherwise, one would end up with lots of layers)

l.467 over Europe and parts of Middle East and Africa (...)

l.470 is there any reason for 2 days (and not for one or more than 2)?

l.473 all the flooding waters

l.476 First you make the point that there is substantial spatial variation of $P_{flooding}$ and then you assume it the same for Europe. Why and what are the consequences? - also for your later comparison, l. 506/507

l.478 Following Zhang et al.

l.493 I suspect you want to refer to Eq. 10 (not 5)?

l.503 as forcing data

l.507 (locations see Fig...)

l.516f if I interpret your caption of Fig. 4 correctly (and to my knowledge), you can derive POC content by TOC-DOC=POC, which should result in a much larger data base you can compare your model simulations to, right? See my main comments.

l.607ff As mentioned in the main comments part, I would like to see the temporal variability of the simulation in contrast to observations more in-depth discussed (also for POC). I believe understanding this model behavior better could provide important insights.

l.613 Note that (typo)

l.625-634 this potentially requires re-working, if TOC-DOC=POC

Fig.7 I might misinterpret here something, but isn't upland loss = delivery to river? If it is, then I cannot relate the values for DOC and POC loss in Fig.7 to the DOC and POC delivery shown in Fig. S11b of the supplementary material (where POC$\approx 18\,\mathrm{Tg\,C\,yr^{-1}}$ and DOC$\approx 7\,\mathrm{Tg\,C\,yr^{-1}}$). Otherwise, some explanations would be helpful to understand the discrepancies.

l.682 Although the ... (no 'but')

l.684 rivers are much smaller

l.723 NEP is undefined, or do you mean NPP? (also in the next line)

l.759 becomes part of the surface soil layer (see comment above)

l.771ff As written above, I am a bit puzzled about the choice of the turnover time of passive POC in rivers. If I am getting you right, you try to justify the parameters value choice with the decomposed fraction of the overall POC pool. Since you unfortunately never show the fractions of fast, slow and passive POC (would be nice to find such plot ion the supplementary material), I wonder, which of the POC contributes most to the overall decomposition loss. How relevant is the choice of this value for your results?

l.787 that ignoring lateral

l.791 here NBP is differently defined than before, if NEP was not a typo (see before)

l.820 Estimations by Maavara

l.831 cancel out one of the 'e.g.'

l.843+846 I am a bit uncertain, if evasion is the right word here, maybe better 'efflux' (or net outgassing)?

l.873 do you mean: Even though there are still . . .

l.877 contrasting regions (?)

l.878 to predict

l.879ff I am a bit puzzled with regards to the models versus the observations variability at this point. Usually, I would expect that the instantaneous observations show larger scatter than e.g. monthly averaged model simulations. In your simulations, the model range for DOC and TOC is often even beyond the mean envelope range of observations (Fig. S7 and S8). While I understand that the model cannot capture the observational instantaneous values, I would at least expect/tune it to capture annual (or seasonal) to decadal mean values and the envelope (or are there any known biases in the data base with this respect?). Hence, I am not sure, where the origin of this mismatch is coming from and if I can follow that the problem arises only from too little river observations (while I agree that more -long term- observations would help in other aspects like seeing trends, etc.).While not being an expert in LSM modeling, I feel that the model lacks some (potentially important) sort of buffering or counteracting mechanism that modulates down the DOC amplitudes (and other processes that increase the POC variability...). As pointed out above, I believe it would be of value to investigate this deeper and to discuss potential reasons for this model behavior.

l.900 NEP?

**Supplementary material**

l.11 description of $\omega$ sediment transport capacity (typo)

l.17 shown

l.18 investigated time of two years

l.23 used in this study? (something is missing here)

l.25 (. . . ) of these 57

Fig. S3 no response of DOC and $CO_2$ to $\omega$, $c_{ebank}$ and $e_{ebed}$? - maybe worth to note this explicitly in the caption (or wasn't it analyzed for these parameters?)

Fig. S4   provide offsets and potentially standard deviations. You mention the offset for the Danube river delta in the text, but are there similar explanations for the Elbe (d) and particularly the Rhine (the latter seem even to be underestimated)? I would suggest to extend this discussion part in the main text (l.528ff)

Fig. S11   see comment in the main text on budgets of POC and DOC

---

## Referee Comment (RC2)

**Comments on Zhang et al „Estimating the lateral transfer of organic carbon through the European river network using a land surface model" submitted to ESDyn**

The authors present modelling results of lateral OC transfers based on an improved representation of POC and DOC transfer using the ORCHIDEE LSM model. Model outputs cover European catchments during the period from 1901 to 2014. The authors provide very interesting results on various OC budget component for the given period, which have not been published at the European scale (e.g. how much sediment/POC is stored in floodplains and exported to oceans). Furthermore, the authors compare the C budget with and without lateral C transfer and provide interesting information on the discussion of the net effect of lateral sediment transfer of the carbon budget. Overall, the study is of great interest and certainly the develop model provides a significant step forward on that topic (as far as I can judge this from a non-modeler point of view). I recommend this manuscript to be published in ESDyn after major revisions. Please consider the general and more detailed comments below to improve the submitted manuscript.

Kind regards
Thomas Hoffmann

**General comments**
It is very hard to understand the model setup as a non-model expert and the text is not very communicative to motivate and convince non-model experts to read the paper. I suggest that the authors should critically prove how they can motivate the linkages between the empirical evidences and the model setup. I had hard times to understand the general model approach and still do not fully understand how the authors define headwaters and whether they differentiate headwater sizes depending on the climate /lithology etc. But maybe I was blind and unable to extract the relevant information. In any case, many assumptions/ approaches are not straight forwards and description should be improved.

The general concept, as depicted in Figure 1, draw an arrow (F_up2fld) from the upstream basin to the floodplain. I was wondering if this transport path is needed? The main transfer to floodplains steams from the main river channel that is passing the floodplain. The potential inputs from upstream basins is of academic nature and could be neglected.

Major parts of Europe are missing in terms of observations of suspended sediment loads (e.g. no observation for Spain, France GB, Italy), esp. Mediterranean rivers are not covered in this study. Due to the very different behavior of streams in south and north Europe, the study is strongly biased toward N-Europe. This becomes even more important by the comparison with the model output from WATEM/SEDEM for the two catchments of the Apennine Peninsula, which are simply ignored as 'outliers'.

The authors provide model runs with and without lateral C transport and find that SOC stock only marginally increase of lateral flux is turned on. This very low increase somehow contradicts the large amount of POC retention in floodplains. The authors somehow provide

numbers with SOC stock decrease in mountains and SOC stock increases in floodplains, but does this mean that SOC retention in floodplains is more or less fully compensated by soil degradation at eroding sites? It was argued that at long-term ($10^3$a) OC retention in floodplains is more important than soil degradation, while at shorter terms (couple of years) degradation effect might dominate. I wonder what happens to the model if the authors considerer shorter and much longer time scales than those used in this study. Please discuss this in more detail.

**Detailed comments**
Line 38: 'but also leaching of DOC' → needs some more details, leaching from where?
Line 45: I suggest to refer to a new review on OC sequestration in floodplains (https://doi.org/10.1016/B978-0-12-818234-5.00069-9)
Line 169ff: the many branches/modules etc. make it very hard to understand the model setup. Could you somehow visualize it?
Line 228: I suggest to avoid sediment delivery rate, as this might be confused with sediment delivery ratio. Please use sediment supply instead. How are headwater basins defined in this study?
Line 229: given set f runoff and vegetation cover conditions → could you specify them and motivate, why you refer to the reference runoff condition rather than actual runoff. This might be explained in the original reference. However, I highly recommend to give more details here, to ease the understanding of the approach.
Line 232: Is this really a runoff or rather a precipitation reference (given the 10mm day-1)?
Line 237: DA^(dDA^c) → is that correct? Looks erroneous
Line 237: is assume that DA is drainage area (not defined)
Line 238: same as above; (p.5-6) should be linked once more to the reference where this citation is taken from.
Line 247-249: not sure what the authors want to say here? I guess the model outputs should depend on these parameters.
Line 250: ORCHILEAK $C_{lateral}$ → subscript lateral
Line 251: was this done for various reference conditions?
Line 254: awkward sentence; '…force the simulation of Then…' ?????
Line 260: Same b as in Eq.1? Where is $S_{iday}$ located in Figure 1?
Line 262: $R_{30\_k}$ not in Eq. 3 and 4
Line 304: Is $F_{Fout\_sed}$ identical with $S_{iday}$? Please remove sediment in this sentence, because this is confusing due to the fact that there is not storage in the fast water reservoir.
Figure 1a: Not sure what the direction of arrows indicates. I suggest that they point from the text to the feature in the graphic (if this is not related to vertical fluxes; unlikely for sediment). $S_{river}$ and $S_{flood}$ is used in Figure caption but not within the Figure itself.
Line 323ff: I wonder if the author mix up several things. In rivers, suspended sediment (esp silt and clay which are transport agents of POC) is transported as wash load. The transport of the wash load is not transport capacity limited but supply limited. Whether changes in the channel bed need to be considered depends on the target time scale. Therefore, I am not sure if it is required to discuss Eq. 7 in detail. If the authors specify the relevant scales much earlier in their paper, the lengthy discussed could be reduced.
Line 357: what is e1 in Eq.1?
Line 361: Drainage area in Eq 1 was defined with DA. Use same symbols!

Line 361: In Eq. 8 the term F_down2riv_h20 is used, here in the text you use F_down2riv_sed but talk about water discharge. I am confused. I assume you refer to the Psi-equation of Cohen et.al. If this is true

Line 371: Assuming that channel bank erosion only occurs if no sediment is left at the channel bank is not a meaningful assumption. Many rivers migrate without changing their channel bed.

Line 387: F_up2fld_sed not needed in my point of view. Why was this introduced and and why is there no F_riv2fld?

Line 390: sum in text but negative sign in Eq. 16 → furthermore, I don't understand the approach here. Why does evaporation and infiltration contribute to sediment deposition? Please explain.

Line 400: Same f_topo as on hillslopes? How was this calculated? I am confused!

Line 485: I guess that you run in the problem of equifinality of you simple calibrate five parameters against one observation (sediment yield). Please discuss this problem.

Line 509: Major parts of Europe are missing (e.g. no observation for Spain, France GB, Italy), esp. Mediterranean rivers are not covered in this study.

Line 517: Indicate which stations in Rhine were used. POC is strongly discharge dependent, please indicate how many measurements at which discharge are used.

Line 606: It seems that the model underestimates the observed DOC variability (Fig. 4b), however, this is in contrast to the Figure S8. Please explain this discrepancy.

Line 649: How does this number relates to empirical sediment budgets? Is that in the order of obserations? Please discuss.

Line 661: any idea what causes this decline?

Line 679: are there any empirical values to compare with?

Fig. 10: Does C_riv2land represent the transport from river channels to floodplains? If yes, I suggest to consider floodplains not at 'land'.

Line 753: flooding decreases SOC stored in floodplain soil???? This is total contradicting our expectation and needs discussion

Line 747: can you account for the soil-wettness driven changes in soil temperature? Is this effect significant?

Line 754: any number how this influences the C budget. Many empirical studies argue that this effect is important and strongly increases the OC retention in floodplains. Could this somehow be quantified?

Line 793: this very low increase somehow contradicts the large amount of POC retention in floodplains. You somehow provide numbers with SOC stock decrease in mountains and SOC stock increases in floodplains, but does this mean that SOC retention in floodplains is compensated by soil degradation at eroding sites?

Line 811: Please cite Hoffmann 2013 (GBC): they present results for hillslope and floodplain storage of OC for the Rhine basin.

Line 826ff: Considering NP might not only decrease NPP at eroding site but also increase NPP at depositional site. Correct? If yes, leave some words in the paragraph on depositional sites as well. Certainly, a worthwhile action to link NP here.

Line 839: Hoffmann et al (2020, ESurf) provides a way to differentiate exsitu and insitu OC in rivers. This paper also offers more infos on POC in the Moselle and Rhine rivers.

Line 849: Could the routing be done using DEMs with better spatial resolution to overcome limitations of the routing on low-res DEMs?

Figure S2: bad quality, can't read the text

Figure s4: give names of gauging stations
Figure S5: bad quality of left map

---

## Referee Comment (RC3)

Review of

**Zhang et al. Estimating the lateral transfer of organic carbon through the European river network using a land surface model.**
by: Haicheng Zhang, Ronny Lauerwald, Pierre Regnier, Philippe Ciais, Kristof Van Oost, Victoria Naipal, Bertrand Guenet, and Wenping Yuan

The submitted manuscript describes a development of ORCHIDEE, in which several branches have been combined to better assess the global carbon cycle. As the authors state it has long been ignored that the carbon transport from land, via rivers to the ocean also plays an important role. I was developing something comparable a few years back and am very certain about the importance and high relevance of such a model enhancement.

The results are promising. But the description of the methods and the presentation of the results can be improved, e.g., there are lengthy quotes in the methods which can be reduced, or some of the table/figures in the SI should be moved to the main text for clarification and a better understanding.

Overall, I think the content of the manuscript is convincing but can be strengthened by removing unnecessary parts and moving some explaining information from the SI to the main text. I believe that the manuscript can be improved, and I recommend it for publication after major revision.

Below I list some unclear sentences, problematic points, and questions. This list is chronological and not based on the importance of the points. I was much more detailed in the methods, but think that the results/discussion section can also profit from the listed points.

- l.86. Is the litter also included as a part of OC input? It is mentioned a few times, but not explained or shown in Fig 1?
- Please also mention the branch names (e.g.in l. 114)
- An overview map of the catchments would be helpful or at least of how much of the drainage area of Europe is covered.
- l. 135. Is 'bare soil' a PFT (plant functional type)?
- l. 148: Are the two litter pools part of the POC in soil? Please clarify.
- ll. 156-162 is a quote. Please shorten or refer right away to the cited paper.
- l. 183. How does the DOC 'enter' the water? Does it depend on vegetation cover? Please clarify.
- Table 1: Is a spatial resolution of 0.5° (55km*55km max) sufficient to inform the model about the 'Area fraction of river surface'? Later it is mentioned that for the delta of the Danube the high resolution was problematic (because of gauging station data available). But is there also a problem of scale for other data? E.g. 'maximum water storage in river channel' is also only on 0.5°.
- l. 207. The temporal resolution of 6'' for $CO_2$ evasion is totally reasonable, but for the DOC decomposition it seems to be too much (as for the litter added, l. 212), also given the fact that the temperature (on which the decomposition is also depending) only changes every 3h.
- Why is there a water temperature of 28° assumed? Is this realistic?
- ll. 212-218. Again, a quote. Can be shortened. Is wind speed considered for the $CO_2$ evasion?
- ll. 227-238. Again, a pretty long quote.
- l. 243 mentions a 'management factor', which is only explained in l.264.

- ll. 250. It is not clear on which resolution the model runs. Some of the input is fine scale (e.g. 250m for floodplains), but then the results are aggregated to 0.5°. Please clarify.
- ll. 254-258. The sentence is not understandable.
- l. 268. Why only headwater?
- Figure 1. (I like it.) It would be helpful if the naming in part a and b would be consistent. Having the time steps mentioned in a separate box is helpful and keeps the figure clean, but it is not clear which processes belong to which group. E.g. 'deposition' - mentioned several times in the figure is not part of the 'time step' box; 'decomposition' in the time-step box consists of 'litter and SOC' but the litter can't be found in the figure. It could also be helpful to add the abbreviations from part b to part a, to make the link clearer and easier to understand. (b) is missing in the caption.
- ll. 329-343 and ll. 344-353. This is much more discussion than methods. Can be shortened and should be moved.
- ll. 384. What is the effect of scale here if you move the OC from one catchment to the next? How does is differ between large and small catchments?
- Section 2.2.2. What is the exact difference between sediment flow and the described POC flow, if it's closely linked to sediments? Is the POC calculations the same as clay?
- l. 438. You use the water temperature to calculate the processes, but as input you use air temperature. How do you accommodate for the difference and the time-lag (e.g. over the course of a day)?
- ll. 442. Why do you refer and explain so much about SOC here? The section title is 'POC transport and decomposition'. Maybe some reference to the SOC section would help to clarify.
- L. 468. You refer to Figure S1 in the SI, but this figure shows the return period over two years.
- l. 484. A map or a list of the catchments would be helpful (as in SI Figure S2(d)).
- l. 549. Why is that set to 0.1? It is not clear at this point.
- Fig 3-5 are convincing.
- Section 3.1.3. Your model simulates a too low SOC stock while the TOC and DOC concentrations look better. I would also discuss more here that the temporal pattern of observation and simulation does not match (Fig. S 8), although the mean looks promising.
- Fig 5. I would add the names of the rivers as a side panel instead of having the letters.
- Fig. 6. I suggest to add a panel for discharge (observed vs. simulated).
- ll. 645. I am glad to read about the effect of the vegetation type more explicitly here. It could be mentioned earlier.
- l. 662. Is the difference between 3.0 Pg and 2.3 Pg significant? In the corresponding figure S11a there is a lot of fluctuations over the years.
- Fig 7a. The percentage for the fluxes does not sum up to 100. Why?
- l. 682. You mention several times 'small rivers'. Did you conduct a classification for the rivers or is it more a vague grouping?
- Section 3.3. There are several strangely set parentheses.
- ll. 738. I like this clear listing and explanation (first, second, third). It makes it easier to follow the argumentation.
- ll. 752. What is the effect of anaerobic conditions in the sediment? Wouldn't the decomposition be lower then?
- Section 3.4. I think that is a necessary part of papers like that. Thank you.

Some of the questions have been answered later in the manuscript but already at that (above mentioned) point I would like to know.

There are also some more colloquial phrases such as 'basically' (e.g. l. 299) or 'so-called' (e.g. l. 302 and 303). I suggest exchanging or removing them.

---

## Author Comment (AC1)

**Referee #1**

**1. In the presented manuscript, two branches of the land surface model (LSM) ORCHIDEE, the ORCHILEAK branch (dissolved organic carbon (DOC) processes in the soil and leakage to surface runoff) and the ORCHIDEE-MUSLE branch (DOC and CO2 transport in rivers) are merged and enhanced by an explicit lateral river transport scheme for 3 particulate organic carbon (POC) and 3 sediment classes. The resulting ORCHIDEE-Clateral of Zhang et al. targets and is evaluated against the European river network. By comparing the model results to observations of European river discharge, sediment discharge rates, total organic carbon (TOC), DOC and partially POC concentrations, the study aims and provides insights into i) the lateral redistribution of organic carbon through the European river network and its effect on vegetation/terrestrial ecosystem budgets ii) $CO_2$ fluxes and iii) loss of carbon and sediments to the marine environment.**

**In general, the manuscript is well written and the content presents a valuable contribution in determining lateral fluxes and sediment/TOC budgets and their effects on ecosystem production with the aid of large scale LSMs. While not being an expert in global land surface modeling (-disclaimer-), I would recommend to publish the manuscript after addressing some major points outlined below.**

Thanks a lot for your positive comments, as well as your suggestions and corrections below. Your comments are very helpful to improve our manuscript. We have carefully studied them and revised our manuscript accordingly. Please see our responses to your specific comments below.

**General comments**

**2. My main concerns are centered around the rather low explanatory power of the model with regards to river DOM, TOC (Fig. 4) and particularly POC concentrations (Fig. 6) - the latter being the main development step of the new ORCHIDEE_Clateral branch. I understand that modeling such a complex river network is extremely challenging. Particularly when considering the tradeoffs between computational resources and limited availability of observations (for model input, parametrization and comparison/evaluation), this can lead to such deviations on seasonal timescales, while model results being still**

**robust and valuable for long term simulations on (annual,) decadal and centennial timescales. Nevertheless, I would suggest to consider the following points:**

**First, if I am not mistaken, calculating TOC-DOC=POC (according the authors caption in Fig. 4) would likely provide you with a much higher number of river measurements for POC than currently shown in Fig. 6 (assuming here that there is some overlap between TOC and DOC measurements for the river stations). Hence, a more direct comparison and evaluation between the new model development step and observations could (and should) be achieved beyond the river Rhine and Ems comparison demonstrated in Fig. 6.**

Thanks for your suggestion. Unfortunately, the GLORICH database does not provide any additional samples for which both TOC and DOC would have been observed. In fact, TOC appears most often in the database as a variable replacing DOC measurements, likely when sampled water was not filtered (Ronny Lauerwald, personal communication, co-author in Hartmann et al., 2019). Furthermore, we think that even if TOC and DOC would have been observed independently (from unfiltered and filtered samples), POC values calculated as difference between both variables would be associated with high uncertainties, and thus not necessarily improving the dataset used for model evaluation.

Hartmann, J., Lauerwald, R., and Moosdorf, N.: GLORICH - Global river chemistry database, in: PANGAEA (Ed.), 2019.

**3.  Second, since total mass fluxes are calculated via water volume transport times particulate and dissolved matter concentration, it would be helpful for the reader to show and discuss such comparison to observations in addition to the shown material (i.e. is it improving correlations compared to correlations for concentrations alone or further deteriorating them? - e.g. due to timing and/or the seasonal variability, see next point).**

We have added two supplementary figures to show the comparison between simulated and observed riverine carbon flux rates, in addition to the riverine carbon concentration. As our model captures well the seasonal and interannual variations in riverine water discharge rates, the simulated seasonal and interannual variations are closer to observations for riverine carbon flux

rates than they are for riverine carbon concentrations. Please see Figs. 6, S11 and S12 for further details.

[Figure]

**Figure 6** Comparison between observed (instantaneous measurements) and simulated (monthly average values) riverine POC concentrations and POC discharge rates at three gauging sites. The histograms and error bars denote the means and standard deviations of POC concentrations, respectively. Long-term average water discharge rates at Bad Honnef, Bimmen and Rheine during the observation periods are 2023, 2100 and 80 $m^3$ $s^{-1}$, respectively.

[Figure]

**Figure S11** Comparison between simulated and observed discharge rates of total organic carbon (TOC) in representative European rivers. DA is the drainage area of the corresponding gauging station.

[Figure]

**Figure S12** Comparison between simulated and observed discharge rates of dissolved organic carbon (DOC) in representative European rivers. DA is the drainage area of the corresponding gauging station.

**4. Third, even though the model is targeted at large and long time scale simulations, the authors point out their models capability to potentially capture also sub-year time scales like e.g. seasons. Capturing variability is often an issue for models. The here presented model seem to overestimate the seasonal variability for DOC and TOC, while not for POC concentrations - why is that? I highly encourage a discussion on the variability. I believe it would be very valuable to investigate and discuss it (e.g. with regards to seasonality e.g. with a relative deviation to observations to enable better comparison between rivers in the time domain). It could provide insights into potential consistent deviation pattern and origins of variability (to me it looks as if there is a blurred, but consistent pattern). I suggest to discuss the potential origins of the too high (DOC,TOC) and low (POC) variability in the light of the model shortcomings and potential future development steps.**

Thank you for this highly valuable suggestion. We have investigated the potential reasons for the overestimated seasonal variations in riverine DOC and TOC. Basically, the concentration of soil DOC and the DOC decomposition rate during the lateral transport process in the river network are the two key factors controlling DOC concentration in river flow. Our simulation suggests that only a limited fraction (<20%) of the riverine DOC is decomposed during transport along the river network (Fig. 7). Therefore, we suggest that the uncertainty in the seasonal dynamics of simulated soil DOC concentrations is the main reason for overestimating the seasonal variation in riverine DOC concentrations. In fact, the soil DOC scheme used in the ORCHIDEE model is known to not fully capture the temporal dynamics of soil DOC concentrations (Camino-Serrano et al., 2018), although it reproduces the annual average DOC concentrations rather well. TOC concentrations are calculated as the sum of DOC and POC concentrations, and riverine TOC is often dominated by DOC (Fig. 7). Thus, the overestimated amplitude in TOC concentrations is mainly caused by the DOC seasonal dynamics in the topsoil and, thus, the leaching flux. Following your suggestion, we have added one paragraph to discuss why the seasonal variations TOC and DOC concentrations are overestimated and what future work is needed to improve the model. Please see:

"The simulation of the soil DOC dynamics and leaching in our model need to be further improved to better simulate the seasonal variation of riverine DOC and TOC concentrations. The concentration of soil DOC and the DOC decomposition rate during the lateral transport process in the river network are the two key factors controlling DOC concentration in river flow. As only a small fraction (< 20%) of the riverine DOC is decomposed during lateral transport (Fig. 7), the overestimated (Fig. 5) seasonal amplitude in riverine DOC (and TOC) concentrations is likely caused by the uncertainties in the simulated seasonal dynamics of the leached soil DOC. The current scheme used in our model for simulating soil DOC dynamics has been calibrated against observed DOC concentrations at several sites in Europe (Camino-Serrano et al., 2018). Although the calibrated model can overall capture the average concentrations of soil DOC, it is not able to fully capture the temporal dynamics of DOC concentrations (Camino-Serrano et al., 2018). Given this, it is necessary to collect additional observation data on the seasonal dynamics of soil DOC concentration to further calibrate the soil DOC model" (lines: 841-853)

Averaged over the various DOC and SOC pools we distinguish in the soils, DOC represents a much more reactive fraction of soil carbon (with a turnover time of several days to a few

months) than SOC (with a turnover time of decades to thousands of years). Therefore, soil DOC concentrations experience large seasonal variations, while SOC concentrations generally are much more stable and show very limited seasonal dynamics. Therefore, seasonal variations in riverine POC concentrations are mainly controlled by the seasonal dynamics of soil erosion rates, rather than by the seasonal SOC dynamics, which explains a partial decoupling in the behavior of POC compared to that of DOC. We have also added these texts in the revised manuscript to explain the different seasonal pattern of riverine DOC and POC concentrations. Please see lines: 853-861. In our study, we agree that the simulated riverine POC concentrations at three sites in central Europe are lower than the observed values (Fig. 6). The standard deviation of the simulated POC concentrations at these three sites thus is smaller than that of the observed POC concentrations. However, when the seasonal variation of POC concentrations is measured by the coefficient of variation, i.e. the standard deviation divided by the mean, the simulated seasonal variations are overall comparable to the observed one. Given this, we do not fully agree that our model underestimates the seasonal variation in riverine POC concentrations. At Rheine site of the Ems river, the seasonal dynamics of POC discharge rates are even somewhat overestimated (Fig. 6f).

Reference:

Camino-Serrano, M., Guenet, B., Luyssaert, S., Ciais, P., Bastrikov, V., De Vos, B., Gielen, B., Gleixner, G., Jornet-Puig, A., Kaiser, K., Kothawala, D., Lauerwald, R., Peñuelas, J., Schrumpf, M., Vicca, S., Vuichard, N., Walmsley, D., and Janssens, I. A.: ORCHIDEE-SOM: modeling soil organic carbon (SOC) and dissolved organic carbon (DOC) dynamics along vertical soil profiles in Europe. Geosci. Model Dev., 11, 937-957, 2018.

**Minor comments**

**5. While I am not a native speaker, I make some wording suggestions below.**

Thanks for your suggestions and corrections below. We have carefully studied them and revised our manuscript accordingly. Please see our responses to your specific comments below.

**6. Generally, I would encourage to place the two tables being currently in the supplementary material into the main text.**

We have moved the supplementary Table S1 to the main text according to your suggestion. As we have provided the definitions of all abbreviations in the main text, and the length of the main text exceeds 13,000 words, we prefer to keep the original Table S2 (i.e. the Table S1 in the revised manuscript) in SI.

**7. Main document**

**l.60 predicting or projecting?**

We replaced 'predicting' by 'projecting'. Please see "A reliable model which is able to explicitly simulate the lateral carbon flux along the land-river continuum and also the interactions between these lateral fluxes and the comprehensive terrestrial carbon cycle, would thus be necessary for projecting changes in the global carbon cycle more accurately." (lines: 58-61)

**8. l.75 have been developed**

We corrected the grammar error by changing the original 'has' to 'have'. Please see "Over the past decade, a number of LSMs have been developed which represent leaching of DOC from soils (Nakhavali et al. 2018, Kicklighter et al. 2013) or the full transport of DOC through the land-river continuum (Lauerwald et al., 2017; Tian et al., 2015)." (lines: 75-77)

**9. l.78/79 how does this eventually relate to the seemingly lower POC than DOC concentrations in rivers and to the results presented? → discussion**

With the sentence "However, the erosion-induced transport of POC, which is maybe even more important than the DOC transport in terms of lateral carbon flux (Lal., 2003; Tian et al., 2015; Tan et al., 2017), is still not or poorly represented in LSMs.", we do not intend to say the amount of lateral POC flux is larger than that of DOC, but to state that the lateral POC flux might have larger impact on terrestrial carbon cycle than DOC flux. Because the erosion and deposition processes of POC do not only result in lateral redistribution of the eroded POC, but also alter the vertical SOC profile at the eroding and deposition places, and soil erosion might also result in significant decrease of terrestrial vegetation production. We recognize that the relative importance of POC and DOC transfers on the terrestrial carbon cycle shows large spatial

variation. To ensure a more precise introduction, we have changed the original sentence to "However, the erosion-induced transport of POC, which has also been reported to be able to affect the carbon balance of terrestrial ecosystems strongly (Lal., 2003; Van Oost et al., 2007; Tian et al., 2015), is still not or poorly represented in LSMs." (lines: 77-80)

**10. l.86 How about new production in rivers?"**

From a technical point of view, implementation of instream autrotrophic production is highly challenging, beyond our possibilities for developing riverine C transfers into a LSM. However, although in-stream primary production can be an additional organic carbon source, field studies have indicated that terrestrial carbon is the major source of riverine organic carbon (e.g., Raymond & Bauer, 2001). In addition, in large rivers the majority of primary production (94%) tends to be locally respired rather than transported far downstream (Howarth et al., 1996). Given these reasons, we conclude that the exclusion of in-stream production in the current version of ORCHIDEE-Clateral model is not a critical omission to assess the importance of land-to-ocean C fluxes for the terrestrial C budget. Nonetheless, we have now discussed this shortcoming in the revised ms. Please see "Although soils are the major source of riverine organic carbon, domestic, agricultural and industrial wastes, as well as river-borne phytoplankton can also make significant contributions (Abril et al., 2002; Meybeck, 1993). Moreover, previous studies have shown that sewage generally contains highly labile POC while most of the aquatic production is generally mineralized within a short time (Abril et al., 2002; Caffrey et al., 1998). Omission of organic carbon inputs from manure and sewage could potentially lead to an underestimation of $CO_2$ evasion from the European river network. Inclusion of these additional carbon sources should thus help improve simulation of aquatic $CO_2$ evasion." (lines: 892-899)

Howarth, R. W., R. Schneider, and D. Swaney (1996), Metabolism and organic carbon fluxes in the tidal freshwater Hudson River, Estuaries, 19, 848–865.

Raymond, P. A., and J. E. Bauer (2001), Riverine export of aged terrestrial organic matter to the North Atlantic Ocean, Nature, 409(6819), 497–500.

**11. l.154 go to → enter**

We changed the 'go to' to enter. Please see "The products of litter and SOC decomposition enter free DOC pool". (line: 155)

**12. l.180 a forcing file**

We revised the text based on your comment. Please see "Simulation of the lateral transfer of DOC and $CO_2$ in river networks, i.e. the transfer of DOC and $CO_2$ from one basin to another based on the stream flow directions obtained from a forcing file (0.5°, Table 1)" (lines: 176-178)

**13. l.191 by a basin-specific**

We revised the text based on your comment. Please see "the DOC and dissolved $CO_2$ returning to river channel at the end of each day is calculated based on a time constant of flooding water (= 4.0 days, d'Orgeval et al., 2008) modified by a basin-specific topographic index ($f_{topo}$, unitless) (Lauerwald et al., 2017)." (lines: 187-190)

**14. l.207 finer**

We only describe the fine time step (i.e. 6 minutes) for simulating DOC decomposition and $CO_2$ evasion in inland waters, without comparing it with the time steps for simulating other processes (e.g. soil erosion and lateral C transfer). Thus we still use 'fine' rather than 'finer'.

**15. l.220 Sediment and particulate organic carbon delivery... (opposed to CO2 carbon)**

**Maybe a question of a non-initiated reader (and potentially something to clarify): are basin and grid cell in your model description inter-changable or how is a 'basin' defined in the model realm?**

We changed the subtitle to '2.1.2 Sediment and particulate organic carbon delivery from upland soil to river network' based on your suggestion.

The headwater basin and grid cell are not inter-changeable in our manuscript. The headwater basins are extracted from high-resolution DEM data, and there can be many headwater basins in each 0.5° grid cell of ORCHIDEE-Clateral (Fig. S2, see below). To simulate the sediment and POC deliveries from land soils to river networks, we first extracted the headwater basins and river networks from high-resolution DEM data (Fig. S2a,d). Then the MUSLE model (Eq. 1) was applied at each headwater basin to calculate the reference net soil loss rate (Fig. S2e) under a given set of reference runoff and vegetation cover conditions. By summing up the net soil loss

from all headwater basins in each grid-cell (Fig. S2e), we can calculate the total soil loss rate from land to river network under reference runoff and vegetation conditions (Eq. 3, Fig. S2f). Finally, the aggregated data of soil loss rate at 0.5° is used as a forcing file of ORCHIDEE-Clateral to calculate the actual daily soil loss rate from land to river (Eq. 4, Fig. 2g). The upscaling scheme and the method for extracting headwater basins and river networks has been described in details in Zhang et al. (2020). That is why we only give a brief overview in this manuscript.

Nonetheless, we have now revised the method section in this manuscript and added a supplementary figure (i.e. Fig. S2) to provide a better explanation of our method to simulate the sediment delivery from uplands to river channels. Please see "To give an accurate simulation of sediment delivery from uplands to river network and maintain computational efficiency, an upscaling scheme which integrates information from high-resolution (3″) topographic and soil erodibility data into a LSM forcing file at 0.5° spatial resolution, has been introduced (see details in Zhang et al., 2020, Fig.S2). With this upscaling scheme, the erosion-induced sediment and POC delivery from upland soils to the river network, as well as the changes in SOC profiles due to soil erosion had already been implemented in ORCHIDEE-MUSLE (Zhang et al., 2020). The sediment delivery from small headwater basins (which are basins without perennial stream and are extracted from high-resolution (e.g. 3″) digital elevation model (DEM) data, Figs. S2a&d) to the river network (i.e. gross upland soil erosion – sediment deposition within headwater basins) is simulated using the Modified Universal Soil Loss Equation model (MUSLE, Williams, 1975). As introduced in Zhang et al. (2020), "the daily sediment delivery rate from each headwater basin ($S_{i\_ref}$, Mg day$^{-1}$ basin$^{-1}$) is first calculated for a given set of reference runoff and vegetation cover conditions (Fig. S2e)" (lines: 216-227)

[Figure]

**Figure S2** Upscaling scheme used in ORCHIDEE-MUSLE (Zhang et al., 2020) for calculating the sediment delivery rate from headwater basins to river networks. MUSLE is the Modified Universal Soil Loss Equation; DEM is the digital elevation model (m); $K$ is the soil erodibility factor (Mg MJ-1 mm-1); $R_{ref}$ is the assumed reference daily runoff depth (= 10 mm day$^{-1}$); $R_{30\_ref}$ is the assumed reference maximum 30-minutes runoff depth (= 1 mm 30-minutes$^{-1}$); $C_{ref}$ (= 0.1, dimensionless) is the assumed reference cover management factor; $R_j$, $R_{30\_j}$ and $C_j$ are the simulated daily total runoff depth, daily maximum 30-minutes runoff depth and daily cover management factor on day $j$, respectively.

Zhang, H., Lauerwald, R., Regnier, P., Ciais, P., Yuan, W., Naipal, V., Guenet, B., Van Oost, K., and Camino-Serrano, M.: Simulating Erosion-Induced Soil and Carbon Delivery From Uplands to Rivers in a Global Land Surface Model. J. Adv. Model. Earth Syst., 12, e2020MS002121, 2020.

**16. l.230 check the units - also for a**

We have double-checked the units of all variables to make sure they are all correct.

**17. l.254 sentense ends abruptly - something is missing**

   **l.255 typesetting + space**

   **l.257 conditions to ...? something seems to be missing**

Sorry for the edit errors in this sentence. We have modified it. Please see "This aggregated dataset is then used to force the simulation of the actual daily sediment delivery ($S_j$, g day$^{-1}$ grid$^{-1}$) in ORCHIDEE-C$_{lateral}$, simply based on the estimated reference sediment delivery rates of Eq. (1) and on the ratios between actual runoff and land cover conditions and the assumed reference conditions used to create that forcing file (Eq. 4, Fig. S2g)." (lines: 254-257)

**18. Fig.1 since POCa;p;s appear later in the text, please add the explanation in the caption. Further, I suspect, it's a typo: POCc !POCs?**

We have added the explanation of POC$_a$, POC$_s$ and POC$_p$ in the caption of Fig. 1 and corrected the typo of POCs. Please see "POC$_a$, POC$_s$ and POC$_p$ are the active, slow and passive POC pool, respectively." (lines: 296-297)

**19. l.302f is the S$_{fast\_h2o}$ correctly placed? - I suspect, it should be after 'water reservoir', since the residence time is _fast or not?**

We changed the place of 'S$_{fast\_h2o}$' based on your comment. Please see "the sediment delivery ($F_{RO\_sed}$, g day$^{-1}$) from uplands in each basin (i.e. each 0.5° grid in the case of this study) initially feeds an aboveground water reservoir ($S_{fast\_h2o}$, m$^3$) with a so-called fast water residence time." (lines: 300-302)

**20. l.305 from the fast water reservoir into the stream reservoir**

 **l.308 τ is thus far undefined, if I am not mistaken**

Line 305-308: "Daily water ($F_{Fout\_h2o}$, m$^3$ day$^{-1}$) and sediment ($F_{Fout\_sed}$, g day$^{-1}$) flows from fast water reservoir to stream reservoir are calculated from a grid cell-specific topographic index $f_{topo}$ (unitless, Vörösmarty et al., 2000) extracted from a forcing file (Table 1) and a reservoir-specific factor $\tau$ which translates $f_{topo}$ into a water residence time of each reservoir (Eqs. 5, 6)." (lines: 303-307)

To our knowledge, 'to' is better than 'into' in this sentence. Thus we did not change 'to' to 'into'. For the definition of $\tau$, it is a reservoir-specific factor which translates $f_{topo}$ into a water residence time of each reservoir. We clarified this point in lines 306-307.

**21. l.311 enters the stream reservoir**

We revised this sentence based on your comment. Please see "we assumed that there is no sediment deposition in the fast reservoir, and that all of the sediment in the fast reservoir enters the stream reservoir." (lines: 309-311)

**22. l.313 transports**

We revised this sentence based on your comment. Please see "The belowground drainage ($F_{DR\_h2o}$, m$^3$ day$^{-1}$) only transports DOC and dissolved $CO_2$ to the stream reservoir (Fig. 1b)." (lines: 312-313)

**23. l.316f sediment in the stream . . . determined by $F_{out\ sed}$ . . . sediment input by flooding water**

**    l.318 maybe a matter of specialization of science communities, but is there a difference between resuspension (or erosion) and re-detachment of sediment? (also in Fig.1)**

We have revised this sentence based on your comment. Please see "The budget of the suspended sediment in the stream ($S_{riv\_sed}$, g) is determined by $F_{out\_sed}$, the upstream sediment input ($F_{up2riv\_sed}$, g day$^{-1}$), the sediment input by flooding water returning to the river ($F_{fld2riv\_sed}$, g day$^{-1}$),  the re-detachment of the previously deposited sediment in the river bed ($F_{rero\_sed}$, g day$^{-1}$), the bank erosion ($F_{bero\_sed}$, g day$^{-1}$), the sediment deposition in the river bed ($F_{rd\_sed}$, g day$^{-1}$) and the sediment transported to downstream river stretches ($F_{down2riv\_sed}$, g day$^{-1}$)" (lines: 316-321)

In our model, the resuspension, erosion and re-detachment of sediment are assumed to be the same. All of them describe the sediment flux from previously deposited sediment in river channel to the river stream.

**24. l.343 each grid, and we thus require/apply a different approach described in the following (or something similar - as a reader, I was a bit lost with this last sentence)**

We have revised this sentence to make it easier to be understood. Please see "Given the difficulty to simulate the detailed hydraulic dynamics of the stream flow at large spatial scale, we thus apply a simple approach described below to calculate the sediment transport capacity." (lines:833-835 )

**25. l.408 different particle sizes**

We revised this sentence based on your comment. Please see "Many studies described the selective transport of POC and sediment of different particle sizes." (line: 387)

**26. l.447_ Why did you chose the turnover time for the passive soil organic carbon content the same as the active one and not as the slow one? This also puzzles me a bit later on in the discussion, where you seem to provide good reasons for slower turnover times when soil passive organic carbon is released to rivers (see below).**

The decomposition and transformation of soil organic carbon in ORCHIDEE is simulated following the CENTURY model (Parton et al., 1988). In CENTURY, the passive pool ($SOC_p$) represents the physically protected fraction of SOC. When it is decomposed, 55% of the decayed $SOC_p$ is assumed to be released to the atmosphere in the form of $CO_2$, and all of the remaining fraction of the decayed SOCp feeds into the active SOC pool ($SOC_a$), while none of the decayed $SOC_p$ feeds into the slow pool ($SOC_s$). Following the scheme of carbon flows between different SOC pools in CENTURY, we thus assume that the passive POC which is originally protected by soil aggregates only feeds into the active POC pool when soil aggregates break down during the sediment transport process. Moreover, even if we assume that the turnover time of the passive POC in river flows is similar to that of the active POC (characterized by the shortest turnover time), the total decomposed POC during the transport process in river networks is only 2.9% (Fig. 7) of the total POC delivered from land to river. This means that uncertainties in simulated POC export to the sea caused by this assumption should be very limited. To demonstrate this further, we have conducted a simulation with the turnover time of the passive POC in river flows assumed to be identical to that of the passive SOC pool (characterized by the longest turnover time). In this case, 0.9% of the total riverine POC is decomposed during the transport process in river networks. Overall, we conclude that the decomposited fraction of riverine POC should range from 0.9% to 2.9%.

**27. l.456 I believe this is a bit misleadingly written. I suspect that you mean that the deposited sediment becomes part of the surface soil layer (and does not become a new layer - otherwise, one would end up with lots of layers)**

We have revised this sentence based on your comment. Please see "The sediment deposited on the floodplain becomes part of the surface soil layer, and the active, slow and passive POC flow into the active, slow and passive SOC pools in surface soil layer, respectively." (lines: 435-437)

**28. l.467 over Europe and parts of Middle East and Africa (. . . )**

We have revised this sentence based on your comment. Please see "In this study, ORCHIDEE-C$_{lateral}$ was applied over Europe and parts of Middle East (-30W– 70E, 34N-75N)," (lines: 446-447)

**29. l.470 is there any reason for 2 days (and not for one or more than 2)?**

To give a more accurate description, we have changed 'days' to 'events'. Please see "The return period of daily bankfull flow ($P_{flooding}$, year), which represents the average interval between two flooding events and is used in this study to produce the forcing file of $S_{rivmax}$ from a pre-run of ORCHILEAK." (lines: 448-450)

**30. l.473 all the flooding waters**

We have revised this sentence based on your comment. Please see "as the flooding may occur in several continuous days and all the flooding waters occurring on these continuous days are generally regarded to belong to the same flooding event (supplementary Fig. S3)." (lines: 451-453)

**31. l.476 First you make the point that there is substantial spatial variation of P$_{flooding}$ and then you assume it the same for Europe. Why and what are the consequences? - also for your later comparison, l. 506/507**

To our knowledge, observational data on $P_{flooding}$ are still very limited, and there is no data capable of providing spatially distributed $P_{flooding}$ for each half-degree grid cell. Therefore, following Schneider *et al*. (2011), we use a constant $P_{flooding}$ to simulate the bankfull flows from all European rivers. Both Schneider *et al*. (2011) and this study (Fig. 2b) indicate that the simulated bankfull flows compare reasonably well with observations even with a constant $P_{flooding}$. To give an explanation on why we use a constant $P_{flooding}$, we have deleted the original sentence "$P_{flooding}$ shows substantial spatial variations following climate and topography (Schneider *et al*., 2011).", and added the texts "To our knowledge, existing observational data on $P_{flooding}$ are still very limited. Therefore, following Schneider *et al*. (2011), we also use a constant $P_{flooding}$ to simulate the bankfull flows from European rivers and the observed long-term (1961–2000) average bank full flow rate (m$^3$ s$^{-1}$) at 66 sites obtained from Schneider *et al*. (2011) was used to calibrate $P_{flooding}$ (= 0.1 year, Table 2)." (lines: 453-457)

**32. l.478 Following Zhang et al.**

We have revised this sentence based on your comment. Please see "Following Zhang et al. (2020), the parameters *a*, *b*, *c* and *d* in Eq. 1 and 2 (Table 2) were calibrated at 57 European catchments (Fig. S2d) against the modelled sediment delivery data obtained from the European Soil Data Centre (ESDAC, Borrelli et al., 2018)." (lines: 457-460)

**33. l.493 I suspect you want to refer to Eq. 10 (not 5)?**

Sorry for the mistake. The original Eq. 5 has been changed to Eq. 10.

**34. l.503 as forcing data**

We have revised this sentence based on your comment. Please see "In the 'spin-up' simulation, the PFT maps, atmospheric $CO_2$ concentrations and meteorological data during 1901–1910 were used repeatedly as forcing data." (lines: 486-487)

**35. l.507 (locations see Fig...)**

We have revised this sentence based on your comment. Please see "The finally simulated water discharge rates in European rivers were evaluated using observation data at 93 gauging sites (locations see Fig. S4a) from the Global Runoff Data Base (GRDC, Table 1)." (lines: 488-490)

**36. l.516f if I interpret your caption of Fig. 4 correctly (and to my knowledge), you can derive POC content by TOC-DOC=POC, which should result in a much larger data base you can compare your model simulations to, right? See my main comments.**

Please see our response to your comment #2.

**37. l.607_ As mentioned in the main comments part, I would like to see the temporal variability of the simulation in contrast to observations more in-depth discussed (also for POC). I believe understanding this model behavior better could provide important insights.**

Please see our response to your comment #4.

l.613 Note that (typo)

We have corrected this typo based on your comment. Please see "Note that the mean and standard deviation of the simulated concentrations at each site are calculated based on the monthly average value" (lines: 600-601)

**38. l.625-634 this potentially requires re-working, if TOC-DOC=POC**

Please see our response to your comment #2.

**39. Fig.7 I might misinterpret here something, but isn't upland loss = delivery to river? If it is, then I cannot relate the values for DOC and POC loss in Fig.7 to the DOC and POC delivery shown in Fig. S11b of the supplementary material (where POC≈18 Tg C yr⁻¹ and DOC≈ 7 Tg Cyr⁻¹). Otherwise, some explanations would be helpful to understand the discrepancies.**

Sorry for having not updated the Fig. S15 (i.e. the Fig. S11 in the original SI) based on our final simulation results. Now we have replaced the old figure with the latest one. Please see:

[Figure]

**Figure S15** Simulated time series of annual total sediment delivery from upland to river network (a), DOC and POC delivery from land to river network (b), vegetation net primary production

(NPP, c), heterotrophic respiration (Rh, d), respiration due to disturbances like harvest and land cover change (Rd, e), changes in living biomass (f), changes in litter carbon stock (g) and changes in SOC stock (h) in whole Europe from the year 1901 to 2014.

**40. l.682 Although the . . . (no 'but')**

    **l.684 rivers are much smaller**

We have revised this sentence based on your comment. Please see "Although the sediment discharge rates in some rivers in the Middle East can be as high as that in the Danube or Volga river (Fig. 8c), the POC delivery rates in these rivers are much smaller than in the larger ones (Fig. 9c)." (lines: 666-668)

**41. l.723 NEP is undefined, or do you mean NPP? (also in the next line)**

We have provided the definition of NEP in the revised manuscript. Please see "which is about 4.7% of the net ecosystem production (NEP ($993\pm255$ Tg C yr$^{-1}$), defined as the difference between the vegetation primary production (NPP) and the soil heterotrophic respiration (Rh) due to the decomposition of litter and soil organic matter (i.e. NEP=NPP–Rh))" (lines: 702-705)

**42. l.759 becomes part of the surface soil layer (see comment above)**

We revised this sentence based on your comment. Please see "In the floodplains, the newly deposited sediment becomes part of the surface soil layer" (lines: 741-742)

**43. l.771_ As written above, I am a bit puzzled about the choice of the turnover time of passive POC in rivers. If I am getting you right, you try to justify the parameters value choice with the decomposed fraction of the overall POC pool. Since you unfortunately never show the fractions of fast, slow and passive POC (would be nice to find such plot in the supplementary material), I wonder, which of the POC contributes most to the overall decomposition loss. How relevant is the choice of this value for your results?**

Please see our response to your comment #26 for the reason why we assume the turnover time of passive POC in rivers to be the same as that of active POC. Accurately quantifying the contributions of fast, slow and passive POC to the overall POC decomposition rate in the river network is difficult, because these three POC pools can be transformed to each other during the

decomposition process (Fig. 1). Nonetheless, based on the fractions of fast (11%), slow (21%) and passive (67%) POC in the total POC delivery from land to rivers (Fig. R1), and the turnover time of each POC pool, we can still roughly estimate the relative contributions of the three POC pools to the overall decomposition loss. If the turnover time of passive POC is assumed to be same to that of active POC (0.3 year) as performed here, the passive POC then contributes most to the overall decomposition loss. However, had we assumed that the turnover time of passive POC is equal to that of passive SOC (462 years), the overall decomposition loss would have been largely dominated by the two other pools.

[Figure]

**Figure R1.** Fractions of the active ($POC_a$), slow ($POC_s$) and passive ($POC_p$) POC in the total POC delivery from land to river networks from 1901 to 2014. Bars and whiskers in figure (b) represent the mean and standard deviation of the fraction, respectively.

**44. l.787 that ignoring lateral**

We have revised this sentence based on your comment. Please see "Comparison between the simulation results from ORCHIDEE-C$_{lateral}$ with activated and deactivated erosion and river routing modules indicate that ignoring lateral carbon transport processes in LSM may lead to significant biases in the simulated land carbon budget (Figs. 10 and S15)." (lines: 771-774)

**45. l.791 here NBP is differently defined than before, if NEP was not a typo (see before)**

In our study the net ecosystem production (NEP) is defined as the difference between the vegetation primary production (NPP) and the soil heterotrophic respiration (Rh) due to the decomposition of litter and soil organic matter (i.e. NEP=NPP–Rh)), and the net biome production (NBP) is defined as the difference between NEP and the land carbon loss (Rd) due to

the additional disturbances (e.g. harvest, land cover change, and soil erosion and leaching, i.e. NBP=NEP–Rd–DOC and POC to river). Therefore, it is correct to calculate NBP as NPP–Rh–Rd–DOC and POC to river in the original line 791. Nonetheless, to avoid readers misunderstanding the definition of NBP, we have changed the original 'NPP–Rh–Rd–DOC and POC to river' to 'NEP–Rd–DOC and POC to river' in this sentence (lines: 774-777).

**46. l.820 Estimations by Maavara**

We have revised this sentence based on your suggestion. Please see "Estimation by Maavara et al. (2017) suggests that the …" (line: 875)

**47. l.831 cancel out one of the 'e.g.'**

Sorry for the mistake. We have deleted one of the 'e.g.'

**48. l.843+846 I am a bit uncertain, if evasion is the right word here, maybe better 'effux' (or net outgassing)?**

Thanks for your comment. Following many previous studies (e.g. Lauerwald et al., 2015; Duvert et al., 2018; Horgby et al., 2019), we use $CO_2$ evasion to describe the release of $CO_2$ from water bodies to the atmosphere.

Lauerwald, R., Laruelle, G., Hartmann, J., Ciais, P., and Regnier, P.: Spatial patterns in $CO_2$ evasion from the global river network: Spatial patter of riverine $pCO_2$ and $FCO_2$. Global Biogeochem. Cycles, 29, 2015.

Duvert, C., Butman D.E., Marx, A., Ribolzi O., Hutley, L.B. $CO_2$ evasion along streams driven by groundwater inputs and geomorphic controls. *Nature Geoscience*, 11, 813-818, 2018.

Horgby, A. et al. Unexpected large evasion fluxes of carbon dioxide from turbulent streams draining the world's mountains. *Nature Communications*, 10, 4888, https://doi.org/10.1038/s41467-019-12905-z

**49. l.873 do you mean: Even though there are still . . .**

We have changed this sentence based on the comments from you and other referees. Please see "Even though our model can overall capture the lateral transfers of sediment and carbon in many

rivers in central and northern Europe, more observation data are crucially needed to further evaluate the performance of our model, in particular in southern Europe." (lines: 922-925)

**50. l.877 contrasting regions (?)**

**l.878 to predict**

We have changed this sentence based on the referees' comments. Please see "Thus, in the future, an important aim will be to further calibrate our model against more detailed observation data (e.g. sediment deposition rate in river channels and floodplains) and extend the model application to regions of contrasting climate, vegetation and topography." (lines: 929-932)

**51. l.879_ I am a bit puzzled with regards to the models versus the observations variability at this point. Usually, I would expect that the instantaneous observations show larger scatter than e.g. monthly averaged model simulations. In your simulations, the model range for DOC and TOC is often even beyond the mean envelope range of observations (Fig. S7 and S8). While I understand that the model cannot capture the observational instantaneous values, I would at least expect/tune it to capture annual (or seasonal) to decadal mean values and the envelope (or are there any known biases in the data base with this respect?). Hence, I am not sure, where the origin of this mismatch is coming from and if I can follow that the problem arises only from too little river observations (while I agree that more - long term- observations would help in other aspects like seeing trends, etc.).While not being an expert in LSM modeling, I feel that the model lacks some (potentially important) sort of buffering or counteracting mechanism that modulates down the DOC amplitudes (and other processes that increase the POC variability...). As pointed out above, I believe it would be of value to investigate this deeper and to discuss potential reasons for this model behavior.**

We have investigated and discussed the potential reasons for the overestimation of the seasonal dynamics of DOC and TOC. Please see our response to your comment #4 for details.

**52. l.900 NEP?**

Yes, it is NEP here.

**53. Supplementary material**

**l.11 description of ω sediment transport capacity (typo)**

Thanks. We have corrected the typo.

**54. l.17 shown**

   **l.18 investigated time of two years**

We have revised this sentence based on your comment. Please see "$P_{flooding}$ shown in this figure is 0.1 year as the bankfull flow occurred in 20 days during the investigated time of two years." (see caption of Fig. S3)

**55. l.23 used in this study? (something is missing here)**

Thanks for your comment. We have revised this sentence. Please see "**Figure S4** Geographical location of the gauging stations for river discharge (a), bankfull flow (b), sediment discharge (c) and riverine organic carbon discharge (d) used in this study." (see caption of Fig. S4)

**56. l.25 (. . . ) of these 57**

We have revised this sentence based on your suggestion. Please see "The simulated average net soil loss rates (g m$^{-2}$ yr$^{-1}$) of these 57 catchments were compared to the average net soil loss rates extracted from the sediment delivery data provided by the ESDAC (see section 2.3 of the main text)." (see caption of Fig. S4)

**57. Fig. S3 no response of DOC and CO2 to ω, $c_{ebank}$ and $e_{ebed}$? - maybe worth to note this explicitly in the caption (or wasn't it analyzed for these parameters?)**

We analyzed the response of DOC and $CO_2$ to parameters ω, $c_{ebank}$ and $e_{ebed}$. However, as these parameters mainly control sediment erosion and deposition, only sediment and POC show large response to changes in these parameters, and the responses of DOC and $CO_2$ are very limited. With a 10% changes in these parameters, the relative changes in DOC and $CO_2$ are generally less than 0.1%.

**58. Fig. S4 provide offsets and potentially standard deviations. You mention the offset for the Danube river delta in the text, but are there similar explanations for the Elbe (d) and particularly the Rhine (the latter seem even to be underestimated)? I would suggest to extend this discussion part in the main text (l.528_)**

Thanks for your suggestion. Stream channel bifurcation is one of the important reasons for explaining the difference between simulated river discharges and the observed discharge from

the main river channel. In the Danube river delta, a fraction of the discharge is actually exported to the sea through the Saint George Branch, in addition to the water discharge through the main river channel (Fig. S5b). This explains why the simulated water discharge rate at the outlet of the Danube catchment is larger than the observation at the Ceatal gauging station, Romania, where only the main stream discharge is measured. However, analysis of satellite images of the Rhine and Elbe catchments do not reveal obvious branching channels bypassing the hydrological gauging sites included in our study. In these latter two catchments, the systematic biases in simulated river discharge should thus be due to other factors, and in particular the discrepancy between the stream routing scheme (delineation of catchment boundaries) extracted from the forcing data at 0.5° resolution and the real river network (Fig. S5). More precisely, we found that the area of the Rhine catchment defined by the forcing data at 0.5° resolution is larger than the area in reality, while the area of the Elbe catchment defined by the forcing data is smaller than the real area.

Following your suggestion, we have extended the discussion to give a more detailed explanation regarding the biases in simulated river discharges caused by the uncertainties in the forcing data of the stream routing scheme. Please see "We recognize that ORCHIDEE-C$_{lateral}$ may overestimate or underestimate the water discharge rate in some rivers (Fig. 2a), particularly in smaller rivers where discrepancy between the stream routing scheme (delineation of catchment boundaries) extracted from the forcing data at 0.5° resolution and the real river network (Fig. S5) can be substantial. An overestimation or underestimation of the catchment area by the forcing data as respectively found for the Elbe and Rhine will introduce a proportional bias in the average amount of simulated discharge from these catchment. Another problem are stream channel bifurcations which occur in reality, but which are not represented in a stream network derived from a digital elevation model. For example, in the Danube river delta, a fraction of the discharge is actually exported to the sea through the Saint George Branch, in addition to the water discharge through the main river channel (Fig. S5b). This explains why the simulated water discharge rate at the outlet of the Danube catchment is larger than the observation at the Ceatal gauging station, Romania (identify number in the GRDC database is 6742900, Fig. S4m), where only the main stream discharge was measured." (lines: 515-527)

**59. Fig. S11 see comment in the main text on budgets of POC and DOC**

Please see our response to your comment #39.

---

## Author Comment (AC2)

**Referee #2**

**1. The authors present modelling results of lateral OC transfers based on an improved representation of POC and DOC transfer using the ORCHIDEE LSM model. Model outputs cover European catchments during the period from 1901 to 2014. The authors provide very interesting results on various OC budget component for the given period, which have not been published at the European scale (e.g. how much sediment/POC is stored in floodplains and exported to oceans). Furthermore, the authors compare the C budget with and without lateral C transfer and provide interesting information on the discussion of the net effect of lateral sediment transfer of the carbon budget. Overall, the study is of great interest and certainly the develop model provides a significant step forward on that topic (as far as I can judge this from a non-modeler point of view). I recommend this manuscript to be published in ESDyn after major revisions. Please consider the general and more detailed comments below to improve the submitted manuscript.**

**Kind regards**

**Thomas Hoffmann**

Thanks a lot for your positive comments, as well as your suggestions and corrections below. Your comments are very helpful to improve our manuscript. We have carefully studied them and revised our manuscript accordingly. Please see our responses to your specific comments below.

**2. General comments**

**It is very hard to understand the model setup as a non-model expert and the text is not very communicative to motivate and convince non-model experts to read the paper. I suggest that the authors should critically prove how they can motivate the linkages between the empirical evidences and the model setup. I had hard times to understand the general model approach and still do not fully understand how the authors define headwaters and whether they differentiate headwater sizes depending on the climate /lithology etc. But maybe I was**

**blind and unable to extract the relevant information. In any case, many assumptions/ approaches are not straight forwards and description should be improved.**

Sorry for having not provided a detailed description on our model setup, especially regarding the scheme for simulating sediment and POC delivery from headwater basins to river networks, as this scheme has already been explained in detail in our previous paper by Zhang et al. (2020).

Our simulations are based on the well-established, simple but efficient MUSLE model. Note that this model is usually applied to small headwater basins, and high resolution geospatial data is needed to delineate single headwater basins and to parametrize the model. In order to use that approach in a land surface model (LSM) which runs at a coarse spatial resolution of 0.5 degree (~50km), we applied the following strategy: To simulate the sediment and POC deliveries from land soils to river networks, we first extracted the headwater basins and river networks from high-resolution DEM data (3″ (~ 90 m), Fig. S2a,d). Then the MUSLE model (Eq. 1) was applied to each headwater basin, parametrizing the model with information on topography and soil erodibility derived from similarly highly resolved geodata. For parameters related to vegetation cover, runoff and land management, which are variable over time, we applied freely chosen values which we applied as reference values to all headwater basins. The so obtained estimates of "reference net-soil erosion rates" were then aggregated to the 0.5 degree resolution used for our simulations with ORCHIDEE-Clateral  (Fig. S2e). For the simulations of daily net-soil erosion in ORCHIDEE-Clateral, we took advantage of the fact that according to the MUSLE equation, changes in predicted net-soil erosion rates scale to changes in runoff and vegetation cover (Eq. 3, Fig. S2f).  To that end, we use the rasterized predicted reference erosion rates as a forcing file for ORCHIDEE-Clateral, and use ratios between daily runoff and vegetation cover values simulated with that model and the reference values used to produce the forcing file to estimate the actual daily soil loss rate from land to river in each 0.5 degree grid cell of the model grid (Eq. 4, Fig. 2g). This upscaling scheme and the method for extracting headwater basins and river networks is described in detail in Zhang et al. (2020) , and for that reason we only provide a brief overview of the methodology in this manuscript.

We have thus revised the method section of this manuscript and added a supplementary figure (Fig. S2) to provide a more detailed explanation of the method for simulating sediment delivery from uplands to river channels. Please see "To give an accurate simulation of sediment delivery

from uplands to river network and maintain computational efficiency, an upscaling scheme which integrates information from high-resolution (3″) topographic and soil erodibility data into a LSM forcing file at 0.5° spatial resolution, has been introduced (see details in Zhang et al., 2020, Fig.S2). With this upscaling scheme, the erosion-induced sediment and POC delivery from upland soils to the river network, as well as the changes in SOC profiles due to soil erosion had already been implemented in ORCHIDEE-MUSLE (Zhang et al., 2020). The sediment delivery from small headwater basins (which are basins without perennial stream and are extracted from high-resolution (e.g. 3″) digital elevation model (DEM) data, Figs. S2a&d) to the river network (i.e. gross upland soil erosion – sediment deposition within headwater basins) is simulated using the Modified Universal Soil Loss Equation model (MUSLE, Williams, 1975). As introduced in Zhang et al. (2020), "the daily sediment delivery rate from each headwater basin ($S_{i\_ref}$, Mg day$^{-1}$ basin$^{-1}$) is first calculated for a given set of reference runoff and vegetation cover conditions (Fig. S2e)" (lines: 216-227)

[Figure]

**Figure S2** Upscaling scheme used in ORCHIDEE-MUSLE (Zhang et al., 2020) for calculating the sediment delivery rate from headwater basins to river networks. MUSLE is the Modified Universal Soil Loss Equation; DEM is the digital elevation model (m); $K$ is the soil erodibility factor (Mg MJ-1 mm-1); $R_{ref}$ is the assumed reference daily runoff depth (= 10 mm day$^{-1}$); $R_{30\_ref}$ is the assumed reference maximum 30-minutes runoff depth (= 1 mm 30-minutes$^{-1}$); $C_{ref}$ (= 0.1, dimensionless) is the assumed reference cover management factor; $R_{iday}$, $R_{30\_iday}$ and $C_{iday}$ are the simulated daily total runoff depth, daily maximum 30-minutes runoff depth and daily cover management factor, respectively.

Zhang, H., Lauerwald, R., Regnier, P., Ciais, P., Yuan, W., Naipal, V., Guenet, B., Van Oost, K., and Camino-Serrano, M.: Simulating Erosion-Induced Soil and Carbon Delivery From Uplands to Rivers in a Global Land Surface Model. J. Adv. Model. Earth Syst., 12, e2020MS002121, 2020.

**3. The general concept, as depicted in Figure 1, draw an arrow (F_up2fld) from the upstream basin to the floodplain. I was wondering if this transport path is needed? The main transfer to floodplains steams from the main river channel that is passing the floodplain. The potential inputs from upstream basins is of academic nature and could be neglected.**

The scheme used to simulate flooding flows in ORCHIDEE-Claeral is identical to that used in the ORCHILEAK model (Lauerwald et al., 2017). In the routing scheme of the model, all water coming from upstream lying cells of the modelling grid enter the mainstream of the river in a particular cell. If these inflows exceed the predefined bankfull flow, the excess water is flowing into the floodplains instead of into the river mainstream, while only the amount of water that equals the bankfull flow enters the mainstream directly. In a way, this model set-up represents the idea that most of the water and sediments entering the floodplains is coming from the mainstream. We realize that the name of the flux "$F_{up2fld}$" might have been a bit misleading here. Indeed, no water coming from the hinterland of the floodplains is actually entering these floodplains. We have changed the original "$F_{up2fld}$" to "$F_{riv2fld}$" in the revised MS to avoid this confusion. Please see:

[Figure]

[Figure]

**Figure 1** Simulated lateral transfer processes of water, sediment and carbon (POC, DOC and $CO_2$) in ORCHIDEE-C$_{lateral}$ (a) and a schematic plot for the reservoirs and flows of water, sediment and carbon represented in the routing module of ORCHIDEE-C$_{lateral}$ (b). S$_{soil}$ is the soil pool. S$_{rivbed}$ is the sediment (also POC) deposited in river bed. S$_{fast}$, S$_{slow}$, S$_{riv}$ and S$_{fld}$ are the 'fast', 'slow', stream and flooding water reservoir, respectively. F$_{RO}$ and F$_{DR}$ are the surface runoff and belowground drainage, respectively. F$_{Fout}$ and F$_{Sout}$ are the flows from fast and slow reservoir to the stream reservoir, respectively. F$_{up2riv}$ and F$_{down2riv}$ are the upstream inputs and downstream outputs, respectively. F$_{riv2fld}$ is the outputs from river stream to the flooding reservoir. F$_{fld2riv}$ is the return flow from flooding reservoir to stream reservoir. F$_{bed2fld}$ is the transform from deposited sediment in river bed to floodplain soil. F$_{bero}$ is bank erosion. F$_{rd}$ and F$_{rero}$ are the deposition and re-detachment of sediment and POC in river channel, respectively.

$F_{sub}$ is the flux of DOC and CO2 from floodplain soil (originated from the decomposition of submerged litter and soil carbon) to the overlying flooding water. $F_{fd}$ is the deposition of sediment and POC and the infiltration of water and DOC. $F_D$ is the wet and dry deposition of DOC from atmosphere and plant canopy. $DOC_l$ and $DOC_r$ are the labile and refractory DOC pool, respectively. $POC_a$, $POC_s$ and $POC_p$ are the active, slow and passive POC pool, respectively.

**4. Major parts of Europe are missing in terms of observations of suspended sediment loads (e.g. no observation for Spain, France GB, Italy), esp. Mediterranean rivers are not covered in this study. Due to the very different behavior of streams in south and north Europe, the study is strongly biased toward N-Europe. This becomes even more important by the comparison with the model output from WATEM/SEDEM for the two catchments of the Apennine Peninsula, which are simply ignored as 'outliers'.**

Although the simulated sediment loss rates in the two catchments of the Apennine Peninsula is significantly lower than the estimates from WATEM/SEDEM, the simulated sediment loss rates in many catchments in southern and western Europe (e.g. France and Spain) are overall comparable to the estimates from WATEM/SEDEM. Moreover, the average sediment loss rate over all catchments is 40.8 g m$^{-2}$ yr$^{-1}$, which is overall comparable to the estimate by the WaTEM/SEDEM (42.5 g m$^{-2}$ yr$^{-1}$) (this result has been added to the revised manuscript, see lines: 545-546). Given this, our simulation result on sediment delivery rate from upland soils to river networks are, we believe, overall acceptable. Nonetheless, we recognize that further calibration of model using data from southern and western Europe would be very useful to decrease the uncertainties in our model andwe have stressed this aspect in our revised ms. Please see "Further model calibration, evaluation and development is necessary for improving our model. Due to the limitation of observation data, we calibrated the parameters controlling sediment transport, deposition and re-detachment (i.e. $\omega$, $c_{rivdep}$, $c_{flddep}$, $c_{ebed}$ and $c_{ebank}$ in Table S1) in stream and flooding reservoirs only against the observed sediment yield. Even though our model can overall capture the lateral transfers of sediment and carbon in many rivers in central and northern Europe, more observation data are crucially needed to further evaluate the performance of our model, in particular in southern Europe. In addition, it is still unknown whether our model can satisfactorily simulate intermediate processes such as sediment

deposition in river channels and floodplains, as well as the rate of river channel erosion. It is also unknown whether our model would perform satisfactorily in regions with very different climates than Europe such as the tropical region. Thus, in the future, an important aim will be to further calibrate our model against more detailed observation data (e.g. sediment deposition rate in river channels and floodplains) and extend the model application to regions of contrasting climate, vegetation and topography." (lines: 919-932)

**5. The authors provide model runs with and without lateral C transport and find that SOC stock only marginally increase of lateral flux is turned on. This very low increase somehow contradicts the large amount of POC retention in floodplains. The authors somehow provide numbers with SOC stock decrease in mountains and SOC stock increases in floodplains, but does this mean that SOC retention in floodplains is more or less fully compensated by soil degradation at eroding sites? It was argued that at long-term ($10^3$a) OC retention in floodplains is more important than soil degradation, while at shorter terms (couple of years) degradation effect might dominate. I wonder what happens to the model if the authors considerer shorter and much longer time scales than those used in this study. Please discuss this in more detail.**

Thanks for your thoughtful comment and suggestion. From 1901 to 2014, integrated erosion and leaching over Europe induced a loss of 3.03 Pg organic carbon (POC+DOC) from uplands to the river network, and only 0.65 Pg of this carbon flowed back to soils with flooding waters. The total stock of soil organic carbon in Europe thus should have decreased by 2.38 Pg C. However, due to the decrease in decomposition rate of the buried organic carbon (including in-situ and ex-situ carbon) in floodplain soils, the total stock of soil organic carbon in Europe only decreased by 0.91 Pg C. Floodplains in Europe have overall prevented 2.12 (= 3.03 - 0.91) Pg soil organic carbon from being transported to the sea or released back to the atmosphere in the form of $CO_2$. Although the sequestration of organic carbon in floodplains cannot make up for all of the soil organic carbon loss, the increased organic carbon stock in floodplains (2.12 Pg C) is much higher than the sole soil POC loss (0.86 Pg C) induced by soil erosion. We have added one paragraph to discuss this. Please see:

"Consistent with previous studies (Stallard, 1998; Smith et al., 2001; Hoffmann et al., 2013), our simulation results showed the importance of sediment deposition in floodplains with regard to the overall SOC budget. From 1901 to 2014, erosion and leaching over Europe totally induced a loss of 3.03 Pg organic carbon (POC+DOC) from uplands to the river network, and only 0.65 Pg of this carbon was redeposited onto the floodplains. The total stock of soil organic carbon in Europe thus should have decreased by 2.38 Pg C. However, due to the decrease in decomposition rate of the buried organic carbon (including in-situ and ex-situ carbon) in floodplain soils, the total stock of soil organic carbon in Europe only decreased by 0.91 Pg C. Floodplains in Europe have totally protected 2.12 (= 3.03 - 0.91) Pg soil organic carbon from been transported to the sea or be released to the atmosphere in forms of $CO_2$. Although the sequestration of organic carbon in floodplains cannot make up all of the soil organic carbon (POC+DOC) loss, the increased organic carbon stock in floodplains (2.12 Pg C) is much higher than the soil POC loss (0.86 Pg C) induced by soil erosion." (lines: 789-801)

Moreover, we found that the effect of carbon retention in floodplains on the soil carbon budget is highly variable across regions. For example, in northern Europe where the soil is protected by densely growing forests, DOC leached from soil is the main source of riverine carbon, and soil and POC erosion rate is very low. Floodplains in northern Europe thus cannot store a large amount of organic carbon and the lateral carbon transfer in northern Europe generally results in a significant decrease in total soil organic carbon stock. However, in southern Europe, the soil erosion rates in uplands and sediment deposition rates in floodplains are both high. Floodplains in southern Europe are thus simulated to preserve a large amount of organic carbon. Accordingly, the lateral carbon transfer does not induce a strong decrease in the total soil carbon stock in southern Europe. In some catchments with large areas of floodplains and high sediment deposition, the lateral carbon transfer can even result in an increase in the total soil carbon stock.

In this manuscript, the main aims were to present the model developments and the comparison of model results against (limited) observational data and only cover a few key striking features of the lateral sediment and carbon transfers. Our subsequent, ongoing work now targets a model application study that explores in detail the spatiotemporal variation of the lateral carbon transfers over Europe during the period 1901-2014, as well as the impacts of the different processes of lateral carbon transfer on the regional terrestrial carbon budget. Therefore, we do not explore the effects of sediment and carbon deposition on the soil carbon budget for different

time scales here, as this is beyond the scope of the present study. We agree that extending the simulations to the millennial timescale to explore the long-term effects of lateral carbon transfer on the land carbon budget is a great idea, and we will consider such simulations in the future.

**6. Detailed comments**

**Line 38: 'but also leaching of DOC' → needs some more details, leaching from where?**

We added 'soil' before DOC to more accurately describe the source of DOC. Please see "Erosion of soils and the associated organic carbon, but also leaching of soil dissolved organic carbon (DOC), represent a non-negligible leak in the terrestrial carbon budget and a substantial source of allochthonous organic carbon to inland waters and oceans" (lines: 37-40)

**7. Line 45: I suggest to refer to a new review on OC sequestration in floodplains (https://doi.org/10.1016/B978-0-12-818234-5.00069-9)**

Thanks for your suggestion. We have added this new reference in our manuscript. Please see "Meanwhile, the organic carbon that is redeposited and buried in floodplains and lakes might be preserved for a long time, thus creating a $CO_2$ sink (Stallard, 1998; Van Oost et al., 2007; Wang et al., 2010; Hoffmann, 2022)." (lines: 43-46)

**8. Line 169ff: the many branches/modules etc. make it very hard to understand the model setup. Could you somehow visualize it?**

Thanks to your suggestion. We have added a supplementary figure to visualize the difference between different branches of the ORCHIDEE model, as well as the developing history of these branches. Please see:

[Figure]

**Figure S1.** Properties and the developing history of the ORCHIDEE branches mentioned in this study.

**9. Line 228: I suggest to avoid sediment delivery rate, as this might be confused with sediment delivery ratio. Please use sediment supply instead. How are headwater basins defined in this study?**

The sediment delivery rate is very different from sediment delivery ratio, and we have provided the unit of sediment delivery rate to make the readers aware of that difference. In order to be consistent with our previous studies (e.g. Zhang et al., 2020), we prefer to keep using sediment delivery rate. For the explanation of headwater basins, please see our response to your comment #2.

**10. Line 229: given set of runoff and vegetation cover conditions → could you specify them and motivate, why you refer to the reference runoff condition rather than actual runoff. This might be explained in the original reference. However, I highly recommend to give more details here, to ease the understanding of the approach.**

The assumed runoff and vegetation cover conditions are used for our upscaling scheme. We introduce this upscaling scheme because it integrates high-resolution (3″) topographic and soil erodibility data into our simulation of sediment delivery from uplands to river channels at 0.5° spatial resolution while maintaining the computational efficiency of our model. As this upscaling scheme has been explained in detail in our previous study (Zhang et al., 2020), we only provide a brief description in this study. Nonetheless, we have revised our manuscript and added a supplementary figure (Fig. S2) to explain our method in more details. Please see our response to your comment #2 for more details.

**11. Line 232: Is this really a runoff or rather a precipitation reference (given the 10mm day-1)?**

Yes, it is runoff with a unit of mm day$^{-1}$, that means it is a flux rate normalized by area. In ORCHIDEE, the runoff is calculated as: runoff = precipitation – canopy interception – evaporation – infiltration. The unit of all these variables is mm day$^{-1}$. We use water discharge (m$^3$ day$^{-1}$) to describe the water flow in river channels, and the amount of water discharge from

land surface to river channel in each grid cell can be calculated as runoff×$A_{grid}$×$10^{-3}$ ($A_{grid}$ is the area of the target grid cell with a unit of m$^2$).

**12. Line 237: DA^(dDA^c) → is that correct? Looks erroneous**

Yes, it is correct. It represents the nonlinear impacts of drainage area to the peak flow rate at the outlet of a water basin.

**13. Line 237: is assume that DA is drainage area (not defined)**

We have added the definition of DA. Please see "$q_{i\_ref}$ was calculated from the reference maximum 30-minutes runoff (= 1 mm 30-minutes$^{-1}$) depth and drainage area ($DA_i$, m$^2$) according to the following equation" (lines: 233-235)

**14. Line 238: same as above; (p.5-6) should be linked once more to the reference where this citation is taken from.**

We have changed the position of (p. 5-6) from the end of the quoted contents to the place of the reference. Please see:

"As introduced in Zhang et al. (2020, p. 5-6), "the daily sediment delivery rate from each headwater basin ($S_{i\_ref}$, Mg day$^{-1}$ basin$^{-1}$) is first calculated for a given set of reference runoff and vegetation cover conditions (Fig. S2e):

$$S_{i_{ref}} = a \left( Q_{i_{ref}} q_{i_{ref}} \right)^b K_i L S_i C_{ref} P_{ref} \tag{1}$$

where $Q_{i\_ref}$ is the total water discharge (m$^3$ day$^{-1}$) at the outlet of headwater basin $i$ for the daily reference runoff condition ($R_{ref}$) of 10 mm day$^{-1}$. In Eq. 1, $q_{i\_ref}$ is the daily peak flow rate (m$^3$ s$^{-1}$) at the headwater basin outlet under the assumed reference runoff condition. Similar to the SWAT model (Soil and Water Assessment Tool, Neitsch et al., 2011), $q_{i\_ref}$ was calculated from the reference maximum 30-minutes runoff (= 1 mm 30-minutes$^{-1}$) depth and drainage area ($DA_i$, m$^2$) according to the following equation:

$$q_{i\_ref} = \frac{R_{30\_ref}}{30 \times 60} \left( DA_i{}^{(d\ DA_i{}^c)} \right) 1000 \tag{2}$$

where $R_{30\_ref}$ (= 1 mm 30-minutes$^{-1}$) is the assumed daily maximum 30-minutes runoff"." (lines: 225-237)

**15. Line 247-249: not sure what the authors want to say here? I guess the model outputs should depend on these parameters.**

In our upscaling scheme, the reference runoff ($R_{ref}$, $R_{30\_ref}$) and vegetation cover ($C_{ref}$) conditions are only the intermediary for improving the computational efficiency of the model and reduce the usage of computer memory and storage space (these are important for global land surface modeling). They did not result in any change in the simulated result of sediment delivery. Below we provide the detailed process of mathematical transformation to explain why the assumed values of $R_{ref}$, $R_{30\_ref}$ and $C_{ref}$ have no impact on the model output:

$$S_j = \sum_{i=1}^{n} (S_{i\_j}) \times 10^6$$

$$= \sum_{i=1}^{n} \left( a(Q_j \, q_j)^b K_i LS_i C_j \right) \times 10^6$$

$$= \sum_{i=1}^{n} \left( a \left( (R_j \, DA_i) \times \frac{R_{30\_j}}{30 \times 60} \left( DA_i^{(d \, DA_i^C)} \right) 1000 \right)^b K_i LS_i C_j \right) \times 10^6$$

$$= (R_j \, R_{30\_j})^b C_j \sum_{i=1}^{n} \left( a \left( DA_i \times \frac{1}{30 \times 60} \left( DA_i^{(d \, DA_i^C)} \right) 1000 \right)^b K_i LS_i \right) \times 10^6$$

$$= \frac{(R_j \, R_{30\_j})^b C_j}{(R_{ref} \, R_{30\_ref})^b C_{ref}} \sum_{i=1}^{n} \left( a \left( (R_{ref} \, DA_i) \right. \right.$$

$$\left. \left. \times \frac{R_{30\_ref}}{30 \times 60} \left( DA_i^{(d \, DA_i^C)} \right) 1000 \right)^b K_i LS_i C_{ref} \right) \times 10^6$$

$$= \frac{(R_j \, R_{30\_j})^b C_j}{(R_{ref} \, R_{30\_ref})^b C_{ref}} \sum_{i=1}^{n} (S_{i\_ref}) \times 10^6$$

$$= \left( \frac{R_j \, R_{30\_j}}{R_{ref} \, R_{30\_ref}} \right)^b \frac{C_j}{C_{ref}} S_{ref}$$

where $n$ is the number of headwater basins in the target $0.5° \times 0.5°$ grid cell; $i$ (=1-n) is the serial number of each headwater basin; $S_{i\_j}$ (Mg day$^{-1}$ basin$^{-1}$) is the daily sediment delivery from land to river from headwater basin $i$ on day $j$. $S_{i\_ref}$ (Mg day$^{-1}$ basin$^{-1}$) is the daily sediment delivery from land to river from headwater basin $i$ under reference runoff and vegetation cover conditions. $S_j$ (g day$^{-1}$ grid$^{-1}$) is the total daily sediment delivery from land to river in the target

$0.5°\times0.5°$ grid cell on day $j$; $S_{ref}$ (g day$^{-1}$ grid$^{-1}$) is the total daily sediment delivery from land to river in the target $0.5°\times0.5°$ grid cell under reference runoff and vegetation cover conditions; $R_{ref}$ (mm day$^{-1}$) is the assumed reference daily surface runoff; $R_j$ (mm day$^{-1}$) is the simulated surface runoff on day $j$; $R_{30\_ref}$ (= 1 mm 30-minutes$^{-1}$) is the assumed daily maximum 30-minutes runoff; $R_{30\_j}$ (mm 30-min$^{-1}$) is the maximum value of the 48 half-hour runoffs on day $j$; $C_{ref}$ (0-1, dimensionless) is the cover management factor and is set to 0.1 for the reference state; $C_j$ (0-1, unitless) is the simulated cover management factor on day $j$; $DA_i$ (m$^2$) is the drainage area of headwater basin $i$; $K_i$ and $LS_i$ is the soil erodibility factor and the slope length and steepness factor, respectively. $a$, $b$, $c$ and $d$ are model parameters.

As this upscaling scheme has been explained in detail by Zhang et al. (2020), we only give a brief overview in this study. Nonetheless, we have revised the method section of this manuscript and added a supplementary figure (i.e. Fig. S2) to give a better explanation on the method for simulating sediment delivery from uplands to river channels. Please see our response to your comment #2.

**16. Line 250: ORCHILEAK Clateral → subscript lateral**

We have revised this sentence following your suggestion. Please see "For the use of these reference sediment delivery estimates in ORCHIDEE-Clateral, the values were first calculated for each headwater basin derived from high resolution geodata (Fig. S2e)" (lines: 249-250)

**17. Line 251: was this done for various reference conditions?**

The reference conditions have no impact on model outputs. We only need to calculate the sediment delivery under one reference condition, and the choice of the values for runoff, peak runoff and vegetation cover do not have an influence on the rescaled daily erosion rates simulated in ORCHIDEE Clateral. Please see our response to your comment #15.

**18. Line 254: awkward sentence; '…force the simulation of Then…' ?????**

Sorry for that mistake. We have revised this sentence. Please see "This aggregated dataset is then used to force the simulation of the actual daily sediment delivery ($S_j$, g day$^{-1}$ grid$^{-1}$) in ORCHIDEE-Clateral," (lines: 254-255)

**19. Line 260: Same b as in Eq.1? Where is S_iday located in Figure 1?**

Yes, the b in Eq. 4 is same as that in Eq. 1. Please see our response to your comment #15 for detailed explanation. $S_{iday}$ has been changed to $S_j$ in the revised manuscript and it is represented by the $F_{RO}$ in Fig. 1b (purple line, $F_{RO\_sed}$). $F_{RO}$ represents the surface water flux from land to river network. The eroded sediment flow from land to river network by following $F_{RO}$.

**20. Line 262: R_30_k not in Eq. 3 and 4**

Sorry for the typo. It should be $R_{30\_j}$, the simulated maximum 30-minutes runoff on day $j$.

**21. Line 304: Is $F_{Fout\_sed}$ identical with $S_{iday}$? Please remove sediment in this sentence, because this is confusing due to the fact that there is no storage in the fast water reservoir.**

Yes, the amount of $F_{Fout\_sed}$ is identical to $S_j$ (i.e. the $S_{iday}$ in the last version of our manuscript). In our model, $S_{fast}$ is the reservoir fed by surface runoff. Therefore, sediment, POC, DOC and dissolved $CO_2$ in the surface runoff will first enter the $S_{fast}$ (i.e. the erosion process), and then enter the river streams. Given this, we prefer to keep the sediment in this sentence.

**22. Figure 1a: Not sure what the direction of arrows indicates. I suggest that they point from the text to the feature in the graphic (if this is not related to vertical fluxes; unlikely for sediment). S_river and S_flood is used in Figure caption but not within the Figure itself.**

In Fig. 1a, we have deleted the arrows which do not represent flow directions. $S_{river}$ and $S_{flood}$ read be $S_{riv}$ and $S_{fld}$, respectively, and we have corrected this error. The revised Fig. 1 is:

[Figure]

[Figure]

**Figure 1** Simulated lateral transfer processes of water, sediment and carbon (POC, DOC and $CO_2$) in ORCHIDEE-C$_{lateral}$ (a) and a schematic plot for the reservoirs and flows of water, sediment and carbon represented in the routing module of ORCHIDEE-C$_{lateral}$ (b). $S_{soil}$ is the soil pool. $S_{rivbed}$ is the sediment (also POC) deposited in river bed. $S_{fast}$, $S_{slow}$, $S_{riv}$ and $S_{fld}$ are the 'fast', 'slow', stream and flooding water reservoir, respectively. $F_{RO}$ and $F_{DR}$ are the surface runoff and belowground drainage, respectively. $F_{Fout}$ and $F_{Sout}$ are the flows from fast and slow reservoir to the stream reservoir, respectively. $F_{up2riv}$ and $F_{down2riv}$ are the upstream inputs and downstream outputs, respectively. $F_{riv2fld}$ is the outputs from river stream to the flooding reservoir. $F_{fld2riv}$ is the return flow from flooding reservoir to stream reservoir. $F_{bed2fld}$ is the transform from deposited sediment in river bed to floodplain soil. $F_{bero}$ is bank erosion. $F_{rd}$ and $F_{rero}$ are the deposition and re-detachment of sediment and POC in river channel, respectively. $F_{sub}$ is the flux of DOC and CO2 from floodplain soil (originated from the decomposition of

submerged litter and soil carbon) to the overlying flooding water. $F_{fd}$ is the deposition of sediment and POC and the infiltration of water and DOC. $F_D$ is the wet and dry deposition of DOC from atmosphere and plant canopy. $DOC_l$ and $DOC_r$ are the labile and refractory DOC pool, respectively. $POC_a$, $POC_s$ and $POC_p$ are the active, slow and passive POC pool, respectively.

**23. Line 323ff: I wonder if the author mix up several things. In rivers, suspended sediment (esp silt and clay which are transport agents of POC) is transported as wash load. The transport of the wash load is not transport capacity limited but supply limited. Whether changes in the channel bed need to be considered depends on the target time scale. Therefore, I am not sure if it is required to discuss Eq. 7 in detail. If the authors specify the relevant scales much earlier in their paper, the lengthy discussed could be reduced.**

The scheme used in our model to simulate sediment transport and deposition in river channel is similar to that in many previous sediment transport models such as SWAT (Neitsch et al., 2011), WEPP (Nearing et al., 1989) and ROTO (Arnold et al., 1995). These previous models all assume that suspended sediment in river flows will deposit to river bed when the amount of suspended sediment (including clay and silt sediments) exceeds the sediment transport capacity of the water flow. Moreover, observations also show that clay and silt sediment can deposit in river channels.

The target time scale is decades to a few hundreds of years. Thus, we did not consider the evolution and diversion of river channels. We have added a sentence explaining this important point. We have added a sentence to explain this. Please see "Moreover, as our model mainly aims to simulate the lateral transfer of sediment and carbon at the decadal to centennial timescale, rather than covering the past thousands of years or even longer time periods, we did not consider the evolution and diversion of river channels in our study." (lines: 382-385)

Arnold, J. G., Williams, J. R., and Maidment, D. R.: Continuous-time water and sediment-routing model for large basins. J. Hydraul. Eng., 121, 171-179, 1995.

Nearing, M. A., Foster, G. R., Lane, L. J., and Finkner, S. C.: A Process-Based Soil Erosion Model for USDA-Water Erosion Prediction Project Technology. Transactions of the Asae, 32, 1587-1593, 1989.

Neitsch, S. L., Williams, J. R., Arnold, J. G., and Kiniry, J. R.: Soil and Water Assessment Tool Theoretical Documentation Version 2009. Texas Water Resources Institute, College Station, 2011.

**24. Line 357: what is e1 in Eq.8?**

**Line 361: Drainage area in Eq 1 was defined with DA. Use same symbols!**

**Line 361: In Eq. 8 the term F_down2riv_h20 is used, here in the text you use F_down2riv_sed but talk about water discharge. I am confused. I assume you refer to the Psi-equation of Cohen et.al. If this is true**

Sorry for the mistakes. We have revised the manuscript according to your comments. In the revised manuscript, we provided the explanation of $e_1$ and changed symbol for drainage area to *DA*. Please see: "

$$TC = \frac{\omega\, q_{ave}{}^{0.3}\, A^{0.5} \left(\frac{q_{iday}}{q_{ave}}\right)^{e_1} (24\times60\times60)}{F_{down2riv\_h20}} \tag{8}$$

$$e_1 = 1.5 - max(0.8, 0.145 \log_{10} DA) \tag{9}$$

where ω is the coefficient of proportionality, $q_{ave}$ (m$^3$ s$^{-1}$) is long-term average stream flow rate obtained from an historical simulation by ORCHILEAK (Table 1), $q_j$ (m$^3$ s$^{-1}$) is stream flow rate on day *j*, $e_1$ is an exponent depending on the upstream drainage area ($DA$, m$^2$), $F_{down2riv\_h20}$ (m$^3$ day$^{-1}$) is the daily downstream water discharge from the stream reservoir." (lines: 331-336)

**25. Line 371: Assuming that channel bank erosion only occurs if no sediment is left at the channel bank is not a meaningful assumption. Many rivers migrate without changing their channel bed.**

For a global scale land surface model, it is very hard to simulate in detail the erosion of river channel, as well as the evolution of the river channel, due to the coarse resolution as well as important limitations associated with fragmented forcing data and calibration data. Thus, we have introduced a very simple scheme to simulate the transport and deposition of riverine sediment, and the erosion of the river channel. The geomorphic properties of river channel in our model are assumed to be fixed. To avoid too much sediment deposition and/or too high erosion

rate of the river channel, we assumed that bank erosion only occurs when all of the previously deposited sediment is eroded. We recognize that this assumption might not hold true for all rivers. However, as there is still no well-tested algorithm to simulate river channel erosion at continental/global scales (to our knowledge) to date, we believe that relying on this simplifying assumption is the best we can do for now. Nevertheless, following the reviewer's comment, we now discuss this shortcoming in the revised ms.. Please see "In the present version of ORCHIDEE-C$_{lateral}$, the lateral transfers of sediment and carbon is simulated using a simplified scheme, due to the fragmented nature of large-scale forcing (e.g. geomorphic properties of the river channel) and validation data (e.g. continuous sediment and carbon concentration data in river streams and deposition/erosion rates inriver channels). We recognize that this simplification induces significant uncertainties in model outputs, especially regarding changes in lateral sediment and particulate carbon transfers under climate change and direct human perturbations." (lines: 803-809)

"Overall, we encourage future studies to produce large-scale databases on the geomorphic properties of global river channels (e.g. river depth and width) and to develop large-scale sediment transport models capable of producing more realistic simulations of sediment deposition, re-detachment and transport processes, including the exchanges of water, sediment and carbon between river streams and floodplains." (lines: 835-840)

**26. Line 387: F_up2fld_sed not needed in my point of view. Why was this introduced and and why is there no F_riv2fld?**

Please see our response to your comment #3.

**27. Line 390: sum in text but negative sign in Eq. 16 → furthermore, I don't understand the approach here. Why does evaporation and infiltration contribute to sediment deposition? Please explain.**

Sorry for the mistake in Eq. 16. The total sediment deposition in floodplain is calculated as the sum of a natural deposition + the deposition due to evaporation ($E_{h2o}$, m$^3$ day$^{-1}$) and infiltration ($I_{h2o}$, m$^3$ day$^{-1}$) of the flooding waters. We have changed the negative sign to plus sign (+).

$$F_{fd\_sed} = c_{flddep}\, S_{fld\_sed} + S_{fld\_sed}\frac{E_{h2o} + I_{h2o}}{S_{fld\_h2o}} \tag{16}$$

In our opinion, it is reasonable to assume that the evaporation and infiltration of flooding waters can contribute to sediment deposition (Fig. R1 below). For example, when 10% of the flooding water is infiltrating into soil, the sediment in this part (10%) of the flooding water will be deposited onto floodplains. Similar to infiltration, evaporation also results in decrease in the amount and depth of flooding water, thus can contribute to the sediment deposition.

[Figure]

**Figure R1**. Schematic diagram for the impacts of infiltration and evaporation on sediment deposition in floodplains.

**28. Line 400: Same f_topo as on hillslopes? How was this calculated? I am confused!**

The $f_{topo}$ is calculated from the slope steepness of river channel using the method in Vörösmarty et al., 2000. We have included some more explanation in the text before this line and in Table 1. Please see "Daily water ($F_{Fout\_h2o}$, m$^3$ day$^{-1}$) and sediment ($F_{Fout\_sed}$, g day$^{-1}$) flows from fast water reservoir to stream reservoir are calculated from a grid cell-specific topographic index $f_{topo}$ (unitless, Vörösmarty et al., 2000) extracted from a forcing file (Table 1) and a reservoir-specific factor $\tau$ which translates $f_{topo}$ into a water residence time of each reservoir (Eqs. 5, 6)." (lines: 303-307)

In addition, we have added '(Table 1)' behind the $f_{topo}$ in this line to point the readers to more information. Please see "where $\tau_{flood}$ is a factor which translates $f_{topo}$ (Table 1) into a water residence time of the flooding reservoir." (lines: 375-376)

**29. Line 485: I guess that you run in the problem of equifinality of you simple calibrate five parameters against one observation (sediment yield). Please discuss this problem.**

Agreed and we have now added some discussion around this. Please see "Further model calibration, evaluation and development is necessary for improving our model. Due to the limitation of observation data, we calibrated the parameters controlling sediment transport, deposition and re-detachment (i.e. $\omega$, $c_{rivdep}$, $c_{flddep}$, $c_{ebed}$ and $c_{ebank}$ in Table S1) in stream and flooding reservoirs only against the observed sediment yield. Even though our model can overall capture the lateral transfers of sediment and carbon in many rivers in central and northern Europe, more observation data are crucially needed to further evaluate the performance of our model, in particular in southern Europe. In addition, it is still unknown whether our model can satisfactorily simulate intermediate processes such as sediment deposition in river channels and floodplains, as well as the rate of river channel erosion. It is also unknown whether our model would perform satisfactorily in regions with very different climates than Europe such as the tropical region. Thus, in the future, an important aim will be to further calibrate our model against more detailed observation data (e.g. sediment deposition rate in river channels and floodplains) and extend the model application to regions of contrasting climate, vegetation and topography. Moreover, the GLORICH database (Hartmann et al., 2019) only provides instantaneous observations of riverine organic carbon concentrations and it is therefore difficult to evaluate the model's ability to reproduce temporal trends. Therefore, future modelling efforts should be combined with data mining efforts targeting the collection of continuous (e.g. daily) and long-term observational data of organic carbon content and fluxes in streams and rivers." (lines: 919-936)

**30. Line 509: Major parts of Europe are missing (e.g. no observation for Spain, France GB, Italy), esp. Mediterranean rivers are not covered in this study.**

Please see our response to your comment #4.

**31. Line 517: Indicate which stations in Rhine were used. POC is strongly discharge dependent, please indicate how many measurements at which discharge are used.**

We have added the specific stations of the observed riverine POC data used in this study, as well as the number of POC measurements at these three stations. Please see "POC was measured at only two sites (Bad honnef (51 measurements) and Bimmen (78 measurements)) in the Rhine catchment and one site (Rheine, 36 measurements) in the Ems catchment (Fig. S4d)." (lines: 500-502)

The discharge at these three sites are added to the caption of Fig. 6. Please see:

**Figure 6** Comparison between the observed (instantaneous measurement) and simulated (monthly average value) riverine POC concentrations at three gauging sites. In figure (b), (d) and (f), the histogram and error bar denote the mean and standard deviation of POC concentrations, respectively. Long-term average water discharge rates at Bad Honnef, Bimmen and Rheine during the observation periods are 2023, 2100 and 80 $m^3$ $s^{-1}$, respectively.

**32. Line 606: It seems that the model underestimates the observed DOC variability (Fig. 4b), however, this is in contrast to the Figure S8. Please explain this discrepancy.**

Fig. 4b shows the spatial variation of DOC concentrations across 314 gauging stations. However, in Fig. S8 and Fig. 5 show the temporal variation of DOC concentrations at each of the 15 gauging stations (with relatively long-term observation data) of major European rivers. In addition, the number of measurements of DOC concentrations at many of the 314 gauging stations is very limited (less than 20 or even 10). The calculated seasonal variation in DOC concentrations based on these limited DOC measurements at these sites is highly uncertain and may be smaller than the value calculated based on simulated DOC concentrations.

**33. Line 649: How does this number relates to empirical sediment budgets? Is that in the order of observations? Please discuss. Line 679: are there any empirical values to compare with?**

To our knowledge, many studies have investigated the sediment delivery ratio from upland soils to river network (i.e. the ratio of sediment entering river network to gross upland soil erosion) using empirical soil erosion models or observation data. However, to our knowledge, no study has investigated the fate of the sediment entering the river network (e.g. the fraction of deposited sediment in river channels and floodplains) at continental scale of Europe, mainly because the amount of sediment entering river network is hard to measure at large spatial scale. In this study, although we cannot directly evaluate the simulated deposited fraction of riverine sediment, we have evaluated the simulated sediment discharge rates against observations at 221 gauging sites (Fig. 3). As our model can overall capture the sediment discharge rates in many rivers, in particular the sediment delivery rates from upstream to downstream of some rivers (Fig. S4c), we

infer that the simulated deposited fraction of riverine sediment should overall be reliable too. Nonetheless, we acknowledge that more observation data on the sediment and carbon deposition rate in floodplains would be very useful to further calibrate and evaluate our model. We have discussed this in the revised manuscript. Please see: "In addition, it is still unknown whether our model can satisfactorily simulate intermediate processes such as sediment deposition in river channels and floodplains, as well as the rate of river channel erosion. It is also unknown whether our model would perform satisfactorily in regions with very different climates than Europe such as the tropical region. Thus, in the future, an important aim will be to further calibrate our model against more detailed observation data (e.g. sediment deposition rate in river channels and floodplains) and extend the model application to regions of contrasting climate, vegetation and topography." (lines: 925-932)

**34. Line 661: any idea what causes this decline?**

We now do and these results are actually part of a follow-up manuscript in preparation. By running our model under different climate change and land use scenarios, we found that the decrease in sediment delivery from land to river during the past century is mainly caused by land cover change (afforestation), followed by atmospheric $CO_2$ increase (which increases plant canopy and root biomass and litter cover, then induces the decline in upland erosion), and temperature increase. Of course, in different regions of Europe, the contributions of land use change and climate change to the changes in lateral sediment and carbon transfers can be very different.

**35. Fig. 10: Does C_riv2land represent the transport from river channels to floodplains? If yes, I suggest to consider floodplains not at 'land'.**

In our model the floodplain is indeed regarded as a part of the land, and the carbon deposited (POC) or infiltrated (DOC and dissolved $CO_2$) to floodplains is added to the soil carbon pool. Moreover, flooding generally occurs occasionally in most regions in Europe. During most of the time, the floodplains are not inundated. To give a more accurate definition of the $C_{riv2land}$, we clarified that $C_{riv2land}$ denotes the transport of carbon from river streams to floodplains. Please see "$C_{land2riv}$, $\underline{C_{riv2land}}$ and $C_{riv2sea}$ are the average annual carbon fluxes from land to inland waters, from inland waters to floodplains and from inland waters to the sea, respectively. SD is the standard deviation." (lines: 717-719)

**36. Line 753: flooding decreases SOC stored in floodplain soil???? This is total contradicting our expectation and needs discussion**

You might have misunderstood this sentence. We are saying "Accounting for flooding thus decreases the decomposition rate of litter and SOC stored in floodplain soils.". Flooding decreases the decomposition rate of litter and SOC in floodplain soils, thus favors an increase in the SOC stock.

**37. Line 747: can you account for the soil-wettness driven changes in soil temperature? Is this effect significant?**

Yes, the effects of ecosystem water cycle on land surface and soil temperatures are represented in ORCHIDEE model. By comparing the soil moisture and temperature simulated by ORCHIDEE-Clateral with activated and deactivated lateral water and C transfers, we find that the lateral water transfer, in particular flooding waters, can slightly but significantly change the soil moisture and temperature at grid cells with a large area of floodplains (Fig. S15).

**38. Line 754: any number how this influences the C budget. Many empirical studies argue that this effect is important and strongly increases the OC retention in floodplains. Could this somehow be quantified?**

It is still very hard to quantify the changes in land C budget caused by the changes in vertical SOC distribution alone. By comparing the SOC stocks simulated by ORCHIDEE-Clateral with activated and deactivated lateral C transfer process, we can quantify the changes in SOC stock caused by lateral C transfer. However, lateral C transfer can affect SOC stock at a specific location through several different mechanisms: 1) soil erosion or deposition can directly increase or decrease the SOC stock; 2) lateral water transfer can affect SOC decomposition rate by altering soil moisture; 3) lateral water transfer can affect vegetation productivity, which is the dominant C source of SOC; 4) soil erosion and deposition can affect SOC decomposition by altering vertical SOC profile, as the soil moisture  and priming effect in different soil layers are different. To estimate the influence of each of these four mechanisms on the changes in SOC stock, the other three mechanisms would have to be cknown. However, it is still very hard to evaluate each of the 4 mechanisms individually in ORCHIDEE-Clateral. In particular, the changes in vertical SOC profile are directly determined by the amounts of eroded or deposited sediment and carbon.

**39. Line 793: this very low increase somehow contradicts the large amount of POC retention in floodplains. You somehow provide numbers with SOC stock decrease in mountains and SOC stock increases in floodplains, but does this mean that SOC retention in floodplains is compensated by soil degradation at eroding sites?**

Please see our response to your comment #5.

**40. Line 811: Please cite Hoffmann 2013 (GBC): they present results for hillslope and floodplain storage of OC for the Rhine basin.**

Thanks for your suggestion. We have added Hoffmann et al. (2013) as one of our references. Please see "For example, many studies suggest that a substantial portion of the eroded sediment and carbon is deposited downhill at adjacent lowlands as colluviums, rather than exported to the river (Berhe et al., 2007; Smith et al., 2001; Hoffmann et al., 2013; Wang et al., 2010)." (lines: 863-866)

**41. Line 826ff: Considering NP might not only decrease NPP at eroding site but also increase NPP at depositional site. Correct? If yes, leave some words in the paragraph on depositional sites as well. Certainly, a worthwhile action to link NP here.**

Indeed, we have indicated that both soil erosion and sediment deposition can affect vegetation productivity by modifying soil nutrient availability. Please see "many studies have indicated that the soil erosion and sediment deposition can affect vegetation productivity by modifying soil nutrient (e.g. nitrogen (N) and phosphorus (P)) availability (Bakker et al., 2004; Borrelli et al., 2018; Quine, 2002; Quinton et al., 2010)." (lines: 884-887)

**42. Line 839: Hoffmann et al (2020, ESurf) provides a way to differentiate exsitu and insitu OC in rivers. This paper also offers more infos on POC in the Moselle and Rhine rivers.**

Thanks for your suggestion. Findings in Hoffmann et al (2020, ESurf) are very interesting and should be very helpful for our further model development and evaluation. In addition, we have cited Hoffmann et al (2020, ESurf) as reference for the fact that river-borne phytoplankton can contribute to the riverine organic carbon. Please see "Although soils are the major source of riverine organic carbon, domestic, agricultural and industrial wastes, as well as the river-borne phytoplankton can also make significant contributions (Abril et al., 2002; Meybeck, 1993; Hoffmann et al., 2020)." (lines: 892-894)

**43. Line 849: Could the routing be done using DEMs with better spatial resolution to overcome limitations of the routing on low-res DEMs?**

Yes, some of our colleagues are developing a routing scheme at higher spatial resolution. We will implement the rouging scheme in the future version of our model after their development is finished.

**44. Figure S2: bad quality, can't read the text**

We have changed revised the original Figure S2. The new figure is:

[Figure]

**Figure S4** Geographical location of the gauging stations for river discharge (a), bankfull flow (b), sediment discharge (c) and riverine organic carbon discharge (d) used in this study. Figure (d) also shows the spatial distribution of 57 catchments in Europe. The simulated average net soil loss rates (g m$^{-2}$ yr$^{-1}$) of these 57 catchments were compared to the average net soil loss rates extracted from the sediment delivery data provided by the ESDAC (see section 2.3 of the main text).

**45. Figure S4: give names of gauging stations**

We have provide the names of the gauging stations in the original Figure S4. Please see:

[Figure]

**Figure S6** Comparison between the simulated and observed time series of mean annual water discharge rates at 14 gauging stations.

**46. Figure S5: bad quality of left map**

We have revised the original Figure S5. The new figure is:

[Figure]

**Figure S7** (a) Comparison between the river network extracted from the STN-30p database at 0.5° resolution (blue) (i.e. the forcing data of stream flow directions used in this study) and the river network derived from the HydroSHEDS DEM data at 3″ resolution (red); (b) the real river network in the estuary region of the Danube River (obtained from © Google Maps). GRDC_ID denotes the identify number of the gauging station in the GRDC database (Table 1).

---

## Author Comment (AC3)

**Referee #3**

**1. The submitted manuscript describes a development of ORCHIDEE, in which several branches have been combined to better assess the global carbon cycle. As the authors state it has long been ignored that the carbon transport from land, via rivers to the ocean also plays an important role. I was developing something comparable a few years back and am very certain about the importance and high relevance of such a model enhancement.**

**The results are promising. But the description of the methods and the presentation of the results can be improved, e.g., there are lengthy quotes in the methods which can be reduced, or some of the table/figures in the SI should be moved to the main text for clarification and a better understanding.**

**Overall, I think the content of the manuscript is convincing but can be strengthened by removing unnecessary parts and moving some explaining information from the SI to the main text. I believe that the manuscript can be improved, and I recommend it for publication after major revision.**

**Below I list some unclear sentences, problematic points, and questions. This list is chronological and not based on the importance of the points. I was much more detailed in the methods, but think that the results/discussion section can also profit from the listed points.**

Thanks a lot for your positive comments, as well as your specific suggestions and corrections below. Your comments are very helpful to improve our manuscript. We have carefully studied them and revised our manuscript accordingly. Please see our detailed responses to your comments below.

**2. l.86. Is the litter also included as a part of OC input? It is mentioned a few times, but not explained or shown in Fig 1?**

No, litter is not included as a part of OC input. In our model (also many other erosion models), litter is an important factor (the C-factor in MUSLE, Eq. 1) for protecting soil from being eroded.

**3. Please also mention the branch names (e.g.in l. 114)**

We have added the branch names. Please see "In previous studies, the leaching and fluvial transfer of DOC and the erosion-induced delivery of sediment and POC from upland soil to river network have been implemented in two different branches of the ORCHIDEE LSM (i.e. ORCHILEAK (Lauerwald et al., 2017) and ORCHIDEE-MUSLE (Zhang et al., 2020))." (lines: 112-116)

**4. An overview map of the catchments would be helpful or at least of how much of the drainage area of Europe is covered.**

Our study covers all catchments in Europe. We have provided the detailed information of our study area and map of catchments in the Methods and Results section. Please see section 2.3.

**5. l. 135. Is 'bare soil' a PFT (plant functional type)?**

We have revised the this sentence to give a more accurate description. Please see "In its default configuration, ORCHIDEE represents 13 land cover types, with one for bare soil and 12 for lands covered by vegetation (eight types of forests, two types of grasslands, two types of croplands)." (lines: 134-136)

**6. l. 148: Are the two litter pools part of the POC in soil? Please clarify.**

The litter pools are also organic carbon pools. However, they did not contribute to the lateral POC transfer. In our model, only soil organic carbon (SOC) contributes to the lateral transfer of POC. To avoid misunderstanding, we have revised this sentence from the original "ORCHIDEE-SOM subdivides the particulate organic carbon stored in soil into two litter pools (metabolic and structural) and three SOC pools (active, slow and passive) that differ in their respective turnover times.". to "ORCHIDEE-SOM represents two litter pools (metabolic and structural) and three SOC pools (active, slow and passive) that differ in their respective turnover times." (lines: 148-150)

**7. ll. 156-162 is a quote. Please shorten or refer right away to the cited paper.**

We have shortened and reorganized the quoted contents. Please see "Adsorption and desorption of DOC follows an equilibrium distribution coefficient calculated from soil clay and pH. Free DOC can be transported with the water flux simulated by the soil hydrological module of ORCHIDEE. However, DOC adsorbed to soil minerals can neither be decomposed nor transported (Camino-Serrano et al., 2018)." (lines: 157-160)

**8. l. 183. How does the DOC 'enter' the water? Does it depend on vegetation cover? Please clarify.**

When flooding water is infiltrated into soil, the DOC and $CO_2$ in flooding water will naturally enter into soil along with the infiltrating water. The infiltration rate of flooding water depends on soil properties and soil water content, but does not depend on vegetation cover. As this process is originally developed in the ORCHILEAK model and has been introduced in detail in Lauerwald et al. (2017), we only gave a brief overview about this part of the model.

Lauerwald, R., Regnier, P., Camino-Serrano, M., Guenet, B., Guimberteau, M., Ducharne, A., Polcher, J., and Ciais, P.: ORCHILEAK (revision 3875): a new model branch to simulate carbon transfers along the terrestrial–aquatic continuum of the Amazon basin. Geosci. Model Dev., 10, 3821-3859, 2017.

**9. Table 1: Is a spatial resolution of 0.5° (55km\*55km max) sufficient to inform the model about the 'Area fraction of river surface'? Later it is mentioned that for the delta of the Danube the high resolution was problematic (because of gauging station data available). But is there also a problem of scale for other data? E.g. 'maximum water storage in river channel' is also only on 0.5°.**

The area fractions of river surface or floodplain in each grid cell at 0.5° are derived from high-resolution (e.g. 3″ or 90 m) topographic or satellite data (see the references in the Methods) (e.g. Fig. R1 below). Thus, these area fraction data should be reliable. Indeed, there can be many different river channels in each 0.5°×0.5° pixel in reality. However, it is almost impossible for a global land surface model to explicitly simulate the riverine processes for each individual river channel. Thus we assume that there is one virtual river channel in each 0.5°×0.5° pixel (line 343). The surface area of this virtual river is the sum of all real rivers and the flow direction of this virtual is assumed to be same to the largest real river.

[Figure]

**Figure R1.** (a) Floodplain region derived from high-resolution (3″) DEM data and (b) the floodplain fraction (0-1) in each 0.5°×0.5° pixel. Map (b) is derived from map (a).

**10. l. 207. The temporal resolution of 6'' for CO2 evasion is totally reasonable, but for the DOC decomposition it seems to be too much (as for the litter added, l. 212), also given the fact that the temperature (on which the decomposition is also depending) only changes every 3h.**

**- Why is there a water temperature of 28° assumed? Is this realistic?**

Sorry for the mistake. Only $CO_2$ evasion is simulated at a time step of 6 minutes. DOC decomposition is simulated at daily time step. We have revised our manuscript accordingly. Please see "Decomposition of DOC in stream and flooding waters is calculated at daily time step based on the prescribed turnover times of labile (2 days) and refractory (80 days) DOC in waters (when temperature is 28 °C) and a temperature factor obtained from Hanson et al. (2011). $CO_2$ evasion in inland waters is simulated using a much finer integration time step of 6 minutes." (lines: 205-208)

The temperature of 28° is not assumed in tour study. It is only a reference temperature at which the turnover rate of DOC was originally observed . Using a temperature-dependent function (Hanson et al., 2011), we can then calculate the turnover rate of DOC at any other temperature. For example, the DOC turnover rate ($dDOC_{Ti}$) at temperature $T_i$ is calculated as:

$$dDOC_{Ti} = dDOC_{T28}\ 1.073^{(T_i-28.0)}$$

where $dDOC_{T28}$ is the DOC turnover rate at 28 °C.

**11. ll. 212-218. Again, a quote. Can be shortened. Is wind speed considered for the CO2 evasion?**

We have shortened and reorganized the quoted contents. Please see "The $CO_2$ partial pressures ($pCO_2$) in water column is first calculated based on the temperature-dependent solubility of $CO_2$ and the concentration of dissolved $CO_2$ (Telmer and Veizer, 1999). Then the $CO_2$ evasion is calculated based on the gas exchange velocity, the water–air gradient in $pCO_2$, and the surface water area available for gas exchange (Lauerwald et al., 2017)." (lines: 208-210)

In current version of our model, the effect of wind speed on $CO_2$ evasion is not represented.

**12. ll. 227-238. Again, a pretty long quote.**

With these lines, we describe the MUSLE model used to simulate sediment delivery from uplands to river networks. As it is better to keep the equations and variable definitions identical to that used in our previous publication (Zhang et al., 2020, JAMES), we thus directly quoted the contents used in our previous paper. Actually, we have discussed this with the editor of our manuscript before. In fact, he recommended us to quote these contents.

**13. l. 243 mentions a 'management factor', which is only explained in l.264.**

In the original line l.243, we only give a general explanation on the definition of $C_{ref}$ in MUSLE model (i.e. the cover management factor) (see lines 241-242 in the revised ms). In original line 264, we provided the specific method for calculating the cover and management factor ($C_j$) (see 262-265 in the revised ms).

**14. ll. 250. It is not clear on which resolution the model runs. Some of the input is fine scale (e.g. 250m for floodplains), but then the results are aggregated to 0.5°. Please clarify.**

**ll. 254-258. The sentence is not understandable.**

**l. 268. Why only headwater?**

Sorry for having not provided a detailed description on the upscaling scheme for simulating sediment and POC delivery from headwater basins to river networks, as this scheme has already been explained in detail in our previous paper by Zhang et al. (2020). The headwater basins are extracted from high-resolution (3″ (~ 90 m) in this study) DEM, and there can be many headwater basins in each 0.5° grid cell of ORCHIDEE-Clateral (Fig. S2, see below). To simulate

the sediment and POC deliveries from land soils to river networks, we first extracted the headwater basins and river networks from high-resolution DEM data (Fig. S2a,d). Then the MUSLE model (Eq. 1) was applied at each headwater basin to calculate the reference net soil loss rate (Fig. S2e) under a given set of reference runoff and vegetation cover conditions. By summing up the net soil loss from all headwater basins in each grid-cell (Fig. S2e), we can calculate the total soil loss rate from land to river network under reference runoff and vegetation conditions (Eq. 3, Fig. S2f). Finally, the aggregated data of soil loss rate at 0.5° is used as a forcing file of ORCHIDEE-Claateral to calculate the actual daily soil loss rate from land to river (Eq. 4, Fig. 2g). The upscaling scheme and the method for extracting headwater basins and river networks has be introduced in detail in Zhang et al. (2020) , that is why we only give a brief overview in this manuscript.

Nonetheless, we have revised the method section of this manuscript and added a supplementary figure (i.e. Fig. S2) to give a better explanation on the method for simulating sediment delivery from uplands to the river channels. Please see "To give an accurate simulation of sediment delivery from uplands to river network and maintain  computational efficiency, an upscaling scheme which integrates information from high-resolution (3″) topographic and soil erodibility data into a LSM forcing file at 0.5° spatial resolution, has been introduced (see details in Zhang et al., 2020, Fig.S2). With this upscaling scheme, the erosion-induced sediment and POC delivery from upland soils to the river network, as well as the changes in SOC profiles due to soil erosion had already been implemented in ORCHIDEE-MUSLE (Zhang et al., 2020). The sediment delivery from small headwater basins (which are basins without perennial stream and are extracted from high-resolution (e.g. 3″) digital elevation model (DEM) data, Figs. S2a&d) to the river network (i.e. gross upland soil erosion – sediment deposition within headwater basins) is simulated using the Modified Universal Soil Loss Equation model (MUSLE, Williams, 1975). As introduced in Zhang et al. (2020), "the daily sediment delivery rate from each headwater basin ($S_{i\_ref}$, Mg day$^{-1}$ basin$^{-1}$) is first calculated for a given set of reference runoff and vegetation cover conditions (Fig. S2e)" (lines: 216-227)

From Fig. S2, you can find the model is finally run at a spatial resolution of 0.5°. The headwater basins cover all upland areas, including not only the upland regions where the main streams originate, but also all upland regions where the tributary streams originate.

[Figure]

**Figure S2** Upscaling scheme used in ORCHIDEE-MUSLE (Zhang et al., 2020) for calculating the sediment delivery rate from headwater basins to river networks. MUSLE is the Modified Universal Soil Loss Equation; DEM is the digital elevation model (m); $K$ is the soil erodibility factor (Mg MJ-1 mm-1); $R_{ref}$ is the assumed reference daily runoff depth (= 10 mm day$^{-1}$); $R_{30\_ref}$ is the assumed reference maximum 30-minutes runoff depth (= 1 mm 30-minutes$^{-1}$); $C_{ref}$ (= 0.1, dimensionless) is the assumed reference cover management factor; $R_j$, $R_{30\_j}$ and $C_j$ are the simulated daily total runoff depth, daily maximum 30-minutes runoff depth and daily cover management factor, respectively.

Zhang, H., Lauerwald, R., Regnier, P., Ciais, P., Yuan, W., Naipal, V., Guenet, B., Van Oost, K., and Camino-Serrano, M.: Simulating Erosion-Induced Soil and Carbon Delivery From Uplands to Rivers in a Global Land Surface Model. J. Adv. Model. Earth Syst., 12, e2020MS002121, 2020.

**15. Figure 1. (I like it.) It would be helpful if the naming in part a and b would be consistent. Having the time steps mentioned in a separate box is helpful and keeps the figure clean, but it is not clear which processes belong to which group. E.g. 'deposition' - mentioned several times in the figure is not part of the 'time step' box; 'decomposition' in the time-step box consists of 'litter and SOC' but the litter can't be found in the figure. It could also be helpful to add the abbreviations from part b to part a, to make the link clearer and easier to understand. (b) is missing in the caption.**

**ediment transport**

Thanks to your careful checking and helpful suggestion. In the time step box, we have changed the 'erosion & fluvial transfer' to 'Lateral sediment & C transfers'. The erosion, leaching, transport and deposition processes are all belong to the 'Lateral sediment & C transfers', thus are all simulated at daily time step. We changed the original 'Litter & SOC decomposition' to 'Organic C decomposition', as we only presented the decomposition of POC and DOC in panel (a). In panel (a), we originally intend to show the main processes of lateral sediment and C transfers represented in our model. As we hope the make the readers can understand panel (a) without referring figure caption, we did not include the abbreviations in the original panel (a). In addition, panel (a) is mainly used to show the processes in real world, and panel (b) is mainly used to explain the framework used in our model (including the conceptualized C pools and C fluxes) to simulated the lateral transfers. Processes in panel (a) and (b) are not strictly same. Nonetheless, we have added some abbreviations in panel (b) to panel (a) following your suggestion. The revised Fig. 1 is shown below.

[Figure]

[Figure]

**Figure 1** Simulated lateral transfer processes of water, sediment and carbon (POC, DOC and $CO_2$) in ORCHIDEE-C$_{lateral}$ (a) and a schematic plot for the reservoirs and flows of water, sediment and carbon represented in the routing module of ORCHIDEE-C$_{lateral}$ (b). $S_{soil}$ is the soil pool. $S_{rivbed}$ is the sediment (also POC) deposited on the river bed. $S_{fast}$, $S_{slow}$, $S_{riv}$ and $S_{fld}$ are the 'fast', 'slow', stream and flooding water reservoir, respectively. $F_{RO}$ and $F_{DR}$ are the surface runoff and belowground drainage, respectively. $F_{Fout}$ and $F_{Sout}$ are the flows from fast and slow reservoir to the stream reservoir, respectively. $F_{up2riv}$ and $F_{down2riv}$ are the upstream inputs and downstream outputs, respectively. $F_{riv2fld}$ is the outputs from river stream to the flooding reservoir. $F_{fld2riv}$ is the return flow from flooding reservoir to stream reservoir. $F_{bed2fld}$ is the transform from deposited sediment in river bed to floodplain soil. $F_{bero}$ is bank erosion. $F_{rd}$ and $F_{rero}$ are the deposition and re-detachment of sediment and POC in river channel, respectively. $F_{sub}$ is the flux of DOC and CO2 from floodplain soil (originated from the decomposition of

submerged litter and soil carbon) to the overlying flooding water. $F_{fd}$ is the deposition of sediment and POC and the infiltration of water and DOC. $F_D$ is the wet and dry deposition of DOC from atmosphere and plant canopy. $DOC_l$ and $DOC_r$ are the labile and refractory DOC pool, respectively. $POC_a$, $POC_s$ and $POC_p$ are the active, slow and passive POC pool, respectively.

**16. ll. 329-343 and ll. 344-353. This is much more discussion than methods. Can be shortened and should be moved.**

We have moved these contents to the discussion section. Please see lines: 803-840

**17. ll. 384. What is the effect of scale here if you move the OC from one catchment to the next? How does is differ between large and small catchments?**

As we have explained in our manuscript ("Along with surface runoff ($F_{RO\_h2o}$, $m^3$ $day^{-1}$), the sediment delivery ($F_{RO\_sed}$, g $day^{-1}$) from uplands in each basin (i.e. each 0.5° grid cell in the case of this study) initially feeds an aboveground water reservoir ($S_{fast\_h2o}$, $m^3$) with a so-called fast water residence time" (lines 299-302), the so-called basin here is actually an 0.5°×0.5° grid cell. In other words, each 0.5°×0.5° grid cell is regarded as an individual basin. Therefore, in the original line 384 ('the bankfull flow of a specific basin is assumed to enter the floodplain in the neighboring downstream basin instead of the basin where it originates.'), we intend to explain that the bankfull flow of a specific 0.5°×0.5° grid cell is assumed to enter the floodplain the in neighboring downstream grid cell. So there is no effect of scale here.

**18. Section 2.2.2. What is the exact difference between sediment flow and the described POC flow, if it's closely linked to sediments? Is the POC calculations the same as clay?**

As we have described, 'In ORCHIDEE-$C_{lateral}$, the physical movements of POC in inland water systems are simply assumed to follow the flows of finest clay-sediment (Fig. 1b).' (lines: 392-393). The simulation of transport, deposition and re-detachment processes of POC in the river network and floodplains is similar to that of clay-sediments. Of course, POC is also impacted by decomposition, although this process is relatively minor in quantitative terms.

**19. l. 438. You use the water temperature to calculate the processes, but as input you use air temperature. How do you accommodate for the difference and the time-lag (e.g. over the course of a day)?**

The water temperature ($T_{water}$, °C) is calculated from local soil temperature ($T_{soil}$, °C) using an empirical equation (i.e. $T_{water} = 6.13 + 0.8*T_{soil}$) developed in Lauerwald et al. (2017). We have added an explanation on this. Please see "where $T_{water}$ (°C) is the temperature of water reservoirs and is calculated from local soil temperature using an empirical function (Lauerwald et al., 2017)." (lines: 416-417). Actually, Lauerwald et al. (2017) has tried to include a time-lag effect between air and water temperature, but it did not improve the prediction equation nor altered the simulated total $CO_2$ emission significantly.

**20. ll. 442. Why do you refer and explain so much about SOC here? The section title is 'POC transport and decomposition'. Maybe some reference to the SOC section would help to clarify.**

The scheme for simulating POC decomposition in waters follows that for SOC. With the explanation on SOC decomposition here, we intend to explain how the decomposition of POC is simulated and how we represented the accelerated POC decomposition during the transport process due to the breakdown of sediment aggregates.

**21. L. 468. You refer to Figure S1 in the SI, but this figure shows the return period over two years.**

Sorry for the mistake. It should be Fig. S4 in the revised manuscript. We have corrected this error.

**22. l. 484. A map or a list of the catchments would be helpful (as in SI Figure S2(d)).**

In this sentence, we intended to report that the sediment delivery data simulated by the WaTEM/SEDEM model have been calibrated and validated using observed sediment fluxes from 24 European catchments (Borrelli et al., 2018). These 24 catchments have been listed in Borrelli et al. (2018). We did not used these data to calibrate and evaluate our model. Thus we did not provide a table or figure to introduce these catchments.

Borrelli, P., Van Oost, K., Meusburger, K., Alewell, C., Lugato, E., and Panagos, P.: A step towards a holistic assessment of soil degradation in Europe: Coupling on-site erosion with

sediment transfer and carbon fluxes. Environ. Res., 161, 291-298, 2018.

**23. l. 549. Why is that set to 0.1? It is not clear at this point.**

The return period of flooding (=0.1 year) is calibrated against observed bankfull flows obtained from Schneider *et al*. (2011). We have explained this in the Methods (section 2.3). Please see "Existing observational data on $P_{flooding}$ are still very limited to our knowledge. Therefore, following Schneider *et al*. (2011), we also a constant $P_{flooding}$ to simulate the bankfull flows from European rivers and the observed long-term (1961–2000) average bank full flow rate (m$^3$ s$^{-1}$) at 66 sites obtained from Schneider *et al*. (2011) was used to calibrate $P_{flooding}$ (the optimized value is 0.1 year, Table 2).".

To avoid misunderstanding, we have revised the original texts from "By setting the return period of the daily flooding rate to 0.1 year," to "With the calibrated return period (= 0.1 year) of the daily flooding rate (see section 2.3)," (line: 533)

Schneider, C., Flörke, M., Eisner, E., and Voss, F.: Large scale modelling of bankfull flow: An example for Europe. J. Hydrol., 408, 235-245, 2011.

**24. Fig 3-5 are convincing.**

Thank you very much for your positive comment.

**25. Section 3.1.3. Your model simulates a too low SOC stock while the TOC and DOC concentrations look better. I would also discuss more here that the temporal pattern of observation and simulation does not match (Fig. S 8), although the mean looks promising.**

The simulated SOC actually is overall comparable to the SOC obtained from the HWSD database, which has been regarded as one of the most reliable soil database to present (Fig. S8). But in southern Europe, we indeed underestimated the SOC contents. We have indicated this in our manuscript (lines: 575-582).

We have added a discussion on the bias of the simulated temporal pattern (i.e. the overestimated seasonal variation). Please see "The simulation of the soil DOC dynamics and leaching in our model need to be further improved to better simulate the seasonal variation of riverine DOC and

TOC concentrations. The concentration of soil DOC and the DOC decomposition rate during the lateral transport process in the river network are the two key factors controlling DOC concentration in river flow.  As only a small fraction (< 20%) of the riverine DOC is decomposed during lateral transport  (Fig. 7), the overestimated (Fig. 5) seasonal amplitude in riverine DOC (and TOC) concentrations is likely caused by the uncertainties in the simulated seasonal dynamics of the leached soil DOC. The current scheme used in our model for simulating soil DOC dynamics has been calibrated against observed DOC concentrations at several sites in Europe (Camino-Serrano et al., 2018). Although the calibrated model can overall capture the average concentrations of soil DOC, it is not able to fully capture the temporal dynamics of DOC concentrations (Camino-Serrano et al., 2018).  Given this, it is necessary to collect additional observation data on the seasonal dynamics of soil DOC concentration to further calibrate the soil DOC model. In addition, averaged over the various DOC and SOC pools we distinguish in the soils, DOC represents a much more reactive fraction of soil carbon (with a turnover time of several days to a few months) than SOC (with a turnover time of decades to thousands of years). Therefore, soil DOC concentrations experience large seasonal variations, while SOC concentrations generally are much more stable and show very limited seasonal dynamics. Overall, seasonal variations in riverine POC concentrations are mainly controlled by the seasonal dynamics of soil erosion rates, rather than by the seasonal SOC dynamics, which explains a partial decoupling in the behavior of POC compared to that of DOC." (lines: 841-861)

**26. Fig 5. I would add the names of the rivers as a side panel instead of having the letters.**

We have added the name of the stations and rivers to Fig. 5.  Please see:

[Figure]

**Figure 5** Comparison between the observed and simulated concentrations of total organic carbon (TOC, a) and dissolved organic carbon (DOC, b) in river flows. The black and pink lines in each box denote the median and mean value, respectively. Box boundaries show the 25th and 75th percentiles, whiskers denote the 10th and 90th percentiles, the dots below and above each box denote the 5th and 95th percentiles, respectively.

**27. Fig. 6. I suggest to add a panel for discharge (observed vs. simulated).**

We have added a panel to show the observed vs. simulated POC discharge. Please see Figs. 6b, d, e:

[Figure]

**Figure 6** Comparison between the observed (instantaneous measurement) and simulated (monthly average value) riverine POC concentrations and POC discharge rates at three gauging sites. The histograms and error bars denote the means and standard deviations of POC concentrations, respectively. Long-term average water discharge rates at Bad Honnef, Bimmen and Rheine during the observation periods are 2023, 2100 and 80 $m^3$ $s^{-1}$, respectively.

**28. ll. 645. I am glad to read about the effect of the vegetation type more explicitly here. It could be mentioned earlier.**

Thanks to your positive comment. In section 3.1, we showed the evaluation results of our model. Then we presented the effect of vegetation type on lateral sediment and C transfers in the section 3.2. Actually, we only intend to introduce the scheme and algorithms of our model and presented the primary evaluation results of our model. To estimate the effects of vegetation cover and climate change on the spatiotemporal variation of lateral C transfer, we now have conducted a model application study. These results are actually part of a follow-up ms.

**29. l. 662. Is the difference between 3.0 Pg and 2.3 Pg significant? In the corresponding figure S11a there is a lot of fluctuations over the years.**

Yes, based on the independent sample t-test, the decline from 3.0 to 2.3 Pg yr$^{-1}$ is significant, although there is a large inter-annual variation. We have added this information in the revised manuscript. Please see "From 1901 to 1960s, the annual total sediment delivery from uplands to the whole river network of Europe declined significantly ($p<0.01$, independent sample t-test) from about 3.0 Pg yr$^{-1}$ to about 2.3 Pg yr$^{-1}$ (Fig. S13a)." (lines: 646-648)

**30. Fig 7a. The percentage for the fluxes does not sum up to 100. Why?**

Sorry for the mistake. The fraction of deposited sediment should be 63.2%. We have revised this error. Please see:

[Figure]

**Figure 7** Averaged annual lateral redistribution rate of sediment (a), POC (b), DOC (c) and $CO_2$ (d) in Europe for the period 1901-2014. $F_{sub\_DOC}$ and $F_{sub\_CO2}$ are the DOC and $CO_2$ inputs from floodplain soil (originated from the decomposition of submerged litter and soil carbon) to the overlying flooding water, respectively.

**31. l. 682. You mention several times 'small rivers'. Did you conduct a classification for the rivers or is it more a vague grouping?**

In the original line 682-684: "But although the sediment discharge rates in some  rivers in the Middle East can be as high as that in the Danube or Volga river (Fig. 8c), the POC delivery rates in these  rivers is much smaller than in the larger ones (Fig. 9c)." , the 'small river' is only a vague description, and it means the rivers that are smaller than Danube and Volga river. We find it is not necessary to repeatedly use 'small' in this sentence. Thus we have deleted the 'small' in the revised manuscript.

**32. Section 3.3. There are several strangely set parentheses.**

Sorry for the mistakes. We have double-checked the section 3.3 and the other sections of our manuscript to make sure all parentheses are used in the right way.

**33. ll. 738. I like this clear listing and explanation (first, second, third). It makes it easier to follow the argumentation.**

Thank you very much for your positive comment.

**34. ll. 752. What is the effect of anaerobic conditions in the sediment? Wouldn't the decomposition be lower then?**

Sorry for the mistake. The turnover time of litter and SOC under flooding waters is set to be one third, not three times, of the litter and SOC turnover times in upland soil. We have corrected this error in the revised manuscript. Please see "Moreover, in ORCHIDEE-$C_{lateral}$, the turnover times of litter and SOC under flooding waters (assumed to experience anaerobic condition) are set to be one third of the litter and SOC turnover times in upland soil (Reddy & Patrick Jr, 1975; Neckles & Neill, 1994; Lauerwald et al., 2017). Accounting for flooding thus decreases the decomposition rate of litter and SOC stored in floodplain soils." (lines: 732-736)

**35. Section 3.4. I think that is a necessary part of papers like that. Thank you.**

Thank you very much for your positive comment.

---

## Referee Report (RR1)

**Comment on: Estimating the lateral transfer of organic carbon through the European river network using a land surface model. By H. Zhang et al.**

Thanks to the authors for their thorough incorporation of the reviewers comments. This improved the manuscript. I only have some minor general comments and a few small further points left. After this, I believe the manuscript is suitable for publication.

**General**
I highly appreciate the insights into the discharge rates for POC, DOC and TOC (S11/12). It looks, as if the bias becomes reduced for POC, which is promising for long(er) term simulations. Could you enrich the figures by histograms (both DOC and TOC concentration & discharge rate) similar to Fig. 6 (POC) to also enable easier comparison to Fig. 5 in the main text? Can there be anything said for rivers where the model/observations mean ratio flips from smaller to larger 1 or vice versa, when comparing concentration and discharge rate (TOC,DOC, POC; e.g. happened for POC for Ems river at Rheine, where the mean concentration is under-, while the discharge rate overestimated by the model)? - is it purely a bias in measurements or is it catchment area-specific (e.g. land use, different buffering,...)? The latter might be a bit out of scope of the manuscript, though.

**Further small notes (based on the version, where changes are shown)**

p7,l163 enter the free
p10,l225 much finer
p24,l.564: Bad Honnef
p.25,l585 catchments
p41,l920 capacity? - not sure, what you want to say.

Tab S1: Description
Fig S6: f) – should it be Koeln/Köln (Cologne)? - and not Koelin?

---

## Referee Report (RR2)

Review of

**Zhang et al. Estimating the lateral transfer of organic carbon through the European river network using a land surface model.**
by: Haicheng Zhang, Ronny Lauerwald, Pierre Regnier, Philippe Ciais, Kristof Van Oost, Victoria Naipal, Bertrand Guenet, and Wenping Yuan

The authors revised the manuscript thoroughly. It is much clearer now and mistakes/errors have been removed. Most of the questions and comments I had have been answered convincingly. Thanks for that. However, some of the explanations did not make it into the manuscript. I would work on that further. Below are the points listed that should be changed.

I recommend it for publication after minor revision.

Below I list the points that only have been revised insufficiently.

My original comment:
    Is the litter also included as a part of OC input? It is mentioned a few times, but not explained or shown in Fig 1?
Author's response:
    No, litter is not included as a part of OC input. In our model (also many other erosion models), litter is an important factor (the C-factor in MUSLE, Eq. 1) for protecting soil from being eroded.
My comment on that:
    This is still not clear in the manuscript.

My original comment:
    Are the two litter pools part of the POC in soil? Please clarify.
Author's response:
    The litter pools are also organic carbon pools. However, they did not contribute to the lateral POC transfer. In our model, only soil organic carbon (SOC) contributes to the lateral transfer of POC. To avoid misunderstanding, we have revised this sentence from the original "ORCHIDEE-SOM subdivides the particulate organic carbon stored in soil into two litter pools (metabolic and structural) and three SOC pools (active, slow and passive) that differ in their respective turnover times.". to "ORCHIDEE-SOM represents two litter pools (metabolic and structural) and three SOC pools (active, slow and passive) that differ in their respective turnover times." (lines: 151-153)
My comment on that:
    Still not clear in the text. Please clarify.

My original comment:
    How does the DOC 'enter' the water? Does it depend on vegetation cover? Please clarify.
Author's response:
    When flooding water is infiltrated into soil, the DOC and CO2 in flooding water will naturally enter into soil along with the infiltrating water. The infiltration rate of flooding water depends on soil properties and soil water content, but does not depend on vegetation

cover. As this process is originally developed in the ORCHILEAK model and has been introduced in detail in Lauerwald et al. (2017), we only gave a brief overview about this part of the model.

Lauerwald, R., Regnier, P., Camino-Serrano, M., Guenet, B., Guimberteau, M., Ducharne, A., Polcher, J., and Ciais, P.: ORCHILEAK (revision 3875): a new model branch to simulate carbon transfers along the terrestrial–aquatic continuum of the Amazon basin. Geosci. Model Dev., 10, 3821-3859, 2017.

My comment on that:

Thanks for this explanation. Please add this explanation in short form to the manuscript.

My original comment:

Table 1: Is a spatial resolution of 0.5° (55km*55km max) sufficient to inform the model about the 'Area fraction of river surface'? Later it is mentioned that for the delta of the Danube the high resolution was problematic (because of gauging station data available). But is there also a problem of scale for other data? E.g. 'maximum water storage in river channel' is also only on 0.5°.

Author's response:

The area fractions of river surface or floodplain in each grid cell at 0.5° are derived from high-resolution (e.g. 3" or 90 m) topographic or satellite data (see the references in the Methods) (e.g. Fig. R1 below). Thus, these area fraction data should be reliable. Indeed, there can be many different river channels in each 0.5°×0.5° pixel in reality. However, it is almost impossible for a global land surface model to explicitly simulate the riverine processes for each individual river channel. Thus we assume that there is one virtual river channel in each 0.5°×0.5° pixel (line 346). The surface area of this virtual river is the sum of all real rivers and the flow direction of this virtual is assumed to be same to the largest real river.

My comment on that:

Thanks. This is helpful. Please add a short version of it to the manuscript.

My original question:

Is wind speed considered for the CO2 evasion?

Author's response:

In current version of our model, the effect of wind speed on CO2 evasion is not represented.

My comment on that:

Please add this information to the manuscript.

My original comment:

l. 243 mentions a 'management factor', which is only explained in l.264.

Author's response:

In the original line l.243, we only give a general explanation on the definition of Cref in MUSLE model (i.e. the cover management factor) (see lines 241-242 in the revised ms). In original line 264, we provided the specific method for calculating the cover and management factor (Cj) (see 265-268 in the revised ms).

My comment on that:

I still think that a short explanation of 'management factor' should be added at the first appearance of the phrase. Please just move '(calculated based on the fraction of surface vegetation cover)' from the second to the first occurrence.

My original comment:
ll. 250. It is not clear on which resolution the model runs. Some of the input is fine scale (e.g. 250m for floodplains), but then the results are aggregated to 0.5°. Please clarify.
Author's response:
(…) From Fig. S2, you can find the model is finally run at a spatial resolution of 0.5°. The headwater basins cover all upland areas, including not only the upland regions where the main streams originate, but also all upland regions where the tributary streams originate.
My comment on that:
I think that is a very helpful figure. I know that is mainly taken from Zhang et al. 2020. But could you move it to the main manuscript and refer to it as 'adapted after Zhang et al.'? It shows very informative how the different resolution and inputs work together.

My original comment:
ll. 442. Why do you refer and explain so much about SOC here? The section title is 'POC transport and decomposition'. Maybe some reference to the SOC section would help to clarify.
Author's response:
The scheme for simulating POC decomposition in waters follows that for SOC. With the explanation on SOC decomposition here, we intend to explain how the decomposition of POC is simulated and how we represented the accelerated POC decomposition during the transport process due to the breakdown of sediment aggregates.
My comment on that:
Please add this shortly at line 481. 'We assumed that the base 479 turnover times of active (0.3 year) and slow (1.12 years) POC pools are the same as for the 480 corresponding SOC pools. < In this paragraph we therefor refer to the scheme for SOC. >'

---

## Author Response (AR2)

Dear editor,

Thanks for sending us the comments from you and the three referees on our manuscript "*Estimating the lateral transfer of organic carbon through the European river network using a land surface model*" (esd-2022-4). We are grateful for your and the referees' constructive comments and suggested amendments. We have carefully studied them, and revised our manuscript accordingly. As a consequence, we believe that our manuscript has been considerably improved.

The following part is our detailed responses to your comments. Please note that your comments are highlighted in **bold** and followed by our responses in regular text.

Sincerely,

Haicheng Zhang, on behalf of all coauthors

Department Geoscience, Environment & Society, Université Libre de Bruxelles, 1050 Bruxelles, Belgium

Email: *haicheng.zhang@ulb.be*

**Guideline:**

Response to Referee #1: Pages 2 – 4

Response to Referee #2: Pages 5 – 10

Response to Referee #3: Pages 11 – 16

**Referee #1**

**1. General**

**I highly appreciate the insights into the discharge rates for POC, DOC and TOC (S11/12). It looks, as if the bias becomes reduced for POC, which is promising for long(er) term simulations. Could you enrich the figures by histograms (both DOC and TOC concentration & discharge rate) similar to Fig. 6 (POC) to also enable easier comparison to Fig. 5 in the main text? Can there be anything said for rivers where the model/observations mean ratio flips from smaller to larger 1 or vice versa, when comparing concentration and discharge rate (TOC,DOC, POC; e.g. happened for POC for Ems river at Rheine, where the mean concentration is under-, while the discharge rate overestimated by the model)? - is it purely a bias in measurements or is it catchment area-specific (e.g. land use, different buffering,...)? The latter might be a bit out of scope of the manuscript, though.**

Following your suggestion, we have added boxplots of simulated vs. observed TOC and DOC discharge rates per sampling location to Fig. 6. These boxplots give the statistical distributions with mean, median, inter-quartile range, $10^{th}$ and $90^{th}$ percentiles, and $5^{th}$ and $95^{th}$ percentiles. Please see Fig. 6b,c in the revised manuscript.

[Figure]

**Figure 6** Comparison between the observed and simulated concentrations of total organic carbon (TOC, a) and dissolved organic carbon (DOC, b) in river flows, as well as the discharge rates of riverine TOC and DOC. The black and pink lines in each box denote the median and mean value, respectively. Box boundaries show the 25[th] and 75[th] percentiles, whiskers denote the 10[th] and 90[th] percentiles, the dots below and above each box denote the 5[th] and 95[th] percentiles, respectively.

Indeed, for some rivers, the direction of biases in the simulated TOC (also DOC and POC) concentrations is different from that of the simulated TOC discharge rates, which also depends on the simulated water discharge rates. Uncertainties in the observation data, in the representation of river networks and in the simulated carbon and water cycles of terrestrial ecosystems in our model, as well as the omission of organic carbon inputs from manure and sewage might explain the distinct biases in simulated TOC (or DOC, POC) concentrations and discharge rates. In section 3.4, we have discussed the potential reasons for the uncertainties in our simulation results in details. Moreover, the biases of simulated TOC (or DOC, POC) concentrations or discharge rates depend on specific catchments and there is no general overestimation or underestimation in the simulation results. We also recognize that due to limited observational data, it is still a challenge to quantify the uncertainties in simulated riverine TOC, DOC and POC discharge rates across European catchments, as well as to determine the sources of these uncertainties. In our manuscript, we have now called for more continuous observation data of riverine DOC and POC to better calibrate and evaluate our model as well as reduce uncertainties (lines: 835-968).

**2. Further small notes (based on the version, where changes are shown)**
**p7,l163 enter the free**
We have revised the text following your suggestion. Please see "The products of litter and SOC decomposition enter the free DOC pool" (line: 158 )

**3. p10,l225 much finer**
We have revised the text following your suggestion. Please see "$CO_2$ evasion in inland waters is simulated using a much finer integration time step of 6 minutes." (lines: 218-219)

**4. p24,l.564: Bad Honnef**

We have revised the text following your suggestion. Please see "POC was measured at only two sites (Bad Honnef (51 measurements) and Bimmen (78 measurements)) in the Rhine catchment and one site (Rheine, 36 measurements) in the Ems catchment (Fig. S3d)." (lines: 529-531)

**5. p.25,l585 catchments**

We have revised the text following your suggestion. Please see "An over-estimation or underestimation of the catchment area by the forcing data as respectively found for the Elbe and Rhine will introduce a proportional bias in the average amount of simulated discharge from these catchments." (lines: 547-550)

**6. p41,l920 capacity? - not sure, what you want to say.**

Sorry for the confusing description. We have changed the original text "Given the difficulty to simulate the detailed hydraulic dynamics of the stream flow at large spatial scale, we thus apply a simple approach described below to calculate the sediment transport capacity" to "Given the difficulty to simulate the detailed hydraulic dynamics of the stream flow at large spatial scale, we thus apply a simple approach (Eq. 8) to calculate the sediment transport capacity" (lines: 865-867)

**7. Tab S1: Description**

We have corrected the typo of 'Description' in the supplementary Table S1.

**8. Fig S6: f) – should it be Koeln/Köln (Cologne)? - and not Koelin?**

We have revised the typo of 'Koeln' in the title of supplementary Fig. S5f (i.e. the previous Fig. S6).

**Referee #2**

**1. Comment on the revised manuscript: "Estimating the lateral transfer of organic carbon through the European river 1 network using a land surface model"**

**The manuscript greatly improved through the revision. From my point of view, the manuscript can be published with minor revisions.**

Thanks a lot for your previous comments, as well as your new suggestions and corrections below. Your comments are very helpful to improve our manuscript. We have carefully studied them and revised our manuscript accordingly. Please see our responses to your new comments below.

**2. Line 161ff: in this line you use the terminology of labile and stable DOC pools. In the following text you talk about free and absorbed DOC. Later (line 222) you use the term "refractory". Please use same terminology here.**

There are two times two categories of DOC, labile vs. refractory and free vs. adsorbed. Both labile and refractory DOC can be in the soil solution (i.e. free DOC) or adsorbed on the soil minerals (i.e. absorbed DOC). In the previous version of our ms., we sometimes used "stable" as synonym of "refractory". We acknowledge that this was confusing, and to make the terminology consistent throughout our manuscript, we have now changed 'stable DOC pool' to 'refractory DOC pool' everywhere in the text. Please see, e.g. "Soil DOC is represented by a labile and a refractory DOC pools, with a high and low turnover rate, respectively." (lines: 156-157)

**3. Line 182ff: "The adsorption, desorption, production, consumption and 184 transport of DOC within the soil column, as well as DOC export from soil to river along with surface runoff and drainage in ORCHILEAK is simulated using the same method as ORCHIDEE-SOM" ☐ Does this mean that the processes in soil are the same as in river channel? This is confusing!**

With this sentence, we intended to indicate that the method used to simulate soil DOC fluxes in ORCHILEAK is similar to that used in ORCHIDEE-SOM, which we have been introduced at the beginning of section 2.1. These soil DOC fluxes also include the export of DOC from the soil with surface runoff and drainage. These exports were already represented in ORCHIDEE-SOM,

but not transport and reaction of DOC in the river channel, the representation of which was introduced with ORCHILEAK. To give a more accurate description, we have changed this sentence to "The method used in ORCHILEAK to simulate the adsorption, desorption, production, consumption and transport of DOC within the soil column, as well as the DOC export from the soil column with surface runoff and drainage is similar to that used in ORCHIDEE-SOM." (lines: 173-176)

**4. Line 352: TC as defined below, is not the maximum load but maximum suspended sediment concentration. Furthermore, rephrase the sentence along the following line: First TC is maximum concentration. Second, if erosion or deposition occurs will depend on the actual concentration with respect to TC.**

Following your suggestion, we have rephrased this sentence. Please see "Sediment transport capacity ($TC$, g m$^{-3}$) is defined as the maximum concentration of suspended sediment that a given flow rate can carry. TC and the flow rate determine the amount of sediment that can be transported to the downstream grid cell (e.g. $F_{down2riv\_sed}$, $F_{riv2fld\_sed}$). Suspended sediment loads that are in excess to maximum possible amount of transported sediment will deposit on the river bed ($F_{rd\_sed}$). If sediment loads are below that maximum possible amount, erosion of the river bed ($F_{rero\_sed}$) or river bank ($F_{bero\_sed}$) takes place" (lines: 348-353)

**5. Line 385: please highlight that TC is expressed as suspended sediment concentration**

We have revised the text based on your suggestion. Please see "In this study, we used an empirical equation adapted from the WBMsed model, which has been proven effective in simulating the suspended sediment discharges in global large rivers (Cohen et al., 2014), to estimate the $TC$ (g m$^{-3}$) of suspended sediment in stream flow" (lines: 355-357)

**6. Line 386ff: What is the difference between daily stream flow rate and daily downstream water discharge?**

The 'stream flow rate', denoted by $q_i$ (m$^3$ s$^{-1}$) in our manuscript, is the average water flow rate on day $i$. The daily downstream water discharge, denoted by $F_{down2riv\_h20}$ (m$^3$ day$^{-1}$) is the amount of

water flowing out of the stream reservoir of a modelling grid cell to the next downstream grid cell each day. We have provided the definition of these two variables, as well as their units in our manuscript. Please see "$q_j$ (m$^3$ s$^{-1}$) is stream flow rate on day $j$, $e_l$ is an exponent depending on the upstream drainage area ($DA$, m$^2$), $F_{down2riv\_h20}$ (m$^3$ day$^{-1}$) is the daily downstream water discharge from the stream reservoir." (lines: 361-363)

**7. Line 605ff: Indicate why you use WATEM / SEDEM results to compare sediment delivery rates from your model.**

To our knowledge, there is still no large-scale observation data on sediment delivery rates from land to river networks in Europe. Therefore we compared our simulation results to the estimates from WATEM / SEDEM, which simulate soil erosion and upland deposition rates across Europe using high-resolution data of topography, soil erodibility, land cover and rainfall. The WATEM / SEDEM model has been calibrated and validated using observed sediment fluxes from 24 European catchments (Borrelli et al., 2018). We have added some texts to explain why we use the simulation results from WATEM / SEDEM. Please see "To our knowledge, there is still no large-scale observation data on sediment delivery rates from land to river networks in Europe. Therefore, following Zhang et al. (2020), the parameters *a*, *b*, *c* and *d* in Eq. 1 and 2 (Table 2) were calibrated for 57 European catchments (Fig. S3d) against the modelled sediment delivery data obtained from the European Soil Data Centre (ESDAC, Borrelli et al., 2018). The sediment delivery data from ESDAC was derived from WaTEM/SEDEM model simulations using high-resolution data of topography, soil erodibility, land cover and rainfall. This model was calibrated and validated using observed sediment fluxes from 24 European catchments (Borrelli et al., 2018)." (lines: 484-492)

**8. Line 613: 'sediment discharge rates' □ use same terminology as above (e.g. sediment delivery) and highlight that you compared with observed measurements here!**

Similar to previous publications, we actually use the 'sediment delivery' to describe the sediment transfer from land to river channel, and use the 'sediment discharge rate' to describe the sediment transfer in the river channel. That is why we have used different terminologies here. In addition, we have revised our manuscript to highlight that our simulation results of sediment discharge rates were compared with observed measurements. Please see "ORCHIDEE-C$_{lateral}$

reproduces 83% of the inter-site variation of the observed riverine sediment discharge rates across Europe (Fig. 4b)." (lines: 576-577)

**9. Line 644ff: Please shortly describe why the differences between the DBs occur.**

We have added some texts to explain the differences between SOC stocks extracted from the observation-based soil databases. Please see "We noticed that the SOC stocks extracted from these observation-based soil databases show considerable differences (vary from 106 to 249 Pg C), as they have been produced using different clusters of site-level SOC measurements and different interpolation methods to produce global gridded SOC stocks from site-level measurements (Shangguan et al., 2014; Hengl et al., 2014; Sanderman et al., 2017)." (lines: 608-612)

**10. Line 688ff: POC is a function of discharge in many river systems and in the Rhine river (see for instance Hoffmann et al 2020). It would be interesting to see rating plots of POC~discharge for measured and modelled systems. This will highlight importance differences for various flow regimes (e.g. low flow / high flow).**

Thank you for your suggestion. We agree that POC might be a function of discharge in many river systems, and the function generally follows a power law (Syvitski et al., 2000; Hoffmann et al, 2020). Actually, we have used a power law (see Eq. 8 in our manuscript) of discharge rate to calculate the sediment transport capacity of river flow, which can strongly affect riverine POC transport. We also agree that an analysis of the rating curves between POC and water discharges is very interesting and is helpful to better understand the riverine POC transfers in different flow regimes. However, we feel that this analysis is a bit out of the scope of the present study, which is mainly intended to describe our model development and its primary evaluation across all ecosystems it encapsulates. We also discuss uncertainties and shortcomings of the current version of our model.

Nonetheless, following your suggestion, we have analyzed the relationship between riverine POC concentration and river discharge rate (Fig. R1). We find that the water discharge rate cannot well explain the POC concentrations at the three sites included in our study, based on

both observation and simulation data. In addition to the amount of runoff, seasonal variations of vegetation cover, rainfall intensity and SOC content might have also significant effects on the riverine POC concentrations. Note that the simulated POC concentrations and water discharge rates shown in Fig. R1 is the monthly average values, which thus might not be able to represent the actual instantaneous relationship between POC concentration and water discharge.

[Figure]

**Figure R1** Relationship between riverine particulate organic carbon (POC) concentration and riverine water discharge rate at three sites in Europe (a: Rhine river at Bad Honnef; b: Rhine river at Bimmen; c: Ems river at Rheine).

References:

Hoffmann, T. O., Baulig, Y., Fischer, H., and Blöthe, J.: Scale breaks of suspended sediment rating in large rivers in Germany induced by organic matter. Earth Surf. Dynam., 8, 661–678, 2020.

Syvitski, J. P., Morehead, M. D., Bahr, D. B., and Mulder, T.: Estimating fluvial sediment transport: The rating parameters, Water Resour. Res., 36, 2747–2760, 2000.

**11. Line 708: Bare rock and ice are not per se indicative of low erosion rates. Typically, bare rock is observed in mountainous regions, which are characterized by high erosion rates. Ice, if associated with glaciers, is also indicative of increased rates. Please rephrase!**

We have deleted this sentence in the revised manuscript.

**12. Line 720: The Danube suspended sediment yields strongly declines due to the construction of dams, with are considered as the major sediment sinks along the Danube. If you mention the Danube in the context of sediment deposition, you should indicate the importance of dams. Please refer to Habersack et al (2016, Science of the Total Env) in this context.**

We agree that the construction of dams can strongly decrease suspended sediment yield. Omission of the representation of dams in our model might result in an underestimation of sediment deposition in river channels. We actually have discussed this issue in our manuscript. Please see "In addition, the impact of artificial dams and reservoirs on riverine sediment and carbon fluxes is also not represented in our model. Construction of dams generally leads to increased water residence time, nutrient retention, and sediment and carbon trapping in the impounded reservoir (Habersack et al., 2016; Maavara et al., 2017), and can also affect the downstream flooding regime and frequency (Mei et al., 2016; Timpe and Kaplan, 2017). Estimation by Maavara et al. (2017) suggests that the organic carbon trapped or mineralized in global artificial reservoirs is about 13% of the total organic carbon carried by global rivers to the oceans. To more accurately simulate the lateral carbon transport, we plan to include the soil and carbon redistribution within headwater basins and the effects of dams and reservoirs on riverine sediment and carbon fluxes into our model in the near future." (lines: 902-912)

Moreover, we have cited Habersack et al (2016, *Science of the Total Env*) in the revised manuscript.

**Referee #3**

**1. The authors revised the manuscript thoroughly. It is much clearer now and mistakes/errors have been removed. Most of the questions and comments I had have been answered convincingly. Thanks for that. However, some of the explanations did not make it into the manuscript. I would work on that further. Below are the points listed that should be changed.**

**I recommend it for publication after minor revision.**

**Below I list the points that only have been revised insufficiently.**

Thanks a lot for your positive feedback on our responses to most of your previous comments. We are sorry for have not sufficiently solved all of your previous concerns. We have carefully studied your points listed below and revised our manuscript accordingly. Please see our specific responses below.

**2. My original comment:**

**Is the litter also included as a part of OC input? It is mentioned a few times, but not explained or shown in Fig 1?**

**Author's response:**

**No, litter is not included as a part of OC input. In our model (also many other erosion models), litter is an important factor (the C-factor in MUSLE, Eq. 1) for protecting soil from being eroded.**

**My comment on that:**

**This is still not clear in the manuscript.**

As we explained before, litter is not a part of riverine OC input. However, litter cover can affect the cover management factor of MUSLE (denoted by $C_j$, Eq. 4), and further affect the sediment and SOC erosion rates. We assumed that no litter can be eroded and transported to the river networks. Therefore, the lateral transport of litter is not represented in the original Fig. 1 (i.e. the Fig. 2 in the revised ms.). To address your concern, we have given a more clear description of the fate of litter in the revised manuscript. Please see "Daily POC delivery to river headstream in

each 0.5° grid cell is finally simulated based on the sediment delivery rate and the average SOC concentration of surface soil layers (0-20 cm). We assumed that litter cannot be eroded and transported to the river network, however, it can affect soil erosion rate through the cover management factor of the MUSLE model (denoted by $C_j$, Eq. 4)." (lines: 290-294)

And "$C_j$ (0-1, unitless) is the daily actual cover management factor, calculated based on the fraction of surface vegetation cover, the amount of litter stock and the biomass of living roots in each PFT within each 0.5°×0.5° grid cell." (lines: 285-288)

**3. My original comment:**

**Are the two litter pools part of the POC in soil? Please clarify.**

**Author's response:**

**The litter pools are also organic carbon pools. However, they did not contribute to the lateral POC transfer. In our model, only soil organic carbon (SOC) contributes to the lateral transfer of POC. To avoid misunderstanding, we have revised this sentence from the original "ORCHIDEE-SOM subdivides the particulate organic carbon stored in soil into two litter pools (metabolic and structural) and three SOC pools (active, slow and passive) that differ in their respective turnover times.". to "ORCHIDEE-SOM represents two litter pools (metabolic and structural) and three SOC pools (active, slow and passive) that differ in their respective turnover times." (lines: 151-153)**

**My comment on that:**

**Still not clear in the text. Please clarify.**

We actually use POC to describe the particulate organic carbon in river streams. The riverine POC is contributed by SOC pools, without any contribution from upland litter pools. Or we can say that the particulate organic carbon in soil is called SOC, and the particulate organic carbon in streams is call POC. We have clearly indicated this in the revised manuscript (lines: 285-294). Please see our response to your comment #2.

**4. My original comment:**

**How does the DOC 'enter' the water? Does it depend on vegetation cover? Please clarify.**

**Author's response:**

**When flooding water is infiltrated into soil, the DOC and CO2 in flooding water will naturally enter into soil along with the infiltrating water. The infiltration rate of flooding water depends on soil properties and soil water content, but does not depend on vegetation cover. As this process is originally developed in the ORCHILEAK model and has been introduced in detail in Lauwerwald et al. (2017), we only gave a brief overview about this part of the model.**

**Lauerwald, R., Regnier, P., Camino-Serrano, M., Guenet, B., Guimberteau, M., Ducharne, A., Polcher, J., and Ciais, P.: ORCHILEAK (revision 3875): a new model branch to simulate carbon transfers along the terrestrial–aquatic continuum of the Amazon basin. Geosci. Model Dev., 10, 3821-3859, 2017.**

**My comment on that:**

**Thanks for this explanation. Please add this explanation in short form to the manuscript.**

We have added this explanation to the revised manuscript. Please see "DOC and $CO_2$ in flooding waters can enter into soil DOC and $CO_2$ pools along with the flooding water infiltrated into soil. The infiltration rate of flooding water depends on soil properties and soil water content, but does not depend on vegetation cover." (lines: 190-191)

**5. My original comment:**

**Table 1: Is a spatial resolution of 0.5° (55km*55km max) sufficient to inform the model about the 'Area fraction of river surface'? Later it is mentioned that for the delta of the Danube the high resolution was problematic (because of gauging station data available). But is there also a problem of scale for other data? E.g. 'maximum water storage in river channel' is also only on 0.5°.**

**Author's response:**

**The area fractions of river surface or floodplain in each grid cell at 0.5° are derived from high-resolution (e.g. 3″ or 90 m) topographic or satellite data (see the references in the Methods) (e.g. Fig. R1 below). Thus, these area fraction data should be reliable. Indeed, there can be many different river channels in each 0.5°×0.5° pixel in reality. However, it is almost impossible for a global land surface model to explicitly simulate the riverine processes for each individual river channel. Thus we assume that there is one virtual river channel in each 0.5°×0.5° pixel (line 346). The surface area of this virtual river is the sum of all real rivers and the flow direction of this virtual is assumed to be same to the largest real river.**

**My comment on that:**

**Thanks. This is helpful. Please add a short version of it to the manuscript.**

Following your suggestion, we have added these contents to the revised manuscript. Please see "Note that the maximum area fractions of river surface and floodplain in each basin (i.e. each 0.5°×0.5° grid cell in this study) are derived from high-resolution topographic data (Table 1). As it is difficult to explicitly represent all real river channels in a global land surface model (due to the limit of computing efficiency of current computers), we assume that there is one virtual river channel in each 0.5°×0.5° pixel. The surface area of this virtual river is the sum of all real rivers and the flow direction of this virtual is assumed to be same to the largest real river (Lauerwald et al., 2015)." (lines: 179-185)

**6. My original question:**

**Is wind speed considered for the CO2 evasion?**

**Author's response:**

**In current version of our model, the effect of wind speed on CO2 evasion is not represented.**

**My comment on that:**

**Please add this information to the manuscript.**

We have added this information in the revised manuscript. Please see "The effect of wind speed on $CO_2$ evasion is not represented in the current version of ORCHILEAK." (lines: 223-226)

**7. My original comment:**

**l. 243 mentions a 'management factor', which is only explained in l.264.**

**Author's response:**

**In the original line l.243, we only give a general explanation on the definition of Cref in MUSLE model (i.e. the cover management factor) (see lines 241-242 in the revised ms). In original line 264, we provided the specific method for calculating the cover and management factor (Cj) (see 265-268 in the revised ms).**

**My comment on that:**

**I still think that a short explanation of 'management factor' should be added at the first appearance of the phrase. Please just move '(calculated based on the fraction of surface vegetation cover)' from the second to the first occurrence.**

C in the MUSLE model represents the cover management factor, which depends on the vegetation cover and storage of plant debris. In different studies, the C-factor can be calculated using different equations. At the first appearance of C-factor, we intended to give a general explanation of its meaning in the MUSLE model, and also provide the preset value of the C-factor at the reference state (i.e. $C_{ref} = 0.1$, Eq. 1). In the second instance where C-factor appears (i.e. $C_j$, Eq. 4), we introduced the specific method for calculating $C_j$ in our model. Nonetheless, to make the readers better understand the meaning of the C-factor on its first occurrence in the text, we have now added some text. Please see "$C_{ref}$ (0-1, dimensionless) in Eq. 1 represents the cover management factor which depends on vegetation cover and storage of plant debris (see below). The value of $C_{ref}$ is set to 0.1 for the reference state." (lines: 253-254)

**8. My original comment:**

**ll. 250. It is not clear on which resolution the model runs. Some of the input is fine scale (e.g. 250m for floodplains), but then the results are aggregated to 0.5°. Please clarify.**

**Author's response:**

**(…) From Fig. S2, you can find the model is finally run at a spatial resolution of 0.5°. The headwater basins cover all upland areas, including not only the upland regions where the main streams originate, but also all upland regions where the tributary streams originate.**

**My comment on that:**

**I think that is a very helpful figure. I know that is mainly taken from Zhang et al. 2020. But could you move it to the main manuscript and refer to it as 'adapted after Zhang et al.'? It shows very informative how the different resolution and inputs work together.**

Following your suggestion, we have moved the original Fig. S2 to the main text. Please see:

[Figure]

**Figure 1** Upscaling scheme used in ORCHIDEE-MUSLE (Zhang et al., 2020) and ORCHIDEE-C$_{lateral}$ for calculating the sediment delivery rate from headwater basins to river networks. MUSLE is the Modified Universal Soil Loss Equation; DEM is the digital elevation model (m); $K$ is the soil erodibility factor (Mg MJ-1 mm-1); $R_{ref}$ is the assumed reference daily runoff depth (= 10 mm day$^{-1}$); $R_{30\_ref}$ is the assumed reference maximum 30-minutes runoff depth (= 1 mm 30-minutes$^{-1}$); $C_{ref}$ (= 0.1, dimensionless) is the assumed reference cover management factor; $R_{iday}$, $R_{30\_iday}$ and $C_{iday}$ are the simulated daily total runoff depth, daily maximum 30-minutes runoff depth and daily cover management factor, respectively. This figure is adapted from the Fig. 1 in Zhang et al. (2020).

**9. My original comment:**

**ll. 442. Why do you refer and explain so much about SOC here? The section title is 'POC transport and decomposition'. Maybe some reference to the SOC section would help to clarify.**

**Author's response:**

**The scheme for simulating POC decomposition in waters follows that for SOC. With the explanation on SOC decomposition here, we intend to explain how the decomposition of POC is simulated and how we represented the accelerated POC decomposition during the transport process due to the breakdown of sediment aggregates.**

**My comment on that:**

**Please add this shortly at line 481. 'We assumed that the base 479 turnover times of active (0.3 year) and slow (1.12 years) POC pools are the same as for the 480 corresponding SOC pools. < In this paragraph we therefor refer to the scheme for SOC. >'**

We have added this information to the revised manuscript. Please see "The representation of POC deposition and transformation in the aquatic reservoirs and bed sediment involve as well decomposition, which follows largely the scheme used for SOC (Fig. 2a)." (lines: 436-438)